# Differential adhesion during development establishes individual neural stem cell niches and shapes adult behaviour in *Drosophila*

Agata Banach-Latapy[1], Vincent Rincheval[2], David Briand[1], Isabelle Guénal[2], Pauline Spéder[1]*

1 Institut Pasteur, Université Paris Cité, CNRS UMR3738, Structure and Signals in the Neurogenic Niche, Paris, France, 2 Université Paris-Saclay, UVSQ, LGBC, 78000, Versailles, France

* pauline.speder@pasteur.fr

**Data Availability Statement:** All relevant data are within the paper, its Supporting Information files and the Data Source (S1 Data) files. The R script used to perform a Generalised Linear Model

## Abstract

Neural stem cells (NSCs) reside in a defined cellular microenvironment, the niche, which supports the generation and integration of newborn neurons. The mechanisms building a sophisticated niche structure around NSCs and their functional relevance for neurogenesis are yet to be understood. In the *Drosophila* larval brain, the cortex glia (CG) encase individual NSC lineages in membranous chambers, organising the stem cell population and newborn neurons into a stereotypic structure. We first found that CG wrap around lineage-related cells regardless of their identity, showing that lineage information builds CG architecture. We then discovered that a mechanism of temporally controlled differential adhesion using conserved complexes supports the individual encasing of NSC lineages. An intralineage adhesion through homophilic Neuroglian interactions provides strong binding between cells of a same lineage, while a weaker interaction through Neurexin-IV and Wrapper exists between NSC lineages and CG. Loss of Neuroglian results in NSC lineages clumped together and in an altered CG network, while loss of Neurexin-IV/Wrapper generates larger yet defined CG chamber grouping several lineages together. Axonal projections of newborn neurons are also altered in these conditions. Further, we link the loss of these 2 adhesion complexes specifically during development to locomotor hyperactivity in the resulting adults. Altogether, our findings identify a belt of adhesions building a neurogenic niche at the scale of individual stem cell and provide the proof of concept that niche properties during development shape adult behaviour.

## Introduction

Stem cells are multipotent progenitors driving the growth and regeneration of the tissue they reside in through the generation of differentiated cells. Their localisation within the tissue is restricted to carefully arranged cellular microenvironments, or niches, which control their maintenance and activity in response to local and systemic cues [1–3]. The niches comprise the stem cell themselves, their newborn progeny, and a number of cells of various origins and

analysis is available on Zenodo, DOI: 10.5281/zenodo.8426002.

**Funding:** This work has been funded by a starting package from Institut Pasteur/ LabEx Revive (ANR-10-LABX-0073), a JCJC grant from Agence Nationale de la Recherche (NeuraSteNic, ANR-17-CE13-0010-01) to P.S. and a Projet Fondation ARC from the Association pour la Recherche contre le Cancer to P.S. and A.B-L. was supported by a post-doctoral fellowship from the LabEx Revive (ANR-10-LABX-0073). The funders had no role in study design, data collection and analysis, decision to publish, or preparation of the manuscript.

**Competing interests:** The authors have declared that no competing interests exist.

**Abbreviations:** ADHD, attention-deficit/hyperactivity disorder; ALH, after larval hatching; Arm, Armadillo; ATPα, Na K-ATPase pump; CB, central brain; CG, cortex glia; CNS, central nervous system; Cont, Contactin; Cora, Coracle; Dlg1, Discs large; DSCP, *Drosophila* synthetic core promoter; GMC, ganglion mother cell; imINP, immature INP; INP, intermediate neural progenitor; mINP, mature INP; LD, light:dark; MARCM, Mosaic Analysis with a Repressible Cell Marker; Nrg, Neuroglian; Nrx-IV, Neurexin-IV; NSC, neural stem cell; OL, optic lobe; PH3, phospho-histone 3; Shg, Shotgun; VNC, ventral nerve cord.

roles that support stem cell decisions. The diversity of cellular shapes and roles requires a precise spatial organisation to enable proper niche function towards all and every stem cells. Within the central nervous system (CNS) in particular, a highly structured organ dependent on the tight arrangement of cellular connections, the neural stem cell (NSC) niches are anatomically complex microenvironments that must form within such constraint. They comprise multiple cell types such as neurons, various glial cells, vasculature and immune cells [4,5], which are precisely organised with respect to NSCs. While studies have focused on the identification of signalling pathways operating in an established niche and controlling neurogenesis [5–7], how the niche is first spatially built around NSCs, and the importance of its architecture on neurogenesis, from stem cell division to the integration of the newborn neurons, are poorly understood.

The *Drosophila* larval CNS offers a genetically powerful model to study interactions within the niche in vivo. Similar to mammals, *Drosophila* NSCs, historically called neuroblasts, self-renew to produce neuronal and glial progeny, and their behaviour is controlled by their niche, an exquisitely organised yet less complex structure than its mammalian counterpart.

Fly NSCs are born during embryogenesis, during which they cycle to generate primary neurons in a first wave of neurogenesis. They then enter quiescence, a mitotically dormant phase from which they exit to proliferate through the activation of PI3K/Akt signalling in response to nutrition [8,9]. This postembryonic, second wave of neurogenesis generates secondary neurons that will make up 90% of the adult CNS and lasts until the beginning of pupal stage. NSCs finally differentiate or die by apoptosis after pupariation. Larval NSCs populate the different regions of the CNS, namely, the ventral nerve cord (VNC), the central brain (CB), and the optic lobe (OL) (Fig 1A). They nevertheless display distinct properties, mainly through different modes of division and expression of specific transcription factors (Fig 1B) [10]. Type I NSCs reside in the CB and VNC and divide asymmetrically to generate a smaller ganglion mother cell (GMC). GMCs further terminally divide to produce 2 neurons. Type II NSCs, found exclusively in the CB, represent a smaller population with only 8 cells per hemisphere [11–13]. Type II NSC self-renewal produces an intermediate neural progenitor (INP), which undergoes a limited number of asymmetric divisions to produce GMCs that will subsequently divide to give neurons.

These different NSCs are embedded within a sophisticated, multilayered niche made of different cell types (Fig 1C and 1D). The blood–brain barrier forms the interface with the systemic environment and controls NSC reactivation [14] and proliferation [15]. A specific glial subtype, the cortex glia (CG), is in close contact with NSCs and their progeny and is crucial for NSC proliferation and survival [16–20], resistance to stress [21,22], as well as the survival of newborn neurons [23]. Remarkably, the CG form a seemingly continuous glial network that invades the whole CNS (Fig 1C and 1D) while building bespoke encasing of entire NSC lineages (comprising NSC, GMC, and newborn neurons, as well as INP for Type II NSCs), called CG chambers [23–25]. CG also enwrap individual primary neurons and, later on, older, mature secondary neurons (Fig 1D). CG network is progressively built around NSC during larval development in a process that parallels NSC behaviour [23,25–27] (Fig 1E). CG cells, born during embryogenesis, do not form a continuous meshwork nor encase quiescent NSCs at larval hatching (0 h after larval hatching, ALH0). Rather, they initiate growth in response to nutrition, via autonomous activation of the PI3K/Akt pathway, leading to an increase in membrane density yet without NSC encasing. Then, at the time NSCs start dividing (ALH24 to ALH48), CG enwrap individual NSCs, forming a typical chequerboard structure. They further extend their processes to maintain a fitted chamber structure during neuronal production (ALH72 to ALH96). CG thus wrap entire NSC lineages (Fig 1F and 1G, orthogonal view), forming an individual chamber containing one NSC (Fig 1F'–1G', view at the NSC level) and resulting maturing neuronal progeny (Fig 1F"–1G", view at the neuron level). The cell bodies

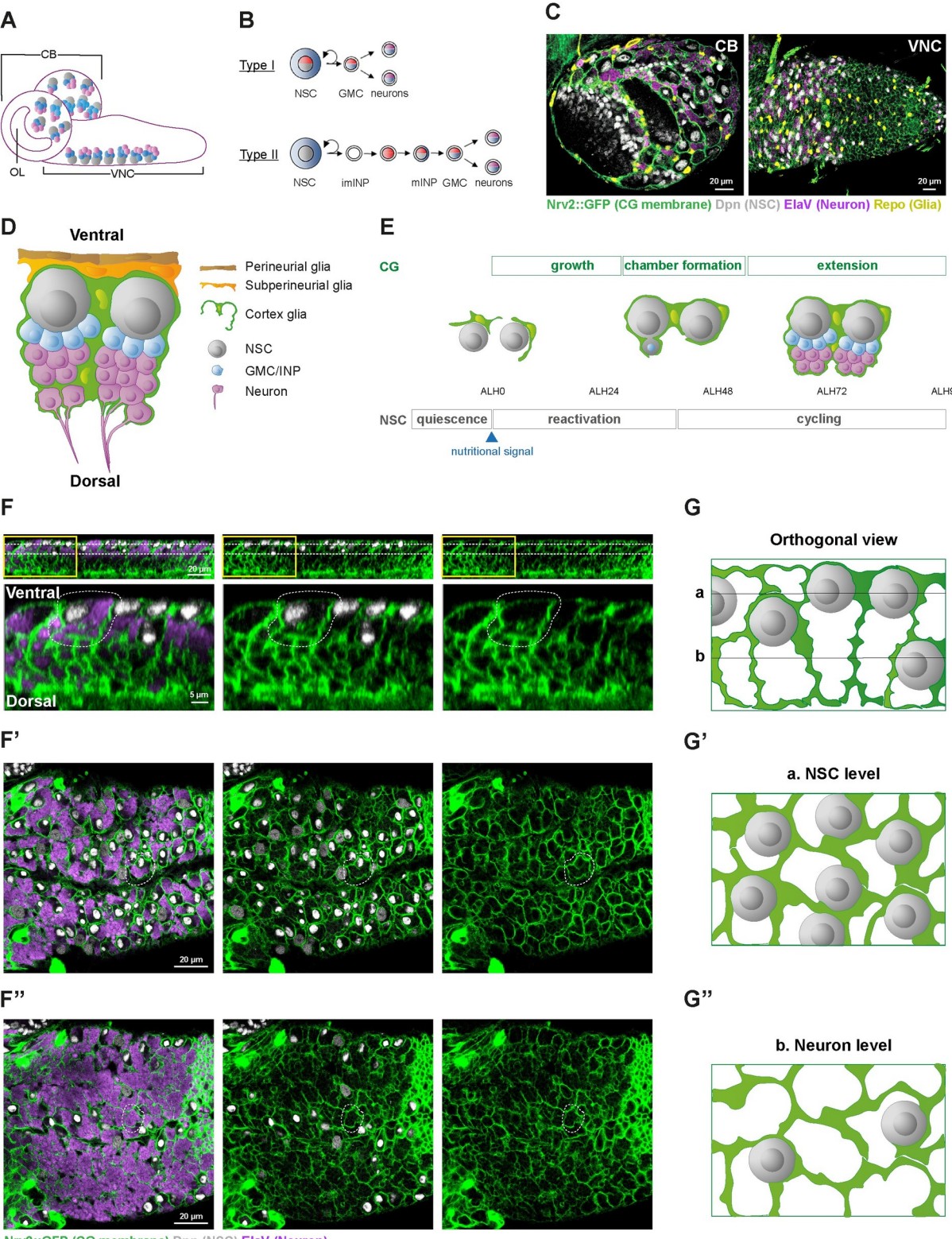

**Fig 1. The CG encase each NSC and its neuronal lineage in an individual membrane chamber.** (**A**) Schematic of the *Drosophila* larval CNS depicting the localisation of the NSC lineages. The 2 main neurogenic regions are the CB, comprising 2 hemispheres, and the VNC. The OL corresponds to the precursor of the visual system and is organised as a neuroepithelial tissue, which will undergo conversion into NSCs. (**B**) Schematic of the Type I and Type II NSC lineages. Type II lineages are only found in the CB, with a number of 8 per hemisphere. Type I

lineages populate both the CB and the VNC. Type I NSCs divide asymmetrically to give birth to a GMC, which itself divides once to generate 2 neurons or glial cells. Type II NSCs self-renew while generating one imINP. The imINP will mature in mINP, which will then divide to generate a GMC. (**C**) Confocal pictures representing the dorsal region of the larval CB (left panel) and ventral region of the VNC (right panel) 72 h after larval hatching (ALH72, at 25˚C) labelled with markers for the CG membrane (*Nrv2::GFP*, green), glia nuclei (anti-Repo, yellow), NSC (anti-Dpn, grey), and neurons (anti-ElaV, magenta). (**D**) Schematic of the NSC niche (orthogonal view), made by the PG (brown), SPG (orange), CG (green), NSCs (grey), GMCs/INPs (blue) and N (light purple). One entire NSC lineage, composed of one NSC and its immature neuronal progeny, is encased within a seemingly continuous layer of CG membrane, forming a chamber. (**E**) Timeline of the encasing of NSC lineages by CG, parallelling NSC behaviour. At the beginning of larval stage (0 h after larval hatching, ALH0), NSCs are quiescent and not individually encased by the CG, whose network is not formed. Upon larval feeding, both NSCs and CG grow. Individual encasing of NSC (chamber formation) correlates with the time of its first division, the final step of NSC reactivation. The CG will then keep growing (extension) to adapt to the production of newborn secondary neurons and the increase in lineage size until the end of larval stage. (**F, G**) The CG chambers individually encase whole NSC lineages, including the NSC (NSC level) and its newborn neuronal progeny (neuron level) throughout its depth (orthogonal view). (**F-F”**) Confocal pictures of a VNC labelled with markers for the CG membrane (*Nrv2::GFP*, green), NSCs (anti-Dpn, grey), and neurons (anti-ElaV, light purple). The orthogonal view (**F**) shows the overall cellular organisation along the dorso-ventral axis of the VNC, with NSCs mostly localised ventral, and neuronal progeny below, more dorsal. Yellow boxes correspond to the close-up panels below. Dashed white lines indicate one NSC chamber. Horizontal white dashed lines indicate the planes of the (**F'**) NSC and (**F”**) neuron levels. (**G-G”**) Schematics representing the respective pattern of NSC and CG membrane for the different views (orthogonal, NSC level, and neuron level) for a group of NSCs. Horizontal black lines in (**G**) indicate the planes of the (**G'**) NSC and (**G”**) neuron levels. ALH, after larval hatching; CB, central brain; CG, cortex glia; CNS, central nervous system; GMC, ganglion mother cell; imINP, immature intermediate neural progenitor; INP, intermediate neural progenitor; mINP, mature intermediate neural progenitor; N, neuron; NSC, neural stem cell; OL, optic lobe; PG, perineurial glia; SPG, subperineurial glia; VNC, ventral nerve cord.

of newborn neurons from one NSC lineage are thus initially found clustered together in one CG chamber, and, as they mature, they will become individually encased by the CG [25,28]. Newborn, immature neurons from a same lineage start to extend axonal projections, which are fasciculated together as a bundle and are also encased by the CG (Figs 1D and S1A–S1A') until they enter the neuropile, a synaptically dense region devoided of cell bodies, where axons connect [28–30] and are surrounded by other glial types [31]. The repeated pattern of CG chambers thus translates both in term of cell bodies and axonal tracts.

The reliable formation of such precise chequerboard structure implies that CG integrate proper cellular cues to encase specific cells, while navigating between a density of diverse cell types. However, the nature of these cues and the importance of such stereotyped encasing of NSC lineages on NSC activity and the function of neuronal progeny remained to be identified.

Here, we investigated the cellular cues driving the correct establishment of a structurally sophisticated and functional CG niche around individual NSC lineages and their impact on neurogenesis. We found that CG are able to group together clonally related cells regardless of their individual identity. Further, we discovered that lineage information and individual encasing are mediated by the existence of multiple adhesion complexes within the niche. First, the cell adhesion protein Neurexin-IV is expressed and crucial in NSC lineages to maintain their individual encasing, through its interaction with Wrapper, a protein with immunoglobulin domains present in the CG. The loss of Neurexin to Wrapper interaction results in large, defined CG chambers containing multiple NSC lineages. In parallel, Neuroglian appears to form strong homophilic interactions between cells of the same lineages, binding them together by providing higher adhesion compared to the weaker interaction between CG and NSC lineages. In absence of Neuroglian, NSC lineages are clumped together in a random fashion. As such, differential adhesion is a core mechanism of NSC lineage encasing. Adherens junctions are also present in NSC lineages; however, they appear mostly dispensable for individual encasing. In addition, Neurexin-IV and Neuroglian adhesions are important for correct axonal projections in the developing CNS. Further, we demonstrated that the loss of Neurexin-IV/ Wrapper and Neuroglian adhesions specifically during development causes a hyperactive loco-motor behaviour in the adult. Our findings unravel a principle of NSC niche organisation based on differential adhesion and link the adhesive property of the niche and NSC lineages during development to adult neurological behaviour.

## Results

### Cortex glia encase lineage-related cells regardless of their identity

During niche formation, the first cell encased within a CG chamber is the NSC, suggesting that cell identity, specifically being a stem cell, may signal individual encasing. Entire NSC lineages, which can be tracked and controlled with the *worniu-GAL4* (*wor >*) driver [32] from the NSC to immature neurons (S1B and S1C Fig), are ultimately encased later on.

To assess the importance of stem cell identity in chamber formation, we took advantage of genetic alterations known to dysregulate NSC division and differentiation and to lead to the formation of tumour-like, NSC-only, lineages [33]. In particular, *pros* knockdown in Type I lineages converts GMC into NSC-like, Dpn+ cells at the expense of neurons [34]. Surprisingly, in these conditions, we found that CG chambers contained not one, but several NSC-like, Dpn+ cells (Fig 2A–2A'). Similar results were obtained for other conditions that lead to Type I NSC-only lineages, including GMC dedifferentiation via Dpn overexpression or loss of asymmetric division via membrane-tethered aPKC overexpression (S2A Fig). We then asked how CG would adapt to the dysregulation of Type II NSCs, which generate bigger lineages than Type I NSCs. Since CG chamber formation was precisely described only for Type I NSCs [23], we first checked the dynamics of CG morphogenesis around Type II and found that they followed similar steps, albeit in a slower fashion (S2B Fig). We then knocked down the cell fate determinant *brat* [35,36], which is necessary for preventing immature INP (imINP) dedifferentiation into NSC-like cells. This led to the formation of large tumours (Fig 2B–2B'). CG were able to adapt to cell overproliferation, at least until this stage, and enwrapped many NSC-like cells within one chamber. These data show that both for Type I and Type II NSCs, stem cell identity is not sufficient to ensure their sorting into individual CG chambers.

We then wondered whether tumour NSCs grouped within one chamber originated from the same mother NSC or had been encased randomly independently of their lineage of origin. To do so, we used a multicolour clonal analysis to label individual NSC lineages. The Raeppli system [37] results in the stochastic and irreversible labelling of a cell at the time of induction, allowing to mark and track a mother cell and its colour-sharing progeny. Heatshock-controlled induction before NSC reactivation of a nuclear tagged version of Raeppli (Raeppli-NLS) ensured most of lineages could be fully tracked, except a few in which the induction had failed. We first confirmed that cells found within each CG chamber belonged to the same lineage for wild-type Type I and Type II NSC lineages (Fig 2C and 2D), a property previously assumed [24,25] or reported for specific lineages only [23]. For Type I *pros* tumours, we found that clonal tumour-like growth coming from single dysregulated NSCs (marked by one colour) were contained within one CG chamber (Fig 2C). In *brat* tumours, most Type II chambers, found in a similar number to control (S2C and S2D Fig), showed a single, large tumour-like growth of one colour (Fig 2D). Yet, we sometimes observed multiple colours within a single CG chamber, mostly as scattered cells or small clones among unmarked cells (S2E and S2F Fig). We inferred this comes from later uncontrolled inductions of Raeppli-NLS in individual tumour NSCs from lineages in which recombination was not successfully induced during heatshock. This would result in differently marked NSC-like Dpn+ clones yet coming from the same Type II NSC mother cell. In support of this interpretation, we observed Raeppli-NLS stainings (isolated cells or small clones) in *brat* tumour without heatshock (S2E and S2F Fig), pinpointing a leakiness in the induction control for this background, which we were not able to bypass technically.

To strengthen our findings on *brat* RNAi, we then used a Coin-FLP clonal analysis [38] to generate *brat RNAi* clones in Type II NSC lineages (induced by *grh^D4^-FLP*, expressed in most NSCs from late embryogenesis; see Methods and S1 Table). We found that 100% of Type II *brat* RNAi clones were individually encased (Fig 2E).

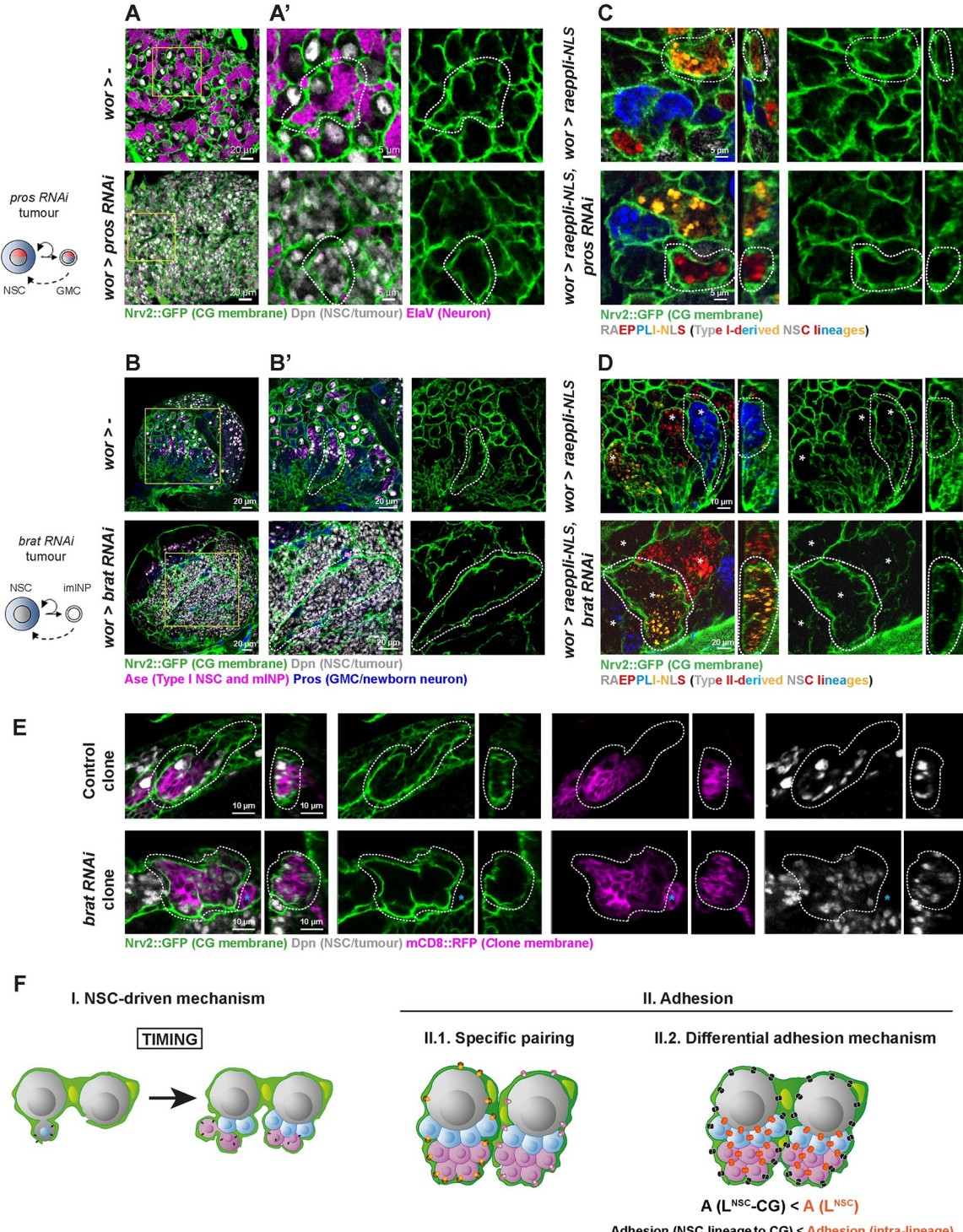

**Fig 2. CG individually encase clonally related cells regardless of their identity.** (**A**, **B**) Adaptation of the CG network to NSC tumours. Type I *pros* (*Nrv2::GFP, wor-GAL4; tub-GAL80$^{ts}$ > pros RNAi*) and Type II *brat* (*Nrv2::GFP, wor-GAL4; tub-GAL80$^{ts}$ > brat RNAi*) tumours were induced, and the organisation of CG membrane was monitored by *Nrv2::GFP* (green). Control line for both (*Nrv2::GFP, wor-GAL4; tub-GAL80$^{ts}$ > w$^{1118}$*). For *pros*, larvae were dissected after 48 h at 18˚C, followed by 2 h heatshock at 37˚C and 48–52 h at 29˚C. For *brat*, larvae were dissected after 72 h at 29˚C. See S1 Table for detailed genetics, timing, and conditions of larval rearing. (**A**) Left panel shows a representative whole thoracic VNC for control (*n* = 10 VNCs) and *pros* tumour (*n* = 10 VNCs), and (**A'**) middle and right panels show a close-up of the yellow boxes in (**A**). NSCs/tumour NSCs are labelled with anti-Dpn (grey), and neuron with anti-ElaV (magenta). (**B**) Left panel shows a whole representative CB for control (*n* = 10 CBs) and *brat* tumour (*n* = 10 CBs), and (**B'**) middle

and right panels show a close-up of the yellow boxes in (**B**). NSCs/tumour NSCs are labelled with anti-Dpn (grey), Type I NSC and mINPs with anti-Asense (Ase, magenta), and GMCs plus newborn neurons with anti-Prospero (Pros, blue). For both, dashed white lines delineate one CG chamber. (**C, D**) Relationship between individual encasing and cell identity. Type I *pros* (*wor > raeppli-NLS, pros RNAi*) and Type II *brat* (*wor > raeppli-NLS, brat RNAi*) tumours were induced together with the multicolour lineage tracing Raeppli-NLS. One out of 4 colours (blue, white, orange, and red) is stochastically induced in the transformed NSCs using *hs-Flp*. For *pros*, larvae were dissected after 24 h at 18°C, followed by 2 h heatshock at 37°C and 48–52 h at 29°C. For *brat*, larvae were dissected after 48 h at 18°C, followed by 2 h heatshock at 37°C and 48–52 h at 29°C. See S1 Table for detailed genetics, timing, and conditions of larval rearing. (**C**) Top and orthogonal close-up views of representative control (*n* = 10 VNCs) and *pros* (*n* = 9 VNCs) Type I lineages marked with Raeppli-NLS. (**D**) Top and orthogonal close-up views of representative control (*n* = 8 CBs) and *brat* (*n* = 7 CBs) Type II lineages marked with Raeppli-NLS. For both, CG membrane is visualised with *Nrv2::GFP* (green). Dashed white lines delineate one CG chamber. White stars indicate Type II lineages. (**E**) *brat* RNAi tumours are individually encased by the CG. Representative confocal pictures of the CG network in control Type II NSC clone and *brat* RNAi Type II NSC clone. Top view and orthogonal views are shown. The CoinFLP system was used to generate wild-type and *brat* RNAi clones in the Type II NSCs, using *grh^{D4}-FLP* for induction. The *grh^{D4}* enhancer drives in most NSCs from late embryonic stage. As such, clones in Type I NSCs can also be induced (marked by a cyan star) yet do not lead to any tumour. Larvae were dissected after 72–96 h at 25°C. See S1 Table for detailed genetics, timing, and conditions of larval rearing. The membrane of the clone is marked by mCD8::RFP (magenta), and CG membrane is visualised with *Nrv2::GFP* (green). Dashed white lines delineate one Type II NSC chamber. *n*(*brat* RNAi clones) = 20 clones in 15 CBs and *n*(control clones) = 15 clones in 11 CBs. All (100%) of the Type II NSC clones in both control and *brat* RNAi conditions were found individually encased by the CG. (**F**) Schematic of the different hypotheses explaining individual encasing of NSC lineages by CG. Panel I depicts the NSC-driven timing of NSC encapsulation, prior to lineage generation, as the instructive cue. Panel II describes the use of adhesion mechanisms: (1) specific CG to NSC lineage and (2) generic, based on difference in strength between intralineage adhesion (A(L^{NSC})) and adhesion linking NSC lineages and CG (A(L^{NSC}-CG)); in this case, the prediction is that A(L^{NSC}-CG) < A(L^{NSC}). CB, central brain; CG, cortex glia; GMC, ganglion mother cell; NSC, neural stem cell; VNC, ventral nerve cord.

Altogether, these results demonstrate that cells from one NSC lineage, whether NSC, GMC, INP, or neurons, are kept together within one CG chamber regardless of their identity and suggest that NSC lineages contain intrinsic information allowing their individual encasing by the CG.

## Individual encasing relies on intrinsic lineage cues

Chamber completion around NSCs occurs around the time of first division and is driven by NSC reactivation [23,26]. A simple explanation for keeping a lineage together and separated from others would thus be the sequential addition of newborn cells within a compartment already defined from the start by NSC-derived signals (Fig 2F, Panel I). Timing would thus be the instructive cue. Indeed, previous studies have shown that signals from reactivated NSCs are paramount to form CG niches [23], although the cues integrated by the glia have not been identified. Within this hypothesis, blocking CG morphogenesis until well after the first NSC division, followed by subsequent release, would result in aberrant chamber formation and random encasing of neurons from different lineages.

To test this hypothesis, we conditionally overexpressed the PI3K/Akt pathway inhibitor PTEN [23] to specifically block CG growth, using the binary QF system (whose repression can be chemically controlled), while Raeppli-NLS expression in NSC lineages was driven by the GAL4/UAS system (Fig 3A). Raeppli-NLS was induced at ALH0, and CG growth was impaired until NSCs cycled actively and neuronal progeny had been already produced (time T1). While CG membranes still infiltrated in between, they failed to form correct chambers to separate individual lineages from each other at this time (S3A and S3B Fig). We then allowed CG to resume their growth and later observed the establishment of a stereotyped chequerboard pattern, with most of the chambers containing a single NSC and its neuronal progeny (Fig 3B, time T2). This was confirmed by quantifying the chambers containing more than one NSC lineage (Methods; Fig 3C). These results thus show that the information directing individual encasing by CG is not exclusive to the NSC but is inherited by its progeny and can be sensed by CG later during larval development.

One elegant way cells could be kept together would be their physical binding through adhesion mechanisms. We hypothesised 2 possibilities (Fig 2F, Panel II). The most complex

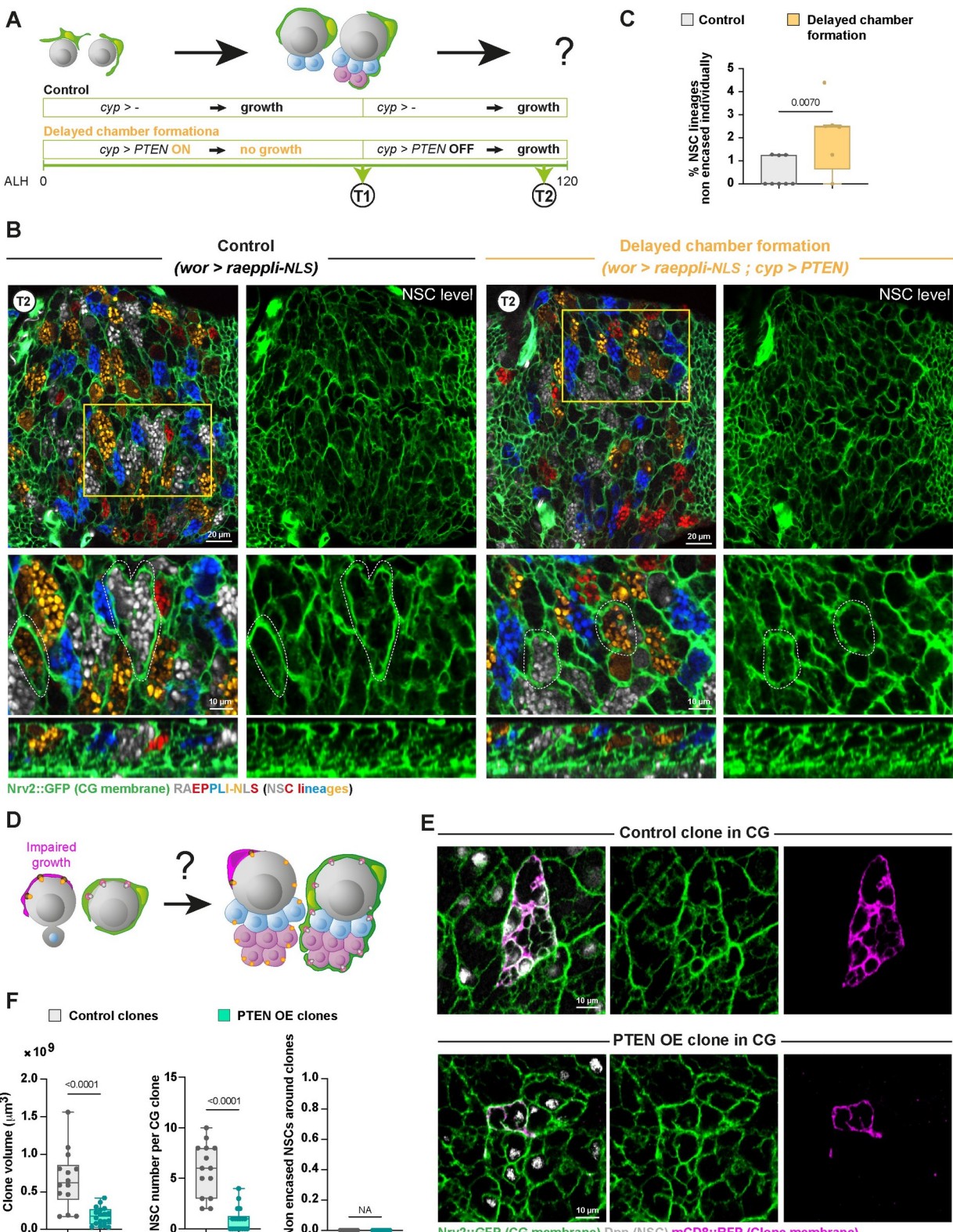

**Fig 3. CG use intrinsic and generic NSC lineage cues to individually encase NSC lineages.** (**A**) Schematic of the timing and genetic conditions used to probe the importance of NSC-driven timing for the encapsulation of individual NSC lineages. To delay chamber formation around NSC lineages ("Delayed chamber formation" condition), CG growth is initially blocked using PTEN expression in the CG through the use of the Q

system (*cyp-QF2* driving *QUAS-PTEN* and combined with *tub-QS* under quinic acid–supplemented food). At T1 (72 h at 29°C from ALH0), CG growth is allowed by preventing PTEN expression (no quinic acid in the food), and CG structure is assessed at T2 (24–30 h at 29°C from T1, so after 100 h at 29°C from ALH0). In the control condition, QUAS-PTEN is not present, while the larvae are subjected to the same temperature, time, and quinic acid regimens. See S1 Table for detailed genetics, timing, and conditions of larval rearing. (**B**) Representative confocal picture at T2 of the extent of individual encasing of NSC lineages by CG after the regimen described in (**A**) both for Control and Delayed chamber formation conditions. Top panel shows the whole thoracic VNC, and bottom panel a close-up of the yellow box. NSC lineages were marked with the multicolour lineage tracing Raeppli-NLS (blue, white, orange, and red), induced under the control of the GAL4/UAS system (*wor-GAL4* driver) at ALH0 using *hs-Flp* (*wor > raeppli-NLS*). Larvae are dissected after 100 h at 29°C. See S1 Table for detailed genetics, timing, and conditions of larval rearing. CG membrane is visualised with *Nrv2::GFP* (green). Two representative CG chambers displaying individual encasing of NSC lineages are delineated with a white dashed line. (**C**) Quantification of the percentage of NSC lineages non-individually encased from (**B**). Control T2 (*n* = 8 VNCs) and Delayed chamber formation T2 (*n* = 5 VNCs). Data statistics: generalised linear model (Binomial regression with a Bernoulli distribution). Results are presented as box and whisker plots, where whiskers mark the minimum and maximum, the box includes the 25th–75th percentile, and the line in the box is the median. Individual values are superimposed. (**D**) Schematic of the experiment designed to probe whether specific adhesions exist between individual CG cells and individual NSC lineages. Mosaic clonal analysis is used to impair the growth of a few CG cells in a random fashion (magenta), preventing the encasing of corresponding NSC lineages. If specific adhesions exist between individual CG cells and individual NSC lineages, neighbouring CG cells would not be able to bind to and encase these lineages. (**E**) Representative confocal picture of the CG network in control CG clone and in clone in which CG growth was blocked (PTEN OE). The CoinFLP system was used to generate rare wild-type and PTEN OE clones in the CG. Clones were induced at late embryogenesis/early larval stage through the expression of *cyp4g15-FLP*. Larvae were dissected after 48–72 h at 25°C. See S1 Table for detailed genetics, timing, and conditions of larval rearing. The membrane of the clone is marked by mCD8::RFP (magenta) and CG membrane is visualised with *Nrv2::GFP* (green). (**F**) Quantification of the clone volume (μm3), the number of NSCs per clone, and the number of non-encased NSCs around the clone between wild-type and PTEN OE clones from (**E**). Clone volume and number of NSCs per clone decrease in PTEN OE clones compared to control, indicating that PTEN-overexpressing CG cells were not able to reach their normal size and average NSC encasing. Yet, all NSCs neighbouring the clones were individually encased. Control (*n* = 14 clones) and PTEN OE (*n* = 18 clones). Data statistics: unpaired Student *t* test for clone volumes and Mann–Whitney U test for number of NSCs per clone. For the number of non-encased NSCs around the clone, there is no variance and, thus, statistics cannot be applied (NA, non-applicable). Results are presented as box and whisker plots, where whiskers mark the minimum and maximum, the box includes the 25th–75th percentile, and the line in the box is the median. Individual values are superimposed. The data underlying this figure's quantifications can be found in S1 Data. ALH, after larval hatching; CG, cortex glia; NSC, neural stem cell; OE, overexpression; VNC, ventral nerve cord.

mechanism would rely on the existence of lineage-specific adhesions, with a code of unique molecular interactions between specific CG cells and all cells of specific NSC lineages (Fig 2F, Panel II.1). In this scenario, CG cells would not be interchangeable regarding their adhesive properties. The simplest solution would see all NSC lineages relying on the same adhesion mechanisms between cells (Fig 2F, Panel II.2). The differential adhesion hypothesis proposes that cells with similar adhesive strength cluster together, ultimately sorting cell populations with different adhesions and creating cellular compartments [39]. The existence of intralineage adhesions ($A(L^{NSC})$) stronger than adhesions between a NSC lineage and the CG ($A(L^{NSC}-CG)$) would form a physical barrier for the CG, preventing their intercalation in between cells from the NSC lineage and thus leading to their sorting from NSC lineages, which they encapsulate. In this case, a given CG cell would be able to enwrap different NSC lineages.

To discriminate between these 2 hypotheses, we first assessed the result of preventing some CG cells to encase NSC lineages to a normal extent (Fig 3D). We used Coin-FLP clonal analysis [38] to randomly impair the growth of a few CG cells within the entire population (using *cyp4g15-FLP*; see Methods and S1 Table). A bias in the Coin-FLP system ensures that only rare clones are generated, allowing us to remove adhesions between individual CG and NSC lineages. Overexpression of PTEN in a few CG cells (marked with RFP) before chamber formation resulted in much smaller clones compared to control wild-type clones (Fig 3E and 3F, left graph) and enwrapping less NSCs (Fig 3F, middle graph). However, the CG network itself appeared gapless around PTEN overexpression clones, revealing that CG were able to compensate for the early loss of their neighbours' membrane to restore NSC chambers (Fig 3E and 3F, right graph). To confirm this result, we used the same approach to express the proapoptotic gene *reaper* [40] to kill a few CG, this time only once the NSC chambers were already formed (S3C–S3E Fig). Induction of apoptosis led to a near complete loss of RFP-marked clones, only visible through cell remnants. Nevertheless, the CG network around these clones appeared

intact. This shows that different CG cells can encase a given NSC lineage and that, in accordance with our previous findings (Fig 3A–3C), they are still able to do so after NSC reactivation. Of note, similar results demonstrating the ability of CG to replace each other were previously obtained for primary neurons [41]. Altogether, our results indicate that NSC lineages can be enwrapped by different CG already encasing other lineages and that specific $L^{NSC}$-CG pairings do not occur. This suggests that the same adhesion mechanism might be repeated for each NSC lineage, providing stronger cohesion between cells of the same NSC lineage than between CG and NSC lineages.

## Intralineage adherens junctions are present but not absolutely required for individual encasing of NSC lineages

Previous studies had reported the localization within larval NSC lineages of the *Drosophila* E-cadherin Shotgun (Shg), a component of adherens junctions usually present in epithelia [24,42]. Shg was also shown to localise at the contact between NSC and newborn GMC [43]. We analysed the expression pattern of Shg during larval development using a GFP fusion produced by homologous recombination (*shg::GFP* [44]; Fig 4A). Shg::GFP was detected from larval hatching, initially present around and between NSCs (ALH24, dashed yellow circle), and also along CG membranes (white arrowhead). At later larval stages, following NSC reactivation and neuronal production (ALH48 to ALH72), Shg::GFP showed a remarkable pattern of expression, with a strong enrichment between cells from the same lineage (yellow star marks the NSC) and decreased staining along the CG membrane (yellow arrowheads). A similar pattern was found for its ß-catenin partner Armadillo (Arm; Fig 4B) through antibody staining. As Shg forms homophilic bonds, this suggests that adherens junctions exist between cells of the same NSC lineage.

We then wondered whether such adhesion was exclusively found in differentiating neuronal lineages. We first looked at NSC-like Dpn$^+$ cells from *pros* and *brat* tumours, which are contained clonally within one CG chamber (Fig 2C–2E). A strong Shg staining was detected between NSC-like cells from the same lineage (S4A Fig), but not between NSC and glia. In line with this finding, we observed a strong Shg staining between NSCs of the OL, another type of neural progenitors [45] that are contained within one CG chamber (S4B Fig). Altogether, these results suggest that adherens junctions could be a mean of keeping NSC lineages together by providing differential adhesion.

Previous studies indeed suggested that Shg expression was required in NSC lineages for proper CG structure [24,43]. We first generated NSC lineages mutant for *shg* by inducing MARCM (Mosaic Analysis with a Repressible Cell Marker) clones [46] during late embryogenesis (see Methods and S1 Table). MARCM allows the generation and labelling of single or clonally related cells homozygous for a mutation in an otherwise unlabelled heterozygous animal. Here, the *shg* null allele used (*shg$^{R64}$*) is homozygous lethal and requires clonal analysis. To our surprise, Shg-depleted NSC lineages (labelled by an RFP marker) still stayed individually encased within one CG chamber (S4C Fig, upper panel). The same result was obtained when clones were induced at a later time point to prevent potential compensation through the up-regulation of other adhesion molecules (S4C Fig, lower panel). In accordance with these results, we found that RNAi knockdown of *shg* in NSC lineages (*wor > shg RNAi)* did not disrupt overall lineage organisation within CG chambers (Fig 4C and 4D), despite successfully decreasing Shg::GFP signal (S4D and S4E Fig, mean of 0.3 normalised to control). To note, we also did not record disruption of CG network when *shg* was knocked down in the CG themselves (S4F Fig), in contrast to previous findings [24]. Driving Raeppli-NLS (induced at ALH0) along with *shg* RNAi driven from embryogenesis in the NSC lineage confirmed and quantified

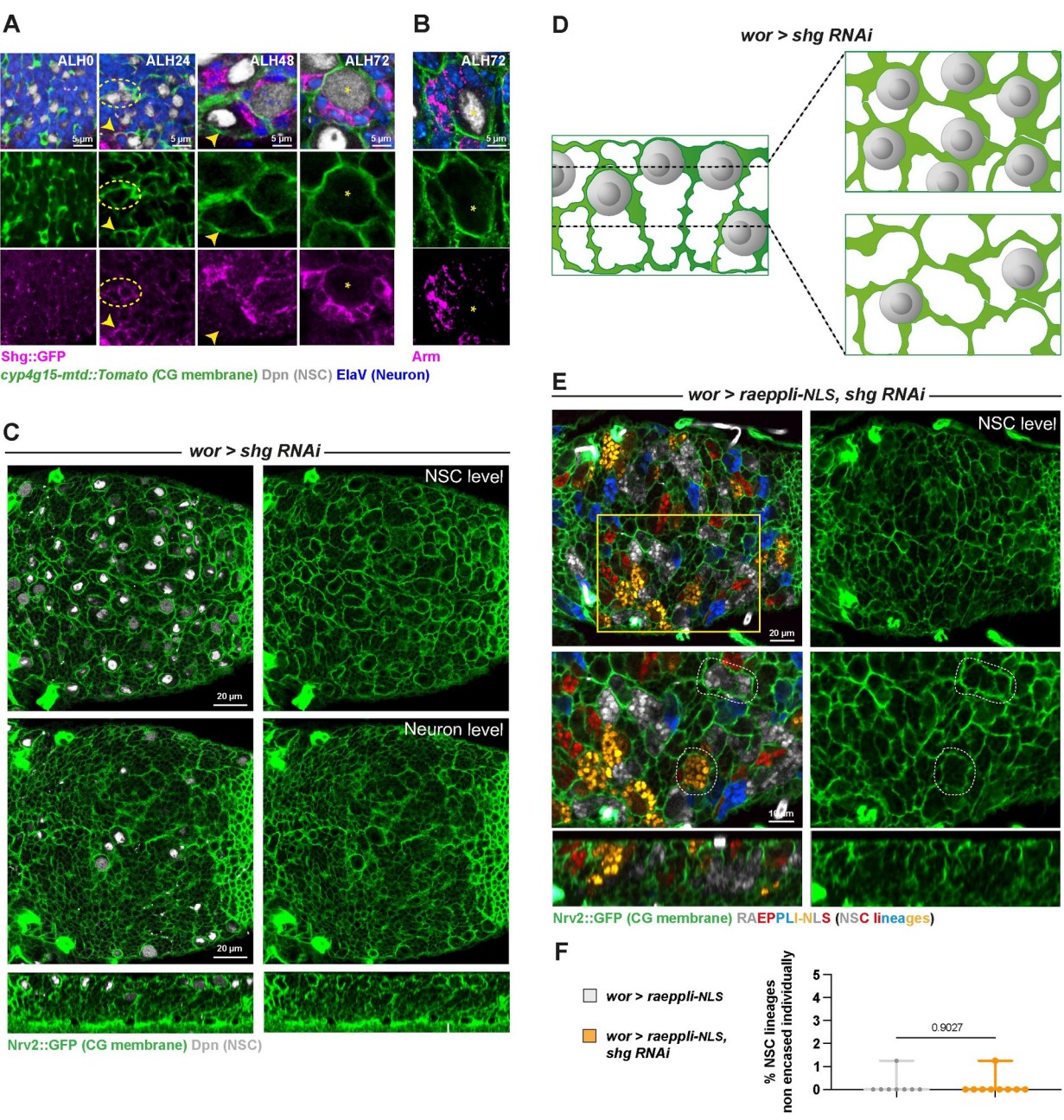

**Fig 4. Intralineage adherens junctions are present but not absolutely required for individual encasing of NSC lineages.** (**A**) Representative confocal images of the expression of the *Drosophila* E-cadherin, Shg, at ALH0, ALH24, ALH48, and ALH72 at 25˚C. $n \geq 7$ VNCs for all time points. Shg is monitored through a *shg::GFP* fusion (magenta), CG membrane is visualised by *cyp4g15-mtd::Tomato* (green), NSCs are labelled with anti-Dpn (grey), and neurons are labelled with anti-ElaV (blue). Yellow stars indicate NSC position, dashed yellow circles indicate Shg::GFP signal between NSCs, and yellow arrowheads indicate Shg::GFP signal between NSC and CG. (**B**) Representative confocal images of the expression of the *Drosophila* β-catenin, Arm, at ALH72 at 25˚C. $n = 6$ VNCs. Arm is detected with a specific antibody (magenta), CG membrane is visualised by Nrv2::GFP (green), NSCs are labelled with anti-Dpn (grey), and neurons are labelled with anti-ElaV (blue). Yellow stars indicate NSC position. (**C**) Representative confocal pictures of the thoracic VNC for control (*wor >—*(x *w$^{1118}$*)) and *shg* knockdown by RNAi (*wor > shg RNAi*, line VDRC 27082) in NSC lineages (driver line *Nrv2::GFP, wor-GAL4; tub-GAL80$^{ts}$*), at the NSC level, at the neuron level, and in orthogonal view. Larvae are dissected after 68 h at 29˚C from ALH0. Control, *wor >—*(x *w$^{1118}$*) ($n \geq 10$ VNCs) and *wor > shg RNAi* ($n \geq 10$ VNCs). CG membrane is visualised by *Nrv2::GFP* (green), and NSCs are labelled with anti-Dpn (grey). (**D**) Schematic of the respective NSC and CG patterns in *shg* knockdown in NSC lineages. (**E**) Representative confocal picture of the thoracic VNC for *shg* knockdown by RNAi (line VDRC 27082; *wor > raeppli-NLS, shg RNAi*) in NSC lineages marked with the multicolour lineage tracing Raeppli-NLS (blue, white, orange, and red). Top panel shows the whole thoracic VNC, and bottom panel a close-up of the yellow box. Raeppli-NLS is induced at ALH0 under the control of *wor-GAL4* using *hs-Flp*. Larvae are dissected after 72 h at 29˚C from ALH0. See S1 Table for detailed genetics, timing, and conditions of larval rearing. CG

membrane is visualised with *Nrv2*::*GFP* (green). Two representative CG chambers displaying individual encasing of NSC lineages are delineated with a white dashed line. (**F**) Quantification of the percentage of NSC lineages non-individually encased from (**E**). Control, *wor > raeppli-NLS* (*n* = 7 VNCs) and *wor > raeppli-NLS*, *shg RNAi* (*n* = 9 VNCs). Data statistics: generalised linear model (Binomial regression with a Bernoulli distribution). Results are presented as box and whisker plots, where whiskers mark the minimum and maximum, the box includes the 25th–75th percentile, and the line in the box is the median. Individual values are superimposed. The data underlying this figure's quantifications can be found in S1 Data. ALH, after larval hatching; Arm, Armadillo; CG, cortex glia; NSC, neural stem cell; Shg, Shotgun; VNC, ventral nerve cord.

the conservation of individual encasing (Fig 4E and 4F) These results argue against the strict requirement of Shg-mediated adhesion for NSC lineage maintenance within one CG chamber [24,43].

We investigated whether intralineage adherens junctions serve as a safety mechanism, ensuring robustness in a system where other strategies would primarily provide intralineage cohesion. As the CG chamber encases NSC at the time they initiate progeny production, timing would first trap these newborn cells together (Fig 2F, Panel I), while adherens junctions would ensure their cohesion if the chamber is affected (such as in Fig 3A–3C). To test this hypothesis, we conditionally blocked CG growth using the Q system, while continuously driving *shg* RNAi and Raeppli-NLS (induced at ALH0) in NSC lineages under the control of the GAL4/UAS system (Fig 5A). We confirmed the effectiveness of *shg* knockdown through antibody staining (S5A Fig). If adherens junctions are important for keeping intralineage cohesion in the absence of proper encasing, progeny born before CG growth is reestablished (i.e., before chamber formation) would not be kept together but mixed with other lineages. Conversely, progeny born after the chamber forms would be encased together. Looking at the deeper level in which differentiating progeny reside (Fig 1F"–1G"), we found that most lineages still appeared correctly encased (Fig 5B and 5C) despite efficient *shg* knockdown and delay in chamber formation (S5A–S5C Fig). We still uncovered a few localised defects in the individual encasing of NSC lineages, with several colours detected within the boundaries of one continuous CG membrane (Fig 5B and 5C). These results indicate that, while adherens junctions are not strictly required for NSC lineage encasing under unchallenged conditions, they may contribute to the robustness of individual NSC lineage encasing when CG are altered, although to a limited extent.

A classical test of the differential adhesion hypothesis is to challenge the balance between the different adhesion complexes. To do so, we overexpressed *shg* in the CG, with the aim to flatten the Shg-based adhesion difference (with now $A(L^{NSC}\text{-}CG) \approx A(L^{NSC})$ for Shg); Fig 2F, Panel II.2). Lineage-expressed endogenous Shg would have the choice to bind either with itself or with CG-provided Shg. Despite the successful expression of *shg* in the CG, the usual pattern of CG chambers was nevertheless maintained in this condition (Fig 5D and 5E), suggesting that in this case, $A(L^{NSC})$ still stays superior to $A(L^{NSC}\text{-}CG)$ and that other adhesions fulfil this role. These data imply that differential adhesion using adherens junction is not the main driver for ensuring the individual encasing of NSC.

## Occluding junction components are expressed in NSC lineages

These findings prompted us to investigate the potential presence and role of other adhesion complexes that could provide intralineage cohesion and differential adhesion to sort NSC lineages from CG.

Occluding junctions (tight junctions in vertebrate and septate junctions in *Drosophila*) [47,48] primarily perform a permeability barrier function to paracellular diffusion. However, they can also provide some adhesion between the cells they link. *Drosophila* septate junctions are formed by the assembly of cell surface adhesion molecules that can interact in *cis* or *trans*, in an homologous or heterologous fashion, and which are linked to the intracellular milieu by

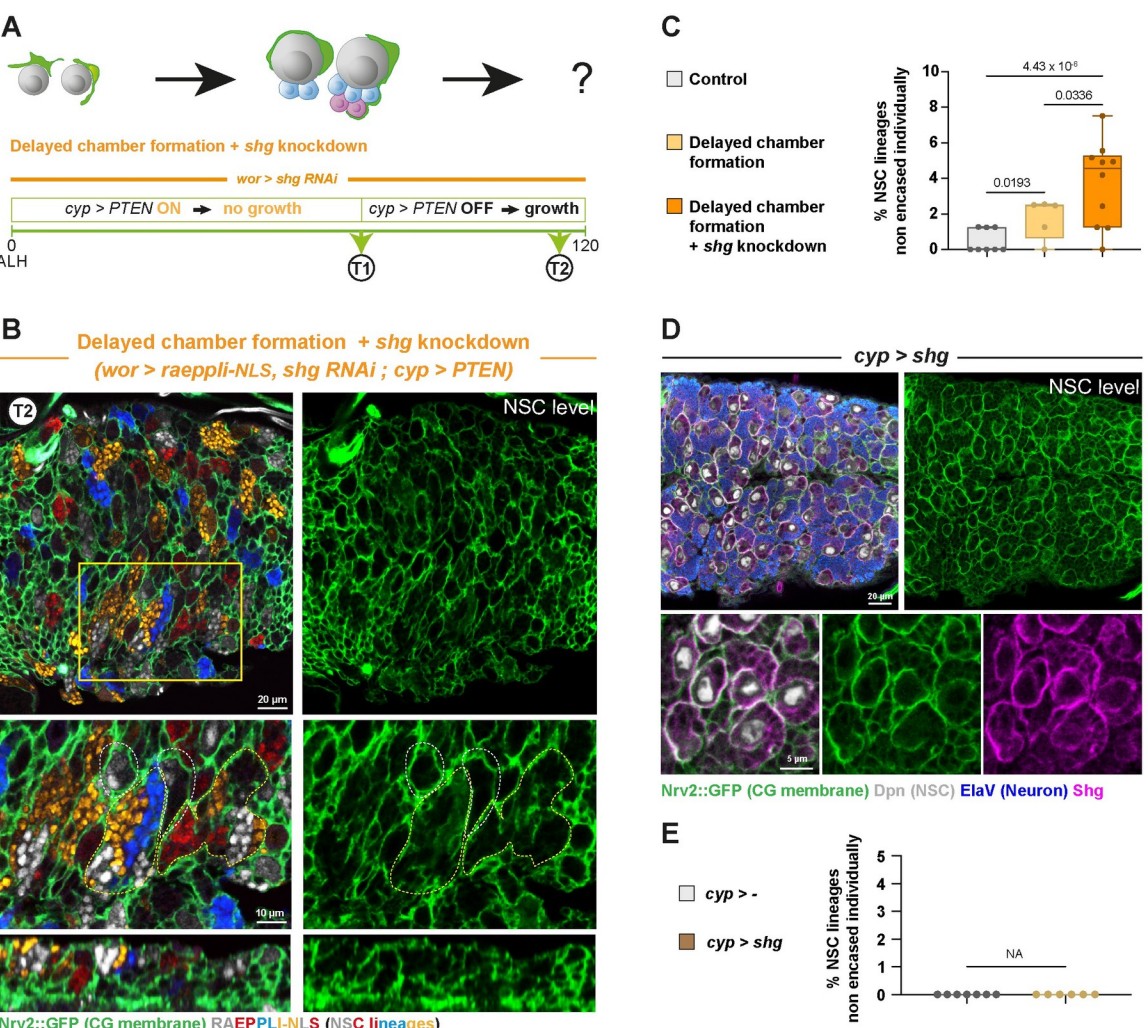

**Fig 5. Intralineage adherens junctions do not drive individual encasing yet might provide robustness.** (**A**) Schematic of the timing and genetic conditions used to probe the importance of Shg intralineage adhesion (*wor-GAL4* > *shg* RNAi, VDRC line 27082) on individual encasing of NSC lineages when CG chamber formation around NSC lineages is delayed. CG growth is initially blocked through PTEN overexpression (*cyp-QF2* driving QUAS-PTEN and combined with *tub-QS* under quinic acid–supplemented food). At T1 (72 h at 29°C from ALH0), CG growth is then allowed by preventing PTEN expression (no quinic acid in the food), and CG structure is assessed at T2 (24–30 h at 29°C from T1, so after 100 h at 29°C from ALH0). In the control condition (see Fig 3A), QUAS-PTEN is not present, while the larvae are subjected to the same temperature, time, and quinic acid regimens. See S1 Table for detailed genetics, timing, and conditions of larval rearing. (**B**) Representative confocal picture at T2 of the extent of individual encasing of NSC lineages by CG after the regimen described in (**A**). Top panel shows the whole thoracic VNC, and bottom panel a close-up of the yellow box. NSC lineages were marked with the multicolour lineage tracing Raeppli-NLS (blue, white, orange, and red), induced under the control of the GAL4/UAS system at ALH0 using *hs-Flp* (*wor* > *raeppli-NLS, shg RNAi*). Larvae are dissected after 100 h at 29°C. See S1 Table for detailed genetics, timing, and conditions of larval rearing. CG membrane was visualised with *Nrv2*::*GFP* (green). Dashed white lines highlight examples of NSC lineages encased individually and dashed yellow lines examples of NSC lineages encased together. (**C**) Quantification of the percentage of NSC lineages non-individually encased from (**B**). Control T2 (*n* = 8 VNCs), Delayed chamber formation T2 (*n* = 5 VNCs) and Delayed chamber + *shg* RNAi in NSC lineages T2 (*n* = 10 VNCs). Data statistics: generalised linear model (Binomial regression with a Bernoulli distribution). $p = 2.46 \times 10^{-9}$ for the grouped dataset. *P* values for individual comparisons test are displayed on the graph. Results are presented as box and whisker plots. (**D**) Representative confocal picture of the thoracic VNC and close-up for *shg* overexpression in the CG (*cyp* > *shg*, driver *Nrv2*::*GFP*, *tub-GAL80^{ts}*; *cyp4g15-GAL4*). Larvae are dissected after 68 h at 29°C from ALH0. CG membrane is visualised by *Nrv2*::*GFP* (green), NSCs are labelled with anti-Dpn (grey), neurons are labelled with anti-ElaV (blue), and Shg is detected with a specific antibody (magenta). (**E**) Quantification of the percentage of NSCs non-individually encased from (**D**). Control, *cyp* >—(x *w^{1118}*) (*n* = 7 VNCs), *cyp* > *shg* (*n* = 6 VNCs). Data statistics: there is no variance and, thus, statistics cannot be applied (NA, non-applicable). Results are presented as box and whisker plots. For all box and whisker plots: whiskers mark the minimum and maximum, the box includes the 25th–75th percentile, and the line in the box is the median. Individual values are superimposed. The data underlying this figure's quantifications can be found in S1 Data. ALH, after larval hatching; CG, cortex glia; NSC, neural stem cell; Shg, Shotgun; VNC, ventral nerve cord.

supporting membrane or cytoplasmic molecules [47]. A core, highly conserved tripartite complex of adhesion molecules comprises Neuroglian (Nrg), Contactin (Cont), and Neurexin-IV (Nrx-IV). Nrg, the *Drosophila* homologue of Neurofascin-155, is an L1-type family transmembrane protein, containing several immunoglobulin domains and is mostly homophilic. Cont, homologous to the human Contactin, also contains immunoglobulin domains, is GPI anchored, and only performs heterophilic interactions. Nrx-IV, homologous to the human Caspr/Paranodin, is a transmembrane protein with a large extracellular domain containing laminin-G domains and EGF repeats [49] and is able to set up heterophilic interactions. Several cytoplasmic or membrane-associated proteins also participate in septate junction formation, such as the FERM-family Coracle (Cora), the MAGUK protein Discs large (Dlg1), and the integral membrane Na K-ATPase pump (ATPα).

We decided to perform a preliminary characterisation of the expression and function of septate junction components in NSC lineages. We found that Nrx-IV, Nrg, Cora, Dlg1, and ATPα are all present in NSC lineages when the chamber is formed (ALH72) and Cora also showed a staining along the CG interface (S6A Fig). We were not able to assess Cont due to lack of access to working reagents. Multiple septate junction components are thus expressed in NSC lineages, localising between cells of the same lineage. We then probed the importance of such expression in the individual encasing of NSC lineages by CG. We knocked down *nrx-IV*, *nrg*, *dlg1*, *cont*, and *ATPα* in NSC lineages from ALH0 using specific RNAi lines (S2 Table). Larvae from *dlg1*, *cont*, and *ATPα* knockdown died at early larval stages. From ATPα knockdown, few larvae still reached late larval stage, displaying restricted irregularities in the CG network. In contrast, *nrx-IV* and *nrg* knockdowns mostly survived and resulted in altered encasing of individual NSC lineages.

We thus decided to focus on Nrx-IV and Nrg functions in NSC lineages. Interestingly, they both perform nervous system–specific roles outside of the septate junction. Nrx-IV is required in the embryonic CNS for axonal wrapping by the midline glia [50–52]. Neuronal Nrg is important for axonal guidance and dendritic arborization of peripheral neurons [53–55], as well as for the function and axon branching of specific larval CB neurons [56,57]. For both proteins, their role in NSC lineages during larval neurogenesis is, however, poorly known.

## Nrx-IV is required in NSC lineages for individual encasing by CG

Using an endogenous GFP fusion (*Nrx-IV::GFP*; Fig 6A), we first analysed Nrx-IV expression along CG chamber formation. Nrx-IV::GFP was detected from early larval stage in embryonic (primary) neurons, and around NSCs (ALH0), an expression maintained while NSCs proceed through reactivation (ALH24, dashed yellow circle). As NSCs have reactivated and CG grown (ALH48), Nrx-IV::GFP appears expressed at the interface between NSC and CG (yellow arrowhead). Further, accompanying the production of newborn, secondary neurons (ALH72), Nrx-IV::GFP is found enriched at the interface of the cells from the same lineage (NSC, GMC, and neurons), while maintaining a strong expression at the interface with CG.

We then performed RNAi knockdown, using a line successfully decreasing the levels of Nrx-IV in NSC lineages (S6B and S6C Fig, mean of 0.4 normalised to control), to assess *nrx-IV* function in NSC lineage encasing. CG pattern first appeared mostly normal when observed at the NSC level, with NSCs seemingly individually separated by CG membranes (compare Fig 6B NSC level with Fig 1F'–1G'). However, we observed a striking, unusual pattern at the level of differentiating progeny, with much larger CG chambers harbouring a clear continuous outline (compare Fig 6B neuron level with Fig 1F"–1G"). A similar result was obtained with another RNAi line against *nrx-IV* (S6D Fig). We then found that expressing *nrx-IV* RNAi under the control of the pan-neuronal driver *ElaV-GAL4* (S6E Fig), which drives both in

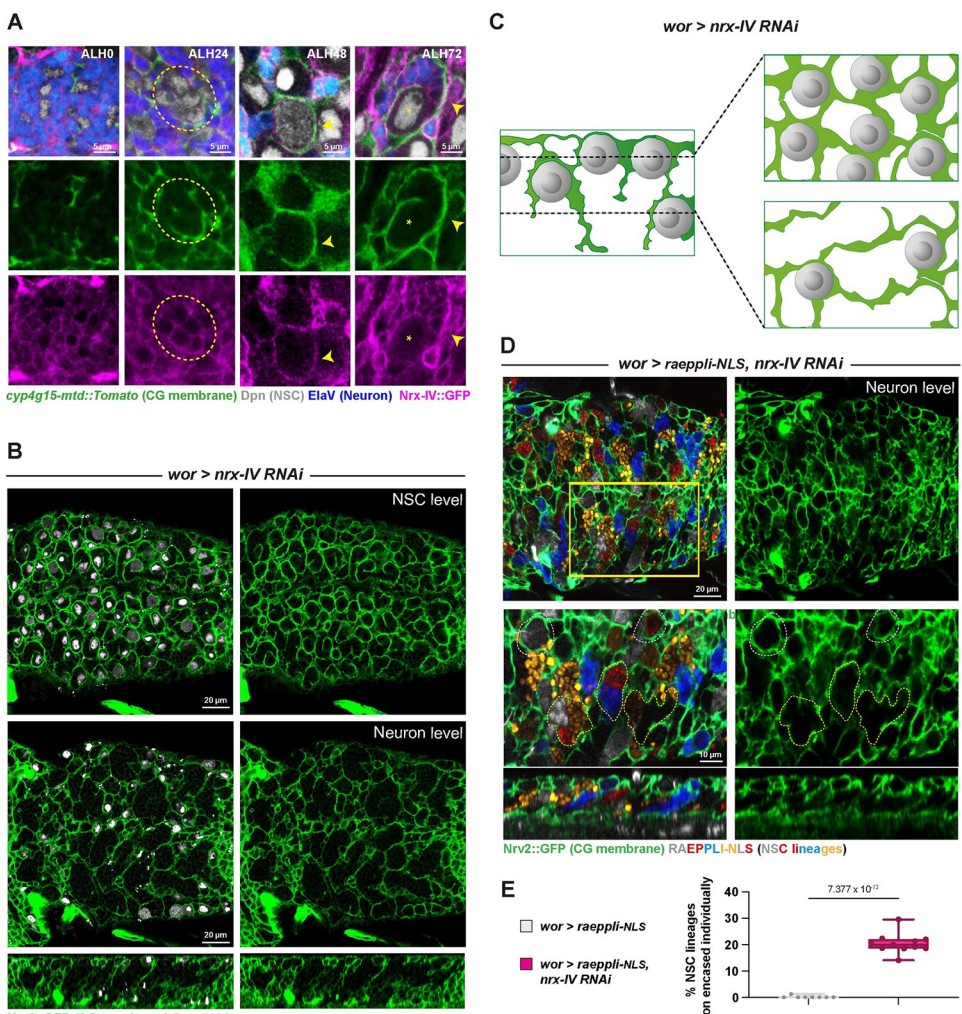

**Fig 6. Nrx-IV is required in NSC lineages for their individual encasing by CG.** (**A**) Representative confocal images of the expression of Nrx-IV at ALH0, ALH24, ALH48, and ALH72 at 25°C. $n \geq 8$ VNCs for all time points. Nrx-IV is monitored through a *Nrx-IV::GFP* fusion (magenta), CG membrane is visualised by *cyp4g15-mtd::Tomato* (green), NSCs are labelled with anti-Dpn (grey), and neurons are labelled with anti-ElaV (blue). Yellow stars indicate NSC position, dashed yellow circles indicate Nrx-IV::GFP signal between NSCs, and yellow arrowheads indicate Nrx-IV::GFP signal between NSC and CG. (**B**) Representative confocal picture of the thoracic VNC for a condition in which *nrx-IV* is knocked down by RNAi (*wor > nrx-IV RNAi*, BDSC line 32424) from ALH0 in NSC lineages (driver line *Nrv2::GFP, wor-GAL4; tub-GAL80^{ts}*) at the NSC level, at the Neuron level, and in orthogonal view. Larvae are dissected after 68 h at 29°C from ALH0. Control, *wor >—*(x *w^{1118}*) ($n \geq 10$ VNCs) and *wor > nrx-IV RNAi* ($n \geq 10$ VNCs). CG membrane is visualised by *Nrv2::GFP* (green), and NSCs are labelled with anti-Dpn (grey). (**C**) Schematic of the respective NSC and CG patterns in *nrx-IV* knockdown in NSC lineages. (**D**) Representative confocal picture of the thoracic VNC for a condition in which *nrx-IV* is knocked down by RNAi (BDSC line 32424) from ALH0 in NSC lineages and marked with the multicolour lineage tracing Raeppli-NLS (*wor > raeppli-NLS, nrx-IV RNAi*). Raeppli-NLS (blue, white, orange, and red) is induced at ALH0 using *hs-Flp*. Larvae are dissected after 72 h at 29°C from ALH0. See S1 Table for detailed genetics, timing, and conditions of larval rearing. CG membrane was visualised with *Nrv2::GFP* (green). Dashed white lines highlight examples of NSC lineages encased individually, and dashed yellow lines examples of NSC lineages encased together. (**E**) Quantification of the percentage of NSC lineages non-individually encased from (**D**). Control, *wor > raeppli-NLS* ($n = 7$ VNCs) and *wor > raeppli-NLS, nrx-IV RNAi* ($n = 11$ VNCs). Data statistics: generalised linear model (Binomial regression with a Bernoulli distribution). Results are presented as box and whisker plots, where whiskers mark the minimum and maximum, the box includes the 25th–75th percentile, and the line in the box is the median. Individual values are superimposed. The data underlying this figure's quantifications can be found in S1 Data. ALH, after larval hatching; CG, cortex glia; Nrx-IV, Neurexin-IV; NSC, neural stem cell; VNC, ventral nerve cord.

immature (secondary here) and mature (primary here) neurons, also resulted in bigger yet well-defined CG chambers. In contrast, no effect was detected under *nrx-IV* knockdown in mature neurons (with formed or forming synapses; *nSyb-GAL4* driver [58]) or in the CG (S6E Fig).

To quantify individual encasing, we further expressed Raeppli-NLS along with *nrx-IV* RNAi in NSC lineages (both induced at ALH0). It replicated the extensive loss of individual encasing seen in Fig 6B, with multiple NSC lineages not separated by CG membranes but rather clustered together in large, defined chambers (Fig 6D and 6E; 22% of NSC lineages sharing a CG chamber with at least another one).

Altogether, our findings show that the expression of Nrx-IV in whole NSC lineages, including newborn neurons, but not in CG or mature neurons, is required for their individual encasing by CG.

## A glia to NSC lineages adhesion through Nrx-IV and Wrapper is required for individual encasing

The dual role of Nrx-IV within and outside septate junctions is sustained by the existence of alternative splicing [51]. Nrx-IV can be produced as a septate junction isoform (Nrx-IV$^{exon3}$) and a neuronal isoform outside of SJ (Nrx-IV$^{exon4}$). Nrx-IV role within the embryonic CNS is through its recruitment by and binding to its glial partner Wrapper, another member of the immunoglobulin family [50–52]. We thus sought to assess whether the role of Nrx-IV in NSC lineages encasing by CG was dependent on Wrapper.

Previous studies had reported that regulatory sequences in the *wrapper* gene drive in the CG during late larval stages [41,59]. In situ hybridization against *wrapper* mRNA (HCR RNA-FISH; see Methods) detected *wrapper* expression in CG throughout larval development and chamber formation (Fig 7A). We then asked whether knocking down *wrapper* in the CG would recapitulate the encasing phenotype found under *nrx-IV* loss of function in NSC lineages. We first checked the efficiency of RNAi knockdown in the CG using RNA FISH and found that it was specifically wiping out *wrapper* signal in the CG (S7A and S7B Fig), while preserving its known expression in the midline glia (see arrowheads in S7A Fig). Driving this RNAi line in the CG reproduced the highly characteristic pattern of large yet defined CG chambers (Fig 7B, observed at the neuron level) found during *nrx-IV* knockdown in NSC lineages (compare with Fig 6B). A similar result was obtained with a second RNAi line against *wrapper* (S7C Fig). We further quantified the loss of individual encasing by marking NSC lineages with Raeppli-NLS while driving *wrapper* RNAi in the CG and found that 25% of the NSC lineages was sharing a CG chamber with at least another one (Fig 7C and 7D).

These results suggest that Nrx-IV in NSC lineages interact with Wrapper in the CG for ensuring NSC individual encasing. To strengthen this relationship, we performed genetic interactions between *nrx-IV* and *wrapper*, comparing CG phenotype between *nrx-IV* only knowdown, *wrapper* only knockdown, and double *nrx-IV* and *wrapper* knockdown in both NSC lineages and CG (see Methods, S1 Table, and S7D Fig for expression of the combined driver lines; single knockdowns were dose compensated). At a qualitative level, we first observed that all 3 combinations displayed the characteristically large and defined CG chambers associated with *nrx-IV* loss of function in NSC lineages and *wrapper* loss of function in CG (Fig 7E). As we could not add a clonal analysis tool to these already complex genotypes, we decided to quantify the volume of CG membrane as a proxy for the density of encasing (i.e., grouped NSC lineages means larger chambers and, conversely, less CG membrane per NSC lineage). We first confirmed the validity of our proxy by assessing whether it was able to detect *nrx-IV* phenotype. To do so, we measured CG volume between a control condition, *shg* RNAi

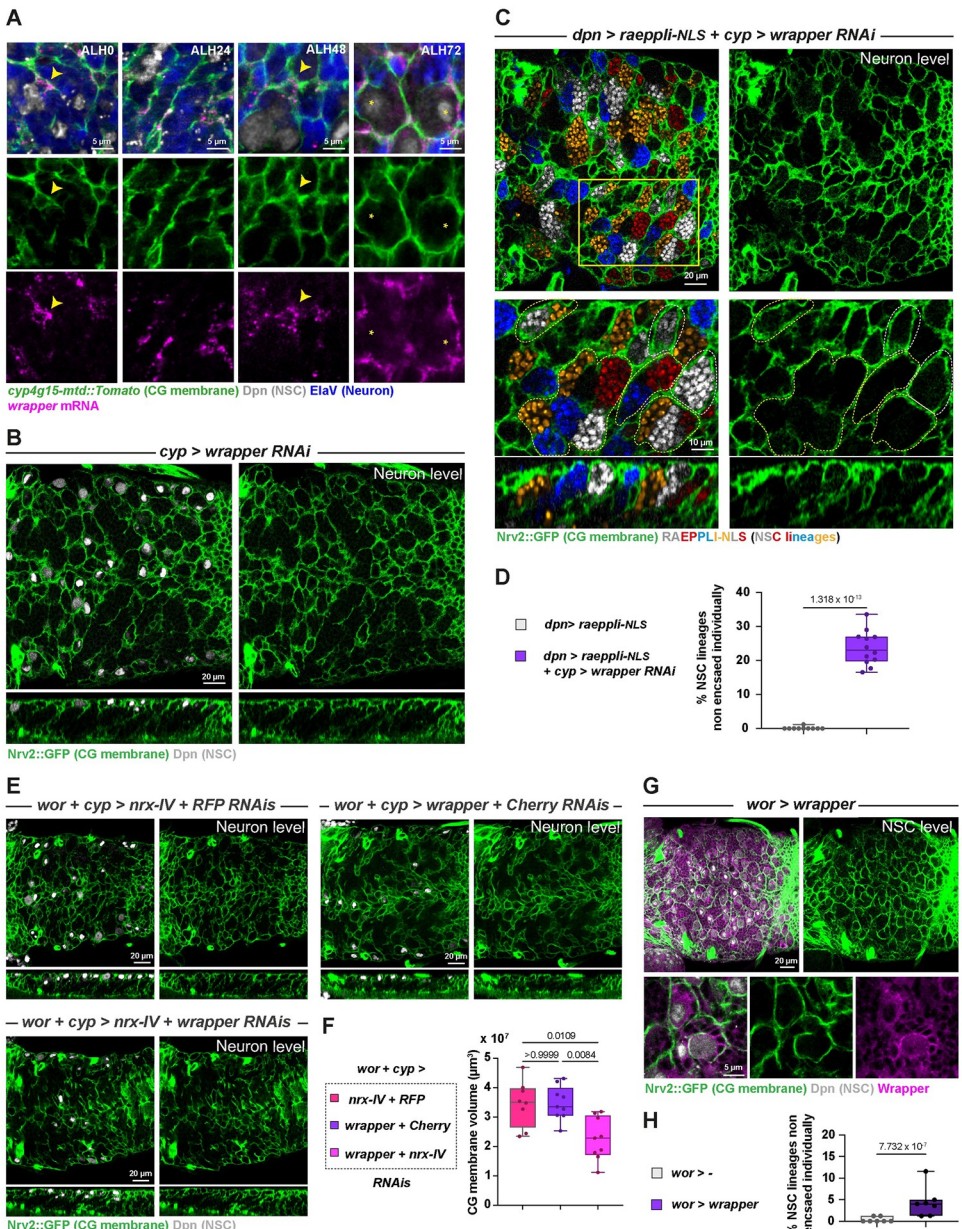

**Fig 7. A CG to lineage interaction through Nrx-IV and Wrapper is required for individual encasing of NSC lineages by the CG.** (**A**) Representative confocal pictures of the localisation of *wrapper* mRNA at ALH0, ALH24, ALH48, and ALH72 at 25°C. *n* ≥ 6 VNCs for all time points. *wrapper* mRNA (magenta) is detected through RNA FISH. CG membrane is visualised by *cyp4g15-mtd::Tomato* (green), NSCs are labelled with anti-Dpn (grey), and neurons are labelled with anti-ElaV (blue). Yellow arrowheads indicate examples of colocalisation between CG membrane and wrapper mRNA signal. (**B**) Representative confocal pictures of the thoracic VNC for a condition in which *wrapper* is knocked down by RNAi (*cyp > wrapper RNAi*, BDSC line 29561) in the CG (driver line *Nrv2::GFP*, *tub-GAL80ts; cyp4g15-GAL4*) at the NSC level, at the neuron level, and in orthogonal view. Larvae are dissected after 68 h at 29°C from ALH0. Control, *cyp >*—(x *w1118*) (*n* ≥ 10 VNCs) and *cyp > wrapper RNAi* (*n* ≥ 10 VNCs). CG membrane is visualised by *Nrv2::GFP* (green), and NSCs are labelled with anti-Dpn (grey). (**C**) Representative confocal picture of the thoracic VNC for a condition in which *wrapper* is knocked down by RNAi (BDSC line 29561) from ALH0 in CG, while NSC lineages are marked with the multicolour lineage tracing Raeppli-NLS (blue, white, orange, and red). Raeppli-NLS is induced at ALH0 using *hs-Flp* and under the control of the LexA/LexAop system (*dpn* enhancer), while *wrapper* RNAi is under the control of the GAL4/UAS system (*dpn > raeppli-NLS + cyp > wrapper RNAi*). Larvae are dissected after 72 h at 29°C from ALH0. See S1 Table for detailed genetics, timing, and conditions of larval rearing. CG membrane was visualised with *Nrv2::GFP* (green). Dashed white lines highlight examples of NSC lineages encased individually, and dashed yellow lines examples of NSC lineages encased together. (**D**) Quantification

of the percentage of NSC lineages non-individually encased from (**C**). Control, *dpn > raeppli-NLS* (*n* = 10 VNCs) and *dpn > raeppli-NLS + cyp > wrapper RNAi* (*n* = 12 VNCs). Data statistics: generalised linear model (Binomial regression with a Bernoulli distribution). Results are presented as box and whisker plots. (**E**) Representative confocal pictures of thoracic VNCs for which combined NSC lineage and CG drivers (*wor + cyp >*; driver line *Nrv2::GFP, wor-GAL4/CyO; cyp4g15-GAL4, tub-GAL80^ts^*) were used to perform double RNAi knockdown against *nrx-IV* and *RFP*; *wrapper* and *mCherry*; and *nrx-IV* and *wrapper. nrx-IV RNAi*, BDSC line 32424; *wrapper RNAi*, VDRC line 105314. Larvae are dissected after 68 h at 29˚C. CG membrane is visualised by *Nrv2::GFP* (green), and NSCs are labelled with anti-Dpn (grey). (**F**) Quantification of the CG membrane volume per NSC from (**E**). See Methods for details. *nrx-IV + RFP* RNAis (*n* = 8 VNCs), *wrapper + mCherry* RNAis (*n* = 9 VNCs), *nrx-IV + wrapper* RNAis (*n* = 9 VNCs). Data statistics: Kruskal–Wallis H test with Dunn's multiple comparisons test. *p* = 0.0029 for the Kruskal–Wallis H test on grouped dataset. *P* values from Dunn's multiple comparisons test are displayed on the graph. Results are presented as box and whisker plots. (**G**) Representative confocal picture of a thoracic VNC for a condition in which *wrapper* is overexpressed in NSC lineages from ALH0 (*wor > wrapper*, driver line *Nrv2::GFP, wor-GAL4; tub-GAL80^ts^*). Larvae are dissected after 68 h at 29˚C. CG membrane is visualised by *Nrv2::GFP* (green), and NSCs are labelled with anti-Dpn (grey). (**H**) Quantification of the percentage of NSCs non-individually encased from (**G**). Control, *wor >—*(x *w^1118^*) (*n* = 7 VNCs), *wor > wrapper* (*n* = 7 VNCs). Data statistics: generalised linear model (Binomial regression with a Bernoulli distribution). Results are presented as box and whisker plots. For all box and whisker plots: whiskers mark the minimum and maximum, the box includes the 25^th^–75^th^ percentile, and the line in the box is the median. Individual values are superimposed. The data underlying this figure's quantifications can be found in S1 Data. ALH, after larval hatching; CG, cortex glia; FISH, fluorescent in situ hybridization; Nrx-IV, Neurexin-IV; NSC, neural stem cell; VNC, ventral nerve cord.

in NSC lineages, and *nrx-IV* RNAi in NSC lineages and found that *nrx-IV* knockdown resulted in a significantly lower CG volume compared to control, while *shg* knockdown did not show a decrease (S7E Fig). We then used the same approach to assess NSC encasing upon double knockdown of *nrx-IV* and *wrapper* and uncovered that it was significantly lower than upon the individual knockdown of either, which displayed a similar decrease (Fig 7F). These data pinpoint a greater effect of the combined *nrx-IV* and *wrapper* knockdowns than their individual contribution on NSC lineage encasing by the CG, showing that the genotype of one affects the phenotype of the other. This demonstrates an epistatic interaction between *nrx-IV* and *wrapper* in this context.

We then wondered how the Nrx-IV to Wrapper interaction would fit in the differential adhesion hypothesis. If CG to NSC lineage adhesion is indeed weaker than intralineage adhesion (Fig 2F, panel II.2; $A(L^{NSC}\text{-}CG) < A(L^{NSC})$), overexpression of Wrapper in NSC lineages should not affect the sorting between NSC lineages and CG, since $A(L^{NSC})$ would still be superior to $A(L^{NSC}\text{-}CG)$. However, if Nrx-IV to Wrapper interaction is stronger than the sum of intralineage adhesions, then forcing its establishment within the lineage would favour the random grouping of NSC lineages together. We found that misexpressing *wrapper* in NSC lineages from larval hatching (ALH0), while successful (as confirmed by staining with an antibody that can also detect endogenous Wrapper; see S7F Fig), resulted in very little alteration of CG encasing of individual NSC lineages (Fig 7G and 7H). These data plead in favour of a CG to NSC lineage adhesion through Nrx-IV and Wrapper being weaker than the sum of intralineage adhesions.

Taken together, our results suggest that Nrx-IV in NSC lineages partners with Wrapper in the CG, outside of a septate junction function. This interaction is essential to produce individual encasing of NSC lineages by CG, and in its absence, NSC lineages are randomly grouped in well-defined, larger chambers.

## Nrg is required in NSC lineages for integrity of the CG network and individual encasing by CG

We then wondered what could provide strong intralineage adhesion and turned our eyes to Nrg, known to perform homophilic interaction and identified as a candidate of interest in a preliminary screen (S2 Table).

We first assessed Nrg expression in NSC lineages. A protein trap for Nrg (*Nrg::GFP*, Fig 8A) revealed a strong enrichment between cells of the same NSC lineages following progeny production (ALH72). Moreover, Nrg::GFP was detected at the interface between lineages and CG (yellow arrowhead). However, contrary to what we observed with Nrx-IV::GFP, Nrg:: GFP did not appear enriched between NSCs before their encasing by CG (ALH0-24, dashed yellow circle).

We then assessed the impact of *nrg* knockdown in NSC lineages, using an RNAi line able to significantly decrease Nrg::GFP signal (S8A and S8B Fig, mean of 0.5 normalised to control). Driving *nrg* knockdown from larval hatching (ALH0) led to few larvae of the right genotype, in which CG displayed some restricted defects in individual encasing of NSC lineages (S2 Table). We thought these animals might have survived due to a weak phenotype and decided to delay the RNAi knockdown, started after 1 day at 18°C (to allow its repression by the GAL80$^{ts}$), and then induced for 2 to 3 days (at 29°C). In this case, we obtained more surviving larvae, which displayed strong alterations of NSC individual encasing by CG. NSCs were indeed clumped together, seemingly touching each other (Fig 8B NSC level, compare with Fig 1F'–1G'). Analysis at the level of the neuronal progeny also revealed larger zones devoided of CG membrane (Fig 8B neuron level, compare with Fig 1F"–1G"). In addition, we noticed that, in contrast to *nrx-IV* knockdown in NSC lineages, CG membranes rather seemed irregular, exhibited gaps and broken/stunted ends, under *nrg* knockdown (see white arrows for examples in Fig 8B). Driving *nrg* RNAi in all neurons (*ElaV* >) also led to defects in the encasing of NSC lineages (S8C Fig). In contrast, *nrg* knockdown in mature neurons only (*nSyb* >) or in CG (*cyp* >) did not lead to observable CG alteration (S8C Fig). We further quantified *nrg* phenotype by expressing Raeppli-NLS (induced at ALH0) along with *nrg* RNAi in NSC lineages (*wor* >). It confirmed the loss of individual encasing, with multiple NSC lineages clumped together and not separated anymore by CG membrane, in a seemingly random fashion (Fig 8D and 8E; mean of 40% of NSC lineages grouped with at least another one). The expressivity of the phenotype was variable, with a class showing nearly no individual encasing (around 100% of NSC lineages grouped with at least another one; Fig 8E) and a class with a range of milder alterations (less than 50% of NSC lineages grouped with at least another one; Fig 8E), something we also noticed qualitatively in the conditions of Fig 8B. In addition, we noticed rare occurrences of CG chambers containing only neurons (S8D Fig).

Altogether, our findings suggest that the expression of Nrg in NSC lineages is required for correct CG network architecture and the individual encasing of NSC lineages.

## Intralineage adhesion through Nrg drives cell sorting and individual encasing by CG

Like Nrx-IV, the dual role of Nrg in and outside of septate junction comes from differential splicing [60]. Nrg comes in 2 isoforms, with the same extracellular domain but different intracellular parts (Fig 9A). While the short isoform, Nrg$^{167}$, localises in the septate junction of epithelial tissues, the long isoform, Nrg$^{180}$, is expressed in neurons of the developing central and peripheral nervous systems [55,56,60].

We first determined which isoform is expressed in NSC lineages during the larval stage, taking advantage of isoform-specific tools. Staining of *Nrg::GFP* CNS (ALH72) with an antibody (BP104) specifically recognising the Nrg$^{180}$ isoform [60] revealed that Nrg$^{180}$ localises in the membranes of all cells from NSC lineages, but not in known septate junctions within the tissue (yellow arrowheads) (Figs 9B and S9A). In neurons, Nrg$^{180}$ was not only found in the cell body but appeared also enriched in their axonal bundle, a localisation reported previously [61]. We then took advantage of an Nrg::GFP fusion shown in other tissues to preferentially target the

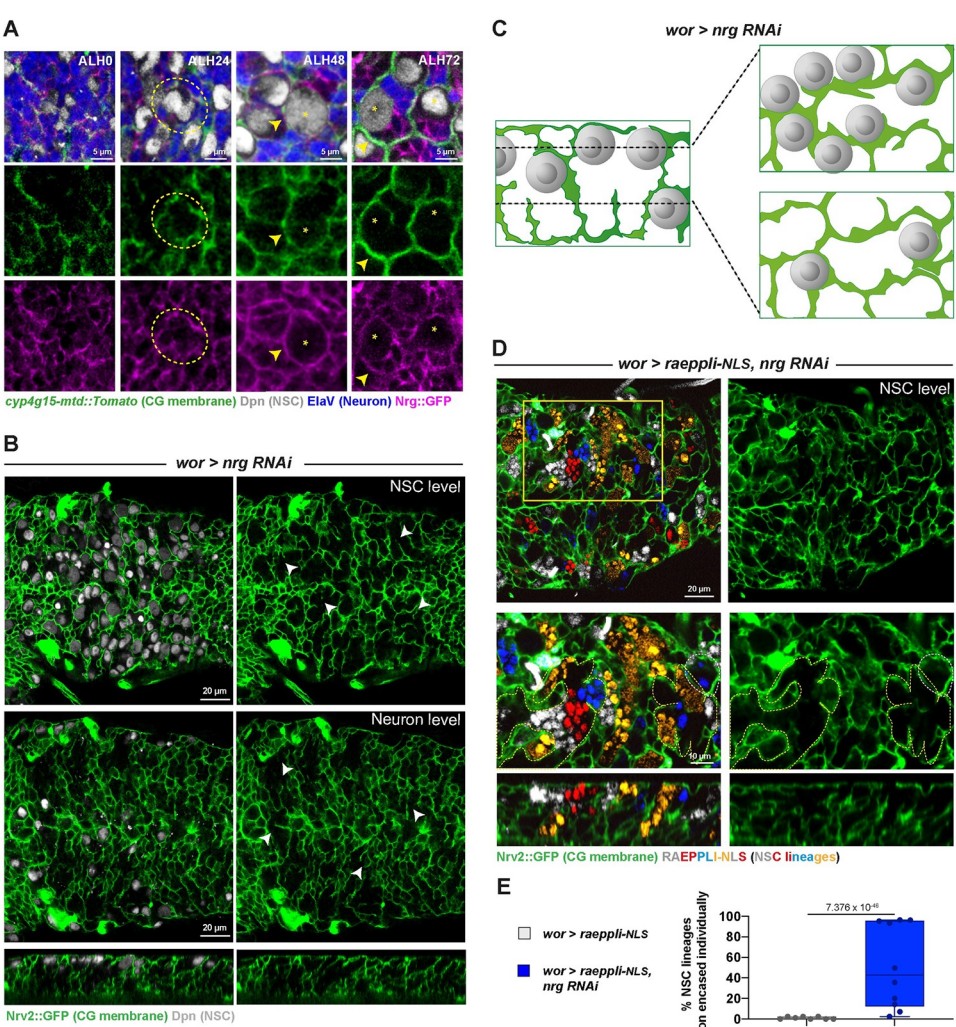

**Fig 8. Nrg is required in NSC lineages for their individual encasing and CG network integrity.** (**A**) Representative confocal images of the expression of Nrg at ALH0, ALH24, ALH48, and ALH72 at 25°C. $n \geq 7$ VNCs for all time points. Nrg is monitored through a *Nrg::GFP* fusion (magenta), CG membrane is visualised by *cyp4g15-mtd::Tomato* (green), NSCs are labelled with anti-Dpn (grey), and neurons are labelled with anti-ElaV (blue). Yellow stars indicate NSC position, dashed yellow circles indicate the lack of Nrg::GFP signal between NSCs, and yellow arrowheads indicate Nrg::GFP signal between NSC and CG. (**B**) Representative confocal pictures of the thoracic VNC for a condition in which *nrg* is knocked down by RNAi (*wor > nrg RNAi*, BDSC line 37496) in NSC lineages (driver line *Nrv2::GFP, wor-GAL4; tub-GAL80ᵗˢ*) at the NSC level, at the neuron level, and in orthogonal view. Larvae are dissected after 24 h at 18°C followed by 54 h at 29°C. Control, *wor >—*(x $w^{1118}$) ($n \geq 10$ VNCs) and *wor > nrg RNAi* ($n \geq 10$ VNCs). CG membrane is visualised by *Nrv2::GFP* (green) and NSCs are labelled with anti-Dpn (grey). White arrows illustrate gaps in CG membranes. This phenotype is seen in 7/10 cases, 3/10 show a milder phenotype. (**C**) Schematic of the respective NSC and CG patterns in *nrg* knockdown in NSC lineages. (**D**) Representative confocal pictures of the thoracic VNC for a condition in which *nrg* is knocked down by RNAi (BDSC line 37496) in NSC lineages marked with the multicolour lineage tracing Raeppli-NLS (*wor > raeppli-NLS, nrg RNAi*). Raeppli-NLS (blue, white, orange, and red) is induced at ALH0 using *hs-Flp*, and RNAi after 24 h at 18°C. Larvae are dissected 60 h after RNAi induction. See S1 Table for detailed genetics, timing, and conditions of larval rearing. CG membrane was visualised with *Nrv2::GFP* (green). Dashed white lines highlight examples of NSC lineages encased individually, and dashed yellow lines examples of NSC lineages encased together. (**E**) Quantification of the percentage of NSC lineages non-individually encased from (**D**). Control, *wor > raeppli-NLS* ($n = 8$ VNCs) and *wor > raeppli-NLS, nrg RNAi* ($n = 10$ VNCs). Data statistics: generalised linear model (Binomial regression with a Bernoulli distribution). Results are presented as box and whisker plots, where whiskers mark the minimum and maximum, the box includes the 25th–75th percentile, and the line in the box is the median. Individual values are superimposed. The data underlying this figure's quantifications can be found in S1 Data. ALH, after larval hatching; CG, cortex glia; Nrg, Neuroglian; NSC, neural stem cell; VNC, ventral nerve cord.

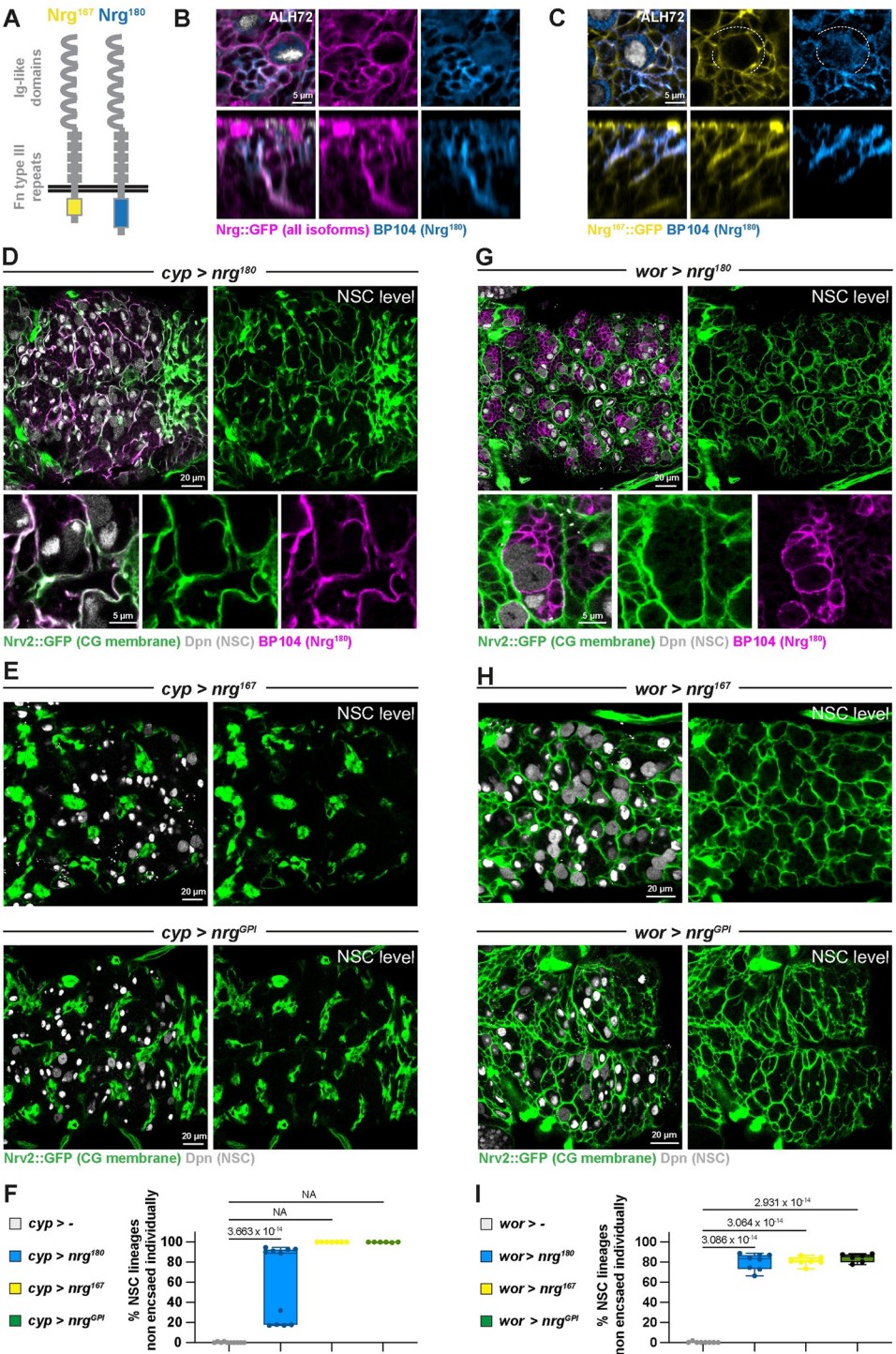

**Fig 9. Individual encasing of NSC lineages relies on strong intralineage adhesion through Nrg. (A)** Schematic depicting the 2 isoforms for Nrg, Nrg[167], and Nrg[180]. Only the intracellular C-terminal part differs and will modulate downstream signalling. (**B**) Representative confocal close-up picture of the localisation of the Nrg[180] isoform in NSC lineages, in a thoracic VNC at ALH72 at 25˚C. $n = 8$ VNCs. All Nrg isoforms are monitored through a *Nrg::GFP* protein trap (magenta) and the Nrg[180] isoform is detected with a specific antibody (BP104, light blue). (**C**) Representative confocal close-up picture of the respective localisations of the Nrg[167] and Nrg[180] isoforms in NSC lineages, in a thoracic VNC at ALH72 at 25˚C. $n = 6$ VNCs. The Nrg[167] isoform is visualised by a protein trap in the *nrg* gene leading to the preferential expression of this isoform (*Nrg[167]::GFP*, yellow). The Nrg[180] isoform is detected with a specific antibody (BP104, light blue). The dashed white line highlights the perimeter of the NSC devoid of BP104

signal. (**D**) Representative confocal picture of a thoracic VNC and close-up for *nrg$^{180}$* overexpression in the CG from ALH0 (*cyp > nrg$^{180}$*, driver *Nrv2::GFP, tub-GAL80$^{ts}$; cyp4g15-GAL4*). Larvae are dissected after 68 h at 29˚C. CG membrane is visualised by *Nrv2::GFP* (green), NSCs are labelled with an anti-Dpn (grey), and Nrg$^{180}$ is detected with a specific antibody (BP104, magenta). (**E**) Representative confocal pictures of thoracic VNCs for *nrg$^{167}$* and *nrg$^{GPI}$* overexpression in the CG from ALH0 (*cyp > nrg$^{167}$* and *cyp > nrg$^{GPI}$*, respectively, driver *Nrv2::GFP, tub-GAL80$^{ts}$; cyp4g15-GAL4*). Larvae are dissected after 68 h at 29˚C. CG membrane is visualised by *Nrv2::GFP* (green), and NSCs are labelled with an anti-Dpn (grey). (**F**) Quantification of the percentage of NSCs non-individually encased from (**D, E**). Control, *cyp >—*(x *w$^{1118}$*) (*n* = 10 VNCs), *cyp > nrg$^{180}$* (*n* = 11 VNCs), *cyp > nrg$^{167}$* (*n* = 7 VNCs), and *cyp > nrg$^{GPI}$* (*n* = 6 VNCs). Data statistics: generalised linear model (Binomial regression with a Bernoulli distribution) for nrg$^{180}$. For nrg$^{167}$ and nrg$^{GPI}$, there is no variance and, thus, statistics cannot be applied (NA, non-applicable). Results are presented as box and whisker plots. (**G**) Representative confocal picture of a thoracic VNC and close-up for Nrg$^{180}$ overexpression in the NSC lineages from ALH0 (*wor > nrg$^{180}$*, driver *Nrv2::GFP, worniu-GAL4; tub-GAL80$^{ts}$*). Larvae are dissected after 68 h at 29˚C. CG membrane is visualised by *Nrv2::GFP* (green), NSCs are labelled with an anti-Dpn (grey), and Nrg$^{180}$ is detected with a specific antibody (BP104, magenta). (**H**) Representative confocal pictures of thoracic VNCs for *nrg$^{167}$* and *nrg$^{GPI}$* overexpressed from ALH0 in NSC lineages (*wor > nrg$^{167}$* and *wor > nrg$^{GPI}$*, respectively, driver line *Nrv2::GFP, wor-GAL4; tub-GAL80$^{ts}$*). Larvae are dissected after 68 h at 29˚C. CG membrane is visualised by *Nrv2::GFP* (green), and NSCs are labelled with anti-Dpn (grey). (**I**) Quantification of the percentage of NSCs non-individually encased from (**G, H**). Control, *wor >—*(x *w$^{1118}$*) (*n* = 8 VNCs), *wor > nrg$^{180}$* (*n* = 8 VNCs), *wor > nrg$^{167}$* (*n* = 7 VNCs), and *wor > nrg$^{GPI}$* (*n* = 6 VNCs). Data statistics: generalised linear model (Binomial regression with a Bernoulli distribution). *p* = $1.93 \times 10^{-113}$ for the grouped dataset. *P* values for individual comparisons test are displayed on the graph. Results are presented as box and whisker plots. For all box and whisker plots: whiskers mark the minimum and maximum, the box includes the 25th–75th percentile, and the line in the box is the median. Individual values are superimposed. The data underlying this figure's quantifications can be found in S1 Data. ALH, after larval hatching; CG, cortex glia; Nrg, Neuroglian; NSC, neural stem cell; VNC, ventral nerve cord.

Nrg$^{167}$ isoform (called *Nrg$^{167}$::GFP*; [53,55]). *Nrg$^{167}$::GFP* also appeared enriched between cells of the same NSC lineage, where it colocalised with BP104 staining, except on the NSC perimeter, devoided of BP104 (Figs 9C and S9B; see dashed white line for lack of BP104). In contrast, only Nrg$^{167}$ is detected in septate junctions. We then wondered whether *nrg* knockdown was able to lower the levels of both isoforms. *nrg* knockdown in NSC lineages completely depleted the BP104 signal (S9C and S9D Fig; mean of 0.1 normalised to control). *Nrg$^{167}$::GFP* levels also were strongly decreased upon nrg knockdown (S9E and S9F Fig; mean of 0.25 normalised to control). While it appears in a lower fashion than Nrg$^{180}$, it might be due to a higher stability of the *Nrg$^{167}$::GFP* fusion, while endogenous Nrg$^{180}$ was detected with an antibody. Taken together, these data suggest that the 2 isoforms of Nrg are expressed, and can be efficiently knocked down, in NSC lineages.

Mostly homophilic interactions (between the same or different isoforms) have been reported for Nrg. We thus wondered whether an Nrg to Nrg interaction within the NSC lineages could fulfil the role of an intralineage adhesion stronger than a CG to NSC adhesion.

Since *nrg* knockdown in CG (S8C Fig) did not recapitulate *nrg* knockdown in NSC lineages, the homophilic Nrg interactions between CG and NSC lineages, if existing, are not involved in individual encasing. We then assessed the relevance of intralineage Nrg interactions in the differential adhesion hypothesis (Fig 2F, panel II.2). If such adhesion is stronger than the CG to NSC lineage interaction, then expressing Nrg in CG would force CG to interact with each other. Strikingly, misexpressing Nrg$^{180}$ in CG from larval hatching (ALH0) resulted in altered CG morphology and loss of individual encasing of NSC lineages (Fig 9D). CG membranes displayed local accumulation as well as unusual curvature, and NSCs were not separated from each other by CG anymore but were rather found grouped close to each other. Overexpressing Nrg$^{167}$ in CG (from ALH0) produced an even more dramatic phenotype, with localised, compact globules of CG membranes and the complete lack of individual encasing of NSC lineages (Fig 9E). Interestingly, overexpression of an Nrg$^{GPI}$ construct in which the transmembrane and cytoplasmic domains are replaced by a GPI anchor signal [62] also resulted in aggregated CG and clustered NSC lineages (Fig 9E). This shows that intracellular signalling through the divergent C-terminal domain is not required for this sorting of CG and NSC lineages, but

rather that adhesion through the extracellular part mediates this effect. The quantification of NSC encasing upon expression of the different Nrg isoforms in CG confirmed our interpretation (Fig 9F).

Altogether, these results demonstrate that providing Nrg homophilic interactions in the CG is sufficient to segregate them from the whole population of NSC lineages, which they normally bind to through a weaker Nrx-IV to Wrapper interaction. This further suggests that Nrg homophilic adhesions between cells of the same NSC lineage are responsible for keeping these cells together and excluding the CG.

If Nrg interactions are indeed responsible for providing binding between cells of the same NSC lineage, including the stem cell, one consequence is that NSCs could bind to each other. Interestingly, Nrg appears expressed in NSCs only after their encasing (see Fig 8A). This fits the idea that early on, when NSCs are not encased yet and separated from other NSCs by the CG, A (NSC-NSC) is kept low. As such, a precocious expression of Nrg in NSCs would be predicted to lead to their grouping (and further the grouping of their neuronal lineages) in a CG chamber. Strikingly, expressing either of the 3 Nrg isoforms from ALH0 resulted in multiple, larger, and well-defined CG chambers containing several NSCs, all in a similar fashion (Fig 9G–9I). As expressing Nrg$^{GPI}$ also led to the grouping of NSC lineages, it implies that the adhesive role of Nrg is responsible for such effect. This contrasts with the lack of effect of misexpressing Wrapper in NSC lineages (also from ALH0; see Fig 7G and 7H), showing that not all adhesion complexes can lead to A(NSC-NSC) high enough to group NSCs together. These results suggest that a proper timing in establishing intralineage adhesion through Nrg is instrumental in ensuring the individual encasing of NSC lineages by CG.

## Nrx-IV and Nrg adhesions are required in NSC lineages for correct axonal path during development

So far, our data show that Nrx-IV- and Nrg-mediated adhesions in NSC lineages are both important for the individual encasing of NSC lineages by CG. We further sought to assess the functional relevance of such adhesions for the cells of the developing NSC lineages themselves. These functions, if any, could be linked to or independent from their role in niche architecture.

We first turned our eyes to the NSCs. We counted their numbers in the VNC and found no significant difference between *shg*, *nrx-IV*, and *nrg* knockdown in NSC lineages and a control condition (S10A and S10B Fig). As NSC core function is dividing to produce differentiated progeny, we assessed NSC proliferation under *shg*, *nrg*, and *nrx-IV* knockdown in NSC lineages, using phospho-histone 3 (PH3) to mark mitotic DNA. We found that both mitotic indexes and phase distribution in mitosis were similar between these conditions and control (S10C–S10E Fig). We further checked whether NSC were still performing asymmetric division, by staining for the asymmetric determinant aPKC, whose localization at the cell cortex is strongly polarised during mitosis along the division and spindle axis [63]. We first recorded whether aPKC was polarised during metaphase and found no significant difference between control and *shg*, *nrg*, and *nrx-IV* knockdown in NSC lineages (S10F and S10G Fig). We also found no significant difference in the alignment between aPKC polarisation and the axis of the mitotic spindle, detected through α-tubulin staining (S10F–S10H Fig). Taken together, these results show that Nrg and Nrx-IV adhesions in the niche are not critical for NSC survival and asymmetric division.

We then wondered whether neurons were affected upon the loss of Nrg and Nrx-IV adhesions.

Following previous studies showing that impaired CG growth and proliferation can lead to neuronal apoptosis [23,27], we first stained for Dcp-1 (apoptotic marker *Drosophila* cleaved

caspase 1) and found no significant difference in the number of positive cells between *shg*, *nrg*, and *nrx-IV* knockdown in NSC lineages and control (S10I–S10L Fig). This implies that the loss of Nrg and Nrx-IV adhesions does not affect neuronal survival at this stage.

During development, immature secondary neurons start sending axons to establish synaptic connections with proper partners in the neuropile, with axons from the same lineage grouped as 1 or 2 tight bundles following the same path [28,61] (Figs 10A and S1A). This axonal fascicle shows a well-defined tract for each lineage, with stereotyped entry in and path within the neuropile. We thus wondered whether disruption of adhesion in NSC lineages could translate into an altered pattern of axonal projections. To assess this possibility, we first marked NSC lineages in a multicolour clonal fashion, this time using a membrane version of Raeppli (CAAX tag) [37] to label both the cell body as well as the extending axons. Due to the high density of marked lineages, we focused on 1 out of the 4 possible fluorophores for display and quantification (Fig 10B and 10C). We first found that the organisation in bundles of axons from neurons of the same lineage appeared preserved in *shg*, *nrx-IV*, and *nrg* knockdowns in NSC lineages compared to control and that most of them still found their way to the neuropile. We, however, noticed a less regular pattern in their path to the neuropile, drifting from the classic boat shape seen from the antero-posterior view (Fig 10A–10C, view 1) and appearing less aligned in a longitudinal view (Fig 10A–10C, view 2). We calculated the angle of axonal extension to the antero-posterior axis of the VNC (Fig 10D). We found that, compared to a control condition at the same stage, the angles were less stereotyped in overall (Fig 10E and 10F), with a broader distribution and slightly shifted, being either more closed (*nrx-IV* RNAi) or more open (*nrg* RNAi). We did not detect any significant changes upon *shg* knockdown in NSC lineages. These data show that Nrx-IV and Nrg adhesions in the NSC lineages influence the extension of axonal tracts from newborn neurons in the VNC.

We then focused on one specific subset of neurons, with stereotyped projections we could track and measure, and identified by the expression of the cell adhesion protein FasIII [64]. FasIII-positive neurons include motor neurons, RP1, RP3 (also called VL3/4), and RP4 [64–66], born from the activity of specific NSCs (neuroblasts NB3-1 [67]) already at embryonic stages. While these lineages have been mostly characterised during embryogenesis with the formation of primary axonal tracts [64,65,68,69], further neurons and corresponding secondary tracts are added during larval development, as with other lineages [70,71]. At larval stage, FasIII-positive lineages appear as 2 symmetric rows of NSC lineages (cell bodies and axons) on each side of the midline (Fig 10G and controls in Fig 10H and 10I, green tracts and pink stars). FasIII-positive axons connect and generate a ladder pattern at the neuropile level. We stained for FasIII upon *shg*, *nrx-IV*, and *nrg* knockdowns compared to a control condition and observed a strong decrease in FasIII signal when Nrx-IV and Nrg adhesions were disrupted (Fig 10H–10K). In contrast, we did not detect significant changes upon *shg* knockdown in NSC lineages. These data indicate that FasIII-positive motoneurons are altered upon loss of Nrx-IV and Nrg adhesions, displaying decreased expression or localisation of FasIII in their axonal projection.

Altogether, our results show that adhesions within NSC lineages and between NSC lineages and the niche are important for axonal features and path of newborn, still immature neurons.

## The function of Nrx-IV and Nrg adhesions in NSC lineages during development shapes adult locomotor behaviour

Our data show that a balance of specific adhesions is required in the neurogenic niche both for its architecture and for developing neuronal circuits. Whether it has any physiological relevance for the health of the resulting mature organism or is a transient phenomenon whose

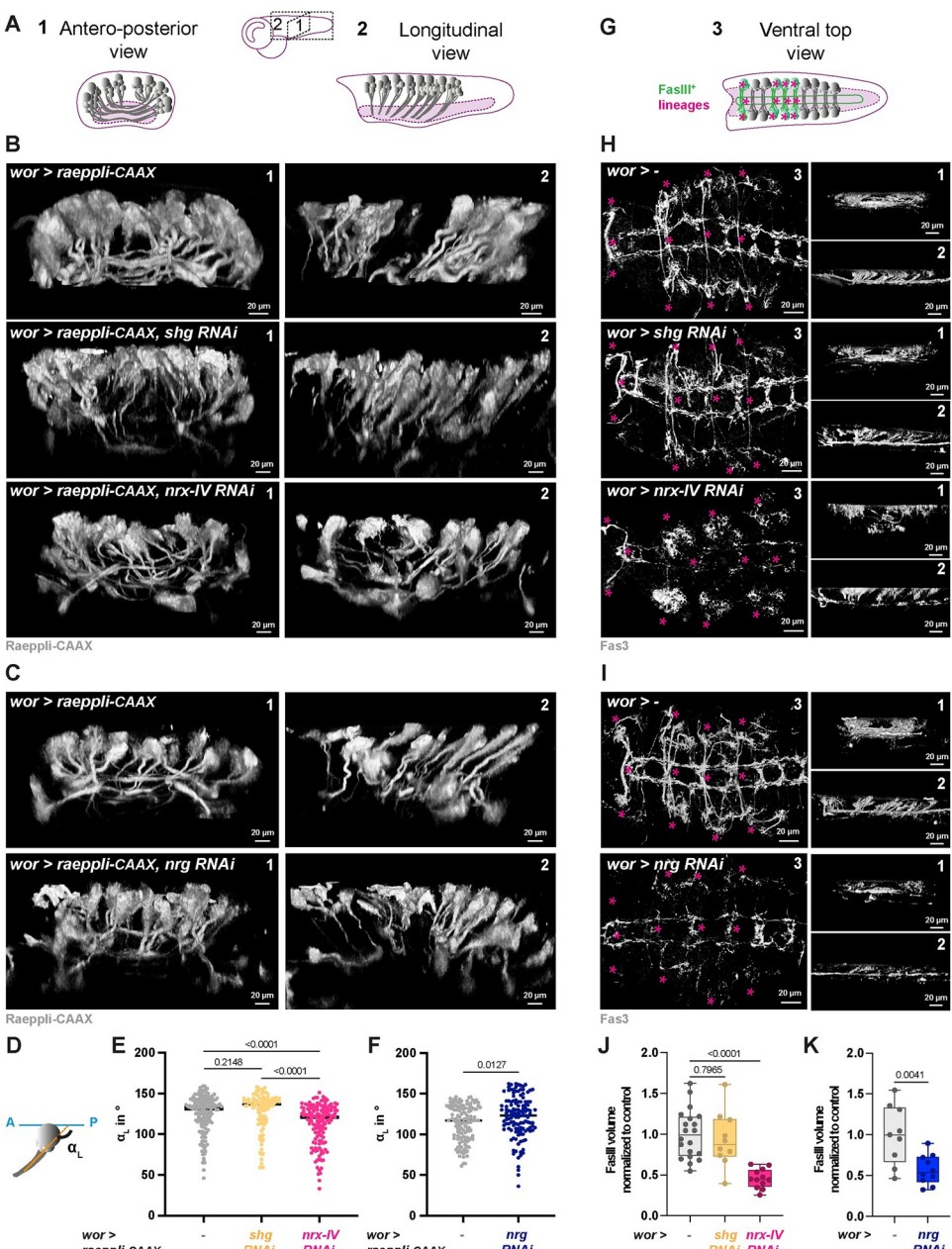

**Fig 10. Loss of Nrx-IV and Nrg adhesions in NSC lineages during development alters axonal projection from newborn secondary neurons.** (**A**) Schematic of the axonal projections (grey) coming from secondary, newborn neurons generated by NSCs during larval neurogenesis. Only the VNC region (third instar larva) is depicted, and not all lineages are represented. The neuropile is shown in a pink shade. (1) Antero-posterior view. (2) Longitudinal view. (**B**) 3D reconstruction of a group of NSC lineages visualised with a membrane marker (mTFP1-CAAX) in antero-posterior (1) and longitudinal (2) views for a control condition (*wor > raeppli-CAAX*), for *shg* knockdown in NSC lineages (*wor > raeppli-CAAX, shg RNAi*, VDRC line 27082) and for *nrx-IV* knockdown (*wor > raeppli-CAAX, nrx-IV RNAi*, BDSC line 32424) in NSC lineages. Clonal labelling was obtained through the induction of Raeppli-CAAX in NSC lineages at ALH0. Larvae were dissected after 72 h at 29˚C. See S1 Table for detailed genetics, timing, and conditions of larval rearing. (**C**) 3D reconstruction of a group of NSC lineages visualised with a membrane marker (mTFP1-CAAX) in antero-posterior (1) and longitudinal (2) views for a control condition (*wor > raeppli-CAAX*) and *nrg* knockdown (*wor > raeppli-CAAX, nrg RNAi*, BDSC line 37496) in NSC lineages. Clonal labelling was obtained through the induction of Raeppli-CAAX in NSC lineages at ALH0. Larvae are dissected after 24 h at 18˚C followed by 60 h at 29˚C. See S1 Table for genetics, timing, and conditions of larval rearing. (**D**) Schematic of the angle ($\alpha_L$) between the main axonal tract projecting from secondary newborn neuron and the antero-posterior axis. (**E**) Quantification of the angle $\alpha_L$ depicted in (**D**) in VNCs for control, *shg* knockdown, and *nrx-IV* knockdown in NSC

lineages, in the same conditions than shown in (**B**). Control, *wor* > *raeppli-CAAX* (*n* = 167 axonal projections from 8 VNCs), *wor* > *raeppli-CAAX*, *shg* RNAi (*n* = 110 axonal projections from 7 VNCs), and *wor* > *raeppli-CAAX*, *nrx-IV* RNAi (*n* = 143 axonal projections from 8 VNCs). Data statistics: Kruskal–Wallis H test with Dunn's multiple comparisons test. *p* < 0.0001 for the Kruskal–Wallis H test on grouped dataset. *P* values from Dunn's multiple comparisons test are displayed on the graph. Results are presented as individual values; the line represents the median. (**F**) Quantification of the angle $\alpha_L$ depicted in (**D**) in VNCs for control and *nrg* knockdown in NSC lineages, in the same conditions than shown in (**C**). Control, *wor* > *raeppli-CAAX* (*n* = 144 axonal projections from 7 VNCs) and *wor* > *raeppli-CAAX*, *nrg* RNAi (*n* = 129 axonal projections from 6 VNCs). Data statistics: Mann–Whitney U test. Results are presented as individual values; the line represents the median. (**G**) Schematic of the axonal projections (green) coming from secondary, newborn neurons generated by FasIII-positive NSCs in the third instar larval VNC in a (3) ventral top view. At this stage, FasIII-positive lineages appear as 2 symmetric groups of 4 lineages each side of the midline; pink stars mark the extremities and middle for each of the 4 groups. FasIII-positive axons connect in a ladder pattern in the neuropile (shown in a pink shade). Not all other NSC lineages (grey) are represented. (**H**) 3D reconstruction of FasIII-positive NSC lineages (stained with an anti-FasIII) in ventral top (3), antero-posterior (1), and longitudinal (2) views for a control condition (*wor* >—(x *w^1118*)), for *shg* knockdown (*wor* > *shg RNAi*, VDRC line 27082) in NSC lineages, and for *nrx-IV* knockdown (*wor* > *nrx-IV RNAi*, BDSC line 32424) in NSC lineages. Larvae are dissected after 68 h at 29°C. Pink stars mark the extremities and middle for each of the 4 groups of FasIII-positive NSC lineages. See S1 Table for genetics, timing, and conditions of larval rearing. (**I**) 3D reconstruction of FasIII-positive NSC lineages (stained with an anti-FasIII) in ventral top (3), antero-posterior (1), and longitudinal (2) views for a control condition (*wor* >—(x *w^1118*)) and for *nrg* knockdown (*wor* > *nrg RNAi*, BDSC line 37496) in NSC lineages. Larvae are dissected after 24 h at 18°C followed by 54 h at 29°C. Pink stars mark the extremities and middle for each of the 4 groups of FasIII-positive NSC lineages. See S1 Table for genetics, timing, and conditions of larval rearing. (**J**) Quantification of the volume of FasIII signal in VNCs for control, for *shg* knockdown, and for *nrx-IV* knockdown in NSC lineages, in the same conditions shown in (**H**). See Methods for volume measure. Volume is normalised to the mean of control. *wor* >—(x *w^1118*) (*n* = 20 VNCs), *wor* > *shg* RNAi (*n* = 10 VNCs), and *wor* > *nrx-IV* RNAi (*n* = 12 VNCs). Data statistics: one-way ANOVA with Tukey's multiple comparisons test. *p* < 0.0001 for the one-way ANOVA test on grouped dataset. *P* values from Tukey's multiple comparisons test are displayed on the graph. Results are presented as box and whisker plots. (**K**) Quantification of the volume of FasIII signal in VNCs for control and for *nrg* knockdown in NSC lineages, in the same conditions shown in (**I**). See Methods for volume measure. Volume is normalised to the mean of control. *wor* >—(x *w^1118*) (*n* = 9 VNCs) and *wor* > *nrg* RNAi (*n* = 10 VNCs). Data statistics: unpaired Student *t* test. Results are presented as box and whisker plots. For all box and whisker plots: whiskers mark the minimum and maximum, the box includes the 25th–75th percentile, and the line in the box is the median. Individual values are superimposed. The data underlying this figure's quantifications can be found in S1 Data. Nrg, Neuroglian; Nrx-IV, Neurexin-IV; NSC, neural stem cell; VNC, ventral nerve cord.

impact is later resolved remain unknown. Beyond selected cases with dramatic outcomes, causally linking defined developmental defects to specific functions of the adult CNS is rarely achieved. Our biological context and findings are particularly relevant to test causality between specific neurodevelopmental processes and adult neurological functions. While secondary neurons are generated during larval development, they are not functional at this stage, but in adults, following their maturation and integration into remodelled circuits during the pupal period [72]. The impact of developmental parameters on their function thus will be detectable, and has to be assessed, at the adult stage.

We decided to determine whether the loss of Nrx-IV and Nrg adhesions in NSC lineages during development impairs adult neurological function. As our analyses of CG and axonal tract phenotypes have focused on the VNC, in which motor neurons are produced (such as the FasIII-positive neurons), we focused on locomotor parameters in the adult. To do so, we took advantage of an ethoscope-based tracking system [73] to record locomotion metrics such as fraction of time moving, velocity, and circadian activity (Fig 11A). This high-throughput platform relies on video acquisition to record positional data in real time for multiple individual flies. Several behavioural parameters can be extracted by calculating the position of the fly overtime, including the fraction of time moving, locomotion speed (velocity), and circadian activity. Statistics on several flies draw an average behaviour for the population.

We recorded locomotion metrics for *shg*, *nrx-IV*, and *nrg* knockdowns in NSC lineages, as well as for a control line, in 2 conditions. First, RNAi expression was only allowed during larval phase and prevented shortly after pupariation (see Methods and Fig 11B, "induced"

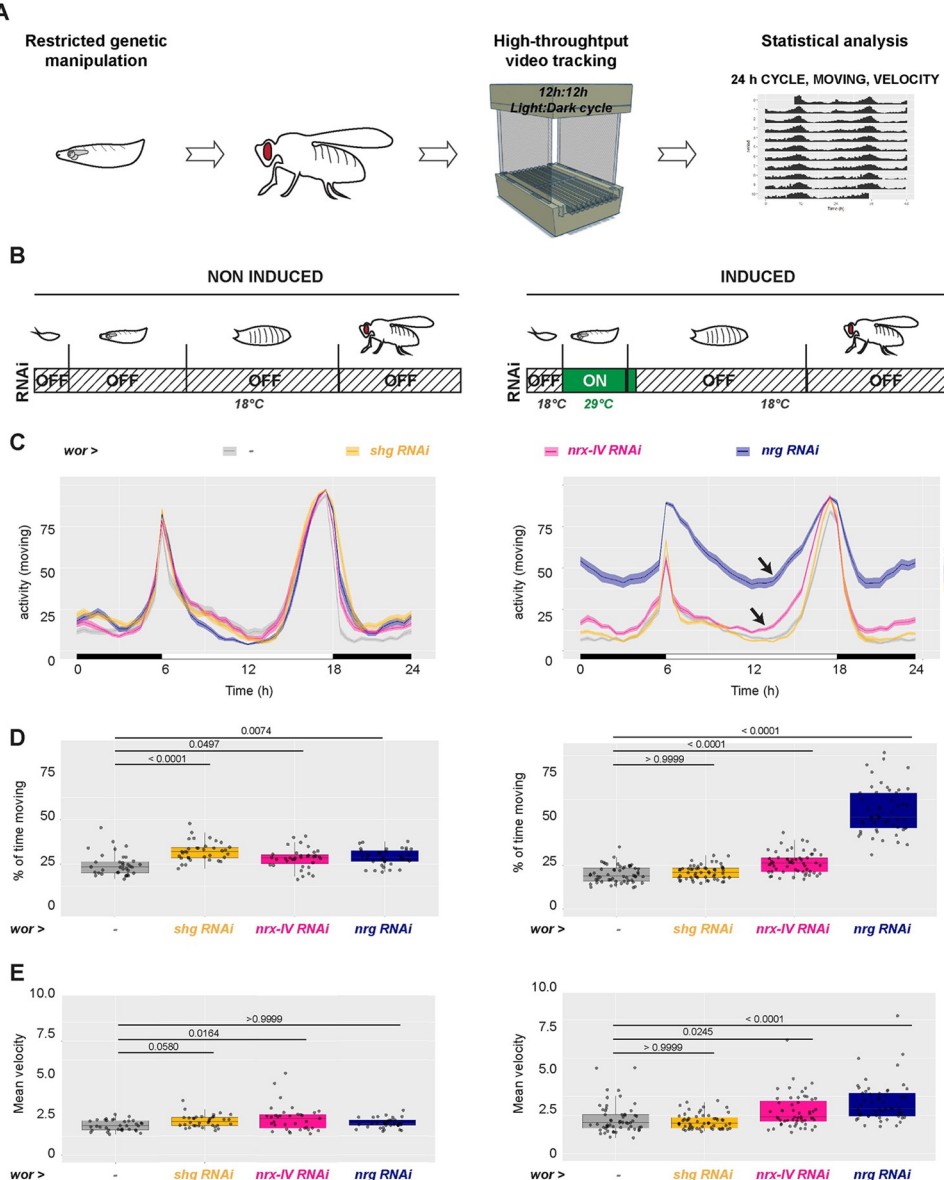

**Fig 11. Loss of Nrx-IV and Nrg adhesions in NSC lineages during development results in locomotor hyperactivity in the resulting adults.** (**A**) Schematics of the behavioural analysis performed on adult flies in which alterations on different adhesion complexes have been performed during the larval stage. The activity of individual flies is recorded by video tracking in ethoscopes. Statistical analysis on the populations allows to infer an average behaviour. (**B**) Schematics of the rearing regimen for non-induced and induced conditions. In non-induced conditions, animals are constantly kept at 18°C before the recordings, a temperature allowing the repression of the GAL4/UAS system by the thermosensitive GAL80^ts and thus blocking expression of the RNAi under the control of the *worniu* driver (*wor-GAL4, tub-Gal80^ts*). In induced conditions, the animals are transiently shifted to 29°C from early larval stage to late larval stage/early pupal stage, a temperature allowing the expression of GAL4/UAS system and thus of the RNAi. Animals were kept at 18°C both during embryonic development and from early pupal stage. Adult flies were assessed 7–10 days after eclosion. See S1 Table for genetics, timing, and conditions of larval rearing. (**C-E**) For all metrics. Control (*wor-GAL4, tub-Gal80^ts x w^1118*), *n* = 35 non-induced adult males and *n* = 55 induced adult males. *shg* RNAi (*wor-GAL4, tub-Gal80^ts x shg RNAi^VDRC27082*), *n* = 35 non-induced adult males and *n* = 55 induced adult males. *nrx-IV* RNAi (*wor-GAL4, tub-Gal80^ts x nrx-IV RNAI^BDSC32424*), *n* = 35 non-induced adult males and *n* = 55 induced adult males. *nrg* RNAi (*wor-GAL4, tub-Gal80^ts x nrg RNAI^BDSC37496*), *n* = 35 non-induced adult males and *n* = 55 induced adult males. (**C**) Plot representing the percentage of global time moving (measured as the fraction of time moving within 30-min intervals), in non-induced and induced conditions. (**D**) Fraction (%) of the time moving across light/dark cycles (% moving, ratio between total moving time and total time) in non-induced and induced conditions. Black arrows indicate anticipation of the evening peak for *nrx-IV* and *nrg* knockdowns. Data statistics: Kruskal–Wallis H test with

Dunn's multiple comparisons test for both non-induced and induced conditions. *p* (non-induced) < 0.0001 and *p* (induced) < 0.0001 for the Kruskal–Wallis H test on grouped dataset. *P* values from Dunn's multiple comparisons test are displayed on the graph. Results are presented as box and whisker plots. (**E**) Mean velocity (in relative units) across light/dark cycles in non-induced and induced conditions. Data statistics: Kruskal–Wallis H test with Dunn's multiple comparisons test for both non-induced and induced conditions. *p* (non-induced) = 0.0115 and *p* (induced) < 0.0001 for the Kruskal–Wallis H test on grouped dataset. *P* values from Dunn's multiple comparisons test are displayed on the graph. Results are presented as box and whisker plots. For all box and whisker plots: whiskers mark the minimum and maximum, the box includes the 25th–75th percentile, and the line in the box is the median. Individual values are superimposed. The data underlying this figure's quantifications can be found in S1 Data. Nrg, Neuroglian; Nrx-IV, Neurexin-IV; NSC, neural stem cell.

condition). Second, gene knockdowns were never activated (same genetic background, but RNAi always off; see Methods and Fig 11B, "non-induced" condition).

We first look at the overall pattern of activity through a 24-h light:dark (LD) cycle. In LD 12 h:12 h conditions, *Drosophila* indeed displays a characteristic rest/activity pattern where they become highly active in anticipation of the transitions between light and dark periods. Rest/sleep takes place mostly during the night and in the middle of light and dark periods. We found that this pattern of activity was kept in the different lines in induced condition, with 2 main peaks of activity (morning and evening, Fig 11C). In addition, we noticed a slightly higher anticipation for the evening peak in the case of *nrx-IV* and *nrg* knockdowns (black arrows), as well as wider peaks for *nrg* knockdown. We then quantified activity metrics by measuring the fraction (%) of time flies spent moving (Fig 11D). We first found a stunning change in the behaviour of *nrg* RNAi flies, which spent 53% of their time moving, while control flies only spent 19% of their time moving (Fig 11D, induced). This dramatic locomotor hyperactivity was apparent throughout both light and dark periods (S11A and S11B Fig). *nrx-IV* RNAi flies also spend significantly more time moving, which was increased to 27% of their time (Fig 11D, induced). In contrast, *shg* RNAi flies appeared similar to control, spending 21% of their time moving. In the non-induced condition, *nrx-IV* RNAI, *nrg* RNAi, and control mostly behaved in a similar fashion (Fig 11D; ctrl = 23%; *nrx-IV* = 27%; *nrg* = 28% of time sleeping), yet with *shg RNAi* displaying increased moving (*shg* = 32%).

We wondered whether this locomotor hyperactivity was only visible as the time flies spent moving, or also in the way they were moving. We then determined the speed of locomotion for the different lines (Fig 11E). In induced conditions, we found that the mean velocity throughout the cycle was significantly increased in *nrg* (3.5, in relative unit (see Methods)) and *nrx-IV* (2.8) but not *shg* (2.1) knockdowns compared to control condition (2.3). In non-induced conditions, all lines exhibited similar values of velocity (ctrl = 2.0; *shg* = 2.2; *nrx-IV* = 2.3; *nrg* = 2.1).

These data show that the loss of Nrx-IV- and Nrg-based adhesions in NSC lineages specifically during development affects adult locomotor behaviour. During the same period, these complexes are also important for building correct niche architecture. We wondered whether these 2 functions were linked or distinct. Nrg perform homophilic interactions within NSC lineages, what makes identifying the contribution of CG-dependent mechanisms challenging. In contrast, Nrx-IV in the NSC lineages works with Wrapper in the CG to set up specific adhesion required for correct niche architecture around NSC lineages. As such, assessing locomotion metrics of *wrapper* knockdown in the larval CG would determine the contribution of such NSC lineage to niche interaction to adult behaviour, beyond potential NSC lineage-intrinsic roles of Nrx-IV. As *wrapper* function is required in the CG for its architecture, it would by itself reveal the dependency of adult locomotion on the developing niche.

We recorded locomotion during a 24-h LD cycle and found that *wrapper* RNAi flies had increased locomotor activity levels, especially for the evening peak (S12A and S12B Fig). They

spent more time moving compared to control (28% compared to 23% of their time, S12C Fig) in induced conditions, a difference we did not detect in non-induced condition (26% compared to 26% of their time, S12C Fig). *wrapper* RNAi flies also displayed increased velocity (ctrl = 1.9 and *wrapper* = 2.4, S12D Fig). These results show that *wrapper* knockdown in the CG during larval development leads to locomotor hyperactivity and thus that its function in the developing niche is required for proper locomotor behaviour in the adult.

Altogether, our results show that the functions of Nrg and Nrx-IV adhesions in the NSC lineages during development are necessary for proper locomotor activity in the adult. This causally links specific processes during CNS development to adult neurological functions. In addition, the results obtained with *wrapper* knockdown in the CG first show that properties of the developing niche shapes adult locomotor behaviour and also suggest that it might be linked to the function of Nrx-IV to Wrapper adhesion in niche architecture. We could not determine whether *nrg* role in adult locomotion was linked to its role in niche formation or rather the output of other NSC lineage-intrinsic functions during development.

## Discussion

The neurogenic niche harbours an elaborate architecture surrounding the stem cells and their differentiating neuronal lineages. However, its mechanisms of formation and its role on NSCs and newborn progeny remain poorly understood. Here, we investigate the formation of glial niches around individual NSC lineages in the *Drosophila* developing CNS. Individual encasing occurs around the NSC itself, before neuronal production, bringing timing as a first mechanism for implementing lineage encasing. Yet, other strategies ensure the formation and maintenance of individual encasing around the entire lineage. We uncovered a sorting between CG and individual NSC lineages through differential adhesion, providing a belt and braces mechanism to ensure lineage encasing regardless of timing. Both adherens and occluding/septate junctions' components are indeed expressed in NSC lineages. While adherens junctions appear mostly dispensable for lineage encasing, 2 components of septate junctions, Nrx-IV and Nrg, are required for this structure, however outside of their junctional roles. Nrx-IV interacts with Wrapper present on the CG, and Nrg, expressed after neuronal production starts, performs homophilic interactions to bind cells from one lineage together. This Nrg-based intralineage adhesion is instrumental in sorting NSC lineage and CG after neuronal production, providing a stronger adhesion compared to the Nrx-IV to Wrapper interaction. Finally, we found that removing Nrg and Nrx-IV to Wrapper adhesions during the larval stage leads to behavioural defects in adult, producing hyperactive flies. Altogether, our findings show that a timely differential adhesion between NSC/NSC lineages and niche cells defines the structure of the niche during development and influences adult behaviour (Fig 12).

Both adherens and occluding junctions have been mostly associated and described in epithelia and epithelial-like tissues. Here, the fact that core components of adherens (Shg, Arm) and occluding (Nrx-IV, Nrg, Dlg1, ATPα, Cora) junctions localise in stem cell and maturing progeny raises questions about their regulation and role in such cell types. While adherens junctions appear specifically set up in NSC lineages, we did not find them to be strictly required for individual encasing by CG, axonal projection, and motor behaviour in adult. Previous studies using a dominant-negative form of Shg [24,43] had reported that Shg disruption altered CG architecture and NSC proliferation. Here, we could not recapitulate such consequences using an efficient RNAi knockdown nor an *shg* null allele (Figs 4C–4F and S4C–S4E). Some of the effects observed with the dominant-negative could be neomorphic and triggered by the activation of other pathways. Another possibility is the fact that knockdown, but not competition by a dominant-negative, could lead to compensation (such as an increase in N-

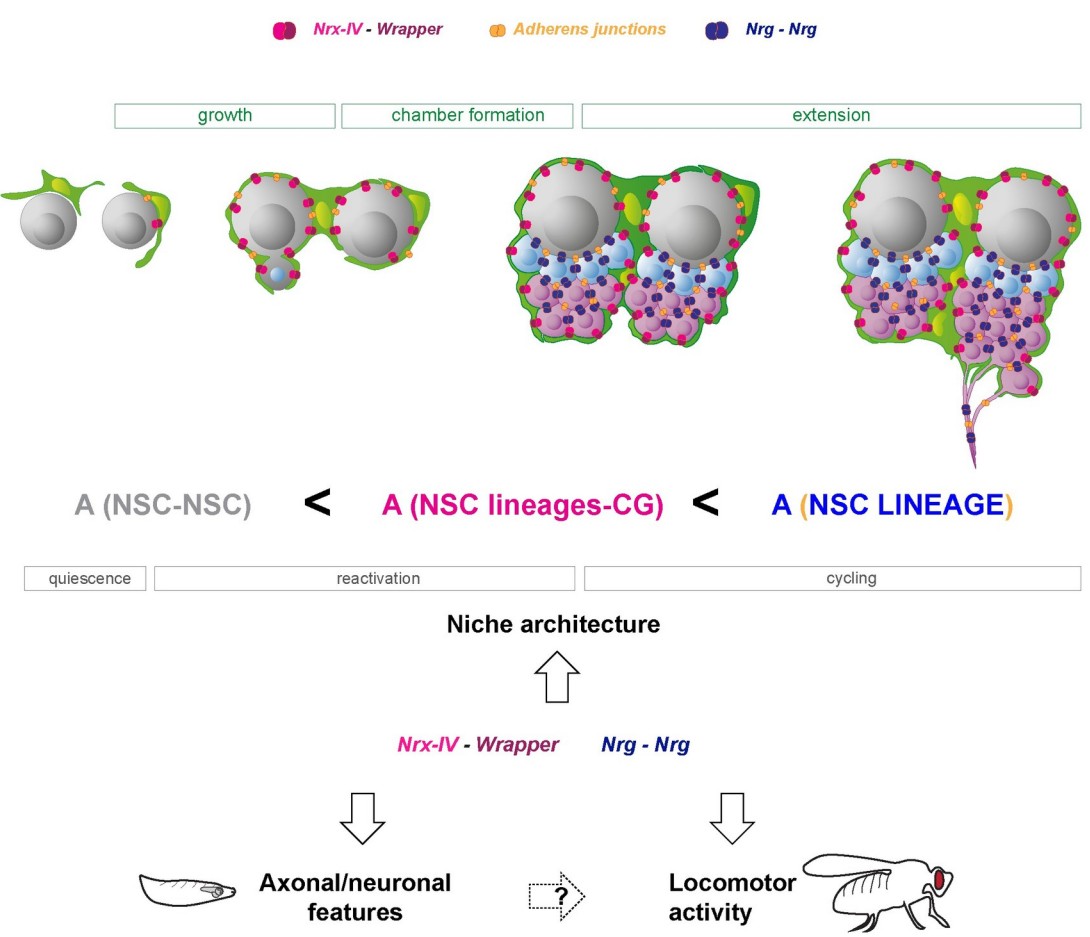

**Fig 12. Timely differential adhesion during development defines the formation of niche architecture around neural stem cells and shapes adult locomotor behaviour.** Schematic depicting the timing and localisation of different adhesion complexes within the NSC niche during larval stages. While Nrx-IV starts to be expressed in NSCs before encapsulation, Nrg only appears afterwards, a timing preventing the clustering of NSCs (by keeping a potential adhesion between NSCs low), and hence later on, NSC lineages, within one chamber. Nrg binds to itself in the NSC lineages (dark blue complexes), while Nrx-IV binds to Wrapper expressed in the CG (pink complexes). Both complexes are crucial in building the individual encasing of NSC lineages by the CG, which is supported by a mechanism of differential adhesion. The adhesion (A) between NSC lineages and CG (involving Nrx-IV and Wrapper) is weaker than the adhesion within the NSC lineage itself (involving Nrg). Adherens junctions (orange complexes) are also present between the cells of the same NSC lineage, where they are mostly dispensable for individual encasing, while potentially providing robustness. The targeted loss of Nrx-IV and Nrg adhesions during larval development is also linked to alterations of axonal features in newborn larval neurons and to locomotor hyperactivity in the resulting adults. CG, cortex glia; Nrg, Neuroglian; Nrx-IV, Neurexin-IV; NSC, neural stem cell.

cadherin), masking the role of Shg and adherens junctions. Other roles in NSC lineages, possibly subtle, also await to be uncovered. For example, *shg* and PDGFR signalling interact in the NSC niche to support NSC survival and CG structure [20].

We propose that a balance between a strong Nrg-based adhesion within the NSC lineages and a weaker, Nrx-IV-based interaction between CG and NSC lineages ($A(L^{NSC}\text{-CG}) < A(L^{NSC})$) builds the stereotyped, individual encasing of NSC lineages by providing differential adhesion. While there is no direct measure of the strength of adhesion between Nrx-IV and Wrapper compared to Nrg with itself, the majority of our findings (see Table 1 for a summary) fit predictions about differential adhesion, based on its definition and description in epithelia. In particular, the fact that misexpressing *nrg* in CG creates CG aggregates and alters individual encapsulation indicates that Nrg can surpass the endogenous Nrx-IV to Wrapper interaction.

**Table 1. Summary of the phenotypes of the individual encasing of NSC lineages by CG depending on adhesion properties.**

| Genetic manipulation | Effect on adhesion | Encasing phenotype |
| --- | --- | --- |
| *shg* knockdown/clones in NSC lineages | Loss of adherens junctions in NSC lineages and between CG and NSC lineages | No effect on encasing |
| *shg* knockdown in CG | Loss of adherens junctions between CG and between CG and NSC lineages | No effect on encasing |
| *shg* overexpression in CG | De novo cadherin-based adhesions in CG and between CG and NSC lineages | No effect on encasing |
| Conditional PTEN expression in CG | - | No effect on encasing |
| Conditional PTEN expression in CG + *shg* knockdown in NSC lineages | Loss of adherens junctions in NSC lineages | Rare occurrence of chambers with multiple NSC lineages |
| *nrx-IV* knockdown in NSC lineages | Loss of Nrx-IV-based adhesion in NSC lineages | Big, round chambers containing several NSC lineages |
| *nrx-IV* knockdown in neurons | Loss of Nrx-IV-based adhesion in neurons | Big, round chambers containing several NSC lineages |
| *nrx-IV* knockdown in CG | Loss of Nrx-IV-based adhesion in CG | No effect on encasing |
| *wrapper* knockdown in CG | Loss of Wrapper-based adhesion in CG | Big, round chambers containing several NSC lineages |
| *wrapper* overexpression in NSC lineages | De novo Wrapper-based adhesion in NSC lineages | No effect on encasing |
| *nrx-IV* and *wrapper* knockdowns in NSC lineages and CG | Loss of Nrx-IV-based adhesion and Wrapper-based adhesions both in NSC lineages and CG | Big, round chambers containing several NSC lineages |
| *nrg* knockdown in NSC lineages | Loss of Nrg-based adhesion in NSC lineages | Clustered NSCs in closed or open CG chambers; rare occurrence of neuron-only chambers. Altered CG membrane morphology |
| *nrg* knockdown in neurons | Loss of Nrg-based adhesion in neurons | Several NSCs per CG chamber |
| *nrg* knockdown in CG | Loss of Nrg-based adhesion in CG | No effect on encasing |
| *nrg* overexpression in CG | De novo Nrg homophilic interaction in CG from ALH0 | Clustered NSCs and aggregated CG membranes |
| *nrg* overexpression in NSC lineages | De novo Nrg homophilic interaction in NSC lineages from ALH0 | Several NSCs per CG chamber |

In this line, misexpressing *wrapper* in the NSC lineages does not alter encasing, suggesting that increasing $A(L^{NSC})$ compared $A(L^{NSC}\text{-}CG)$ does not change the directionality of the difference and, as such, that this difference already exists.

Nrx-IV interaction with Wrapper could provide a scaffold onto which anchoring the glial membrane on the available surface of all lineage cells. When this scaffold is weakened, CG randomly infiltrate in between NSC lineages still tightly linked through Nrg, generating CG chambers of variable size (Figs 6B–6E and 7B–7D). These chambers appear neat, with a clear outline and devoided of CG membrane signal within. Such striking, unmistakable phenotype, which we never observed previously, suggests that upon alteration of Nrx-IV and Wrapper interaction, CG still wrap around NSC lineages as "wholes" but cannot implement their individual encapsulation.

We further propose that Nrg interact with itself in NSC lineages and that such strong interaction is critical for individual encapsulation. The phenotypic expressivity of *nrg* knockdown in NSC lineages is variable, with a minority showing restricted defects, an observation we do

not explain (Fig 8). Nevertheless, in most larvae, we find that *nrg* knockdown in NSC lineages result in their clumping together. Cells from a same lineage still appear to be mostly kept together, one observation not fully fitting the frame of differential adhesion. Indeed, if Nrg binds cells from the same lineage together, its loss would be expected to lead to the individual encapsulation of these cells or at least to their distribution into several chambers, a situation we very rarely observed (S8D Fig). However, this biological system is highly complex, drifting from the classical situation in which differential adhesion had been defined and mostly investigated (i.e., 2 cell types distinguished by different levels of one adhesion complex [39]). Here, 3 cell types (NSC, neuron, and CG) are at play, expressing multiple adhesion complexes. As such, some phenotypes might not follow a toggle rule between sorting and clustering, but rather represent a gradation resulting from multiple, concomitant cues. The existence of other adhesion complexes might be enough to prevent the case where $A(L^{NSC}) < A(L^{NSC}\text{-CG})$ even under *nrg* knockdown. Ultimately, the total sum of adhesions for each cell pair decides of the differential directionality, and we do not know the adhesion strength between CG cells. Finally, other cellular functions might be impacted by *nrg* knockdown, something suggested by the discontinuous aspect of CG membranes (Fig 8), which could pinpoint a growth defect. Interestingly, while most insights on differential adhesion has come from findings on adherens junction/E-cadherin, evidence of intricate interactions involving other adhesion complexes, especially CNS-associated surface immunoglobulins, is steadily emerging [74].

The temporal regulation of Nrg expression appears crucial. We found that Nrg is not present in NSCs before encapsulation (Fig 8A), whereas its precocious expression results in NSC lineages grouped together (Fig 9G–9I). What triggers this timely change, and especially its link with NSC reactivation, is intriguing. Indeed, what first recruits CG membrane to NSC, before creating and clustering a lineage, remains to be identified.

Nrg comes in 2 isoforms, associated with specific cell types and functions. We found that the Nrg[180] isoform was expressed in newborn secondary neurons, fitting its known neuronal localization at other life stages. However, Nrg[167], traditionally associated with junctional localisation, appears also present. Whether Nrg[167] interacts with Nrg[180] for NSC encasing, or has another junctional role, remains to be demonstrated.

The loss of Nrx-IV and Nrg adhesions both result in altered axonal projections from the newborn neurons. It could be a consequence of some autonomous properties of these molecules in neurons, particularly for Nrg performing intralineage homophilic interactions. Improper encasing of NSC lineages could also remove some physical constraints on the axons of newborn neurons required for their path. Interestingly, previous studies have linked a change in CG structure or function to misshaped axonal tracts in the larval CNS, both in the larval OL [17] and CB, where CG ablation resulted in abnormal axonal trajectories and fasciculation [30].

We further linked the loss of Nrx-IV and Nrg adhesions during development to changes in the locomotion of the resulting adults, which appeared hyperactive. In *Drosophila*, hyperactivity had been previously associated with diverse models of neuro/developmental and neurological disorders, including the Fragile X syndrome [75], attention-deficit/hyperactivity disorder (ADHD) [76], and Shwachman–Diamond syndrome [77]. While Nrg could be involved in other ways than through building the niche structure, due its homotypic interactions between neurons and its strong axonal localization, Nrx-IV and Wrapper interaction supports bridging niche architecture with adult behaviour. First, Nrx-IV to Wrapper interaction is between NSC lineages and CG, rather than between neurons. Second, in *nrx-IV* knockdown in NSC lineages and *wrapper* knockdown in CG, CG chambers are still present, neatly delineated around multiple NSC lineages, depicting the loss of the individuality of encasing rather than a comprehensive alteration of CG structure. Third, we observe very similar locomotion metrics between

*nrx-IV* knockdown in NSC lineages and *wrapper* knockdown in CG. This pleads for the contribution of specific niche architecture during development to adult behaviour. *wrapper* results demonstrate regardless that CG properties during development influence adult behaviour.

Here, we propose a mechanism in which the temporal and spatial regulation of different adhesion complexes builds a stereotypic niche organising individual NSC lineages and their progeny. Their function is also important for axonal projection of newborn neurons, and locomotor behaviour in the adult, thus linking niche adhesive properties and developmental neurogenesis to adult health. All these complexes are conserved in mammals, warranting the question of their nonjunctional role in a developing CNS.

## Methods

### Fly lines and husbandry

*Drosophila melanogaster* lines were raised on standard cornmeal food at 25˚C. Lines used in this study are listed in Table 2.

### Larval culture and staging

Embryos were collected within 2 to 4 h window on grape juice agar plates and kept at 25˚C for 20 to 24 h. Freshly hatched larvae were collected within a 1-h time window (defined as 0 h after larval hatching, ALH0), transferred to fresh yeast paste on a standard cornmeal food plate and staged to late first instar (ALH24), late second instar (ALH48), mid third instar (ALH72), and late third instar (ALH96).

For growth on quinic acid, food plates were prepared by mixing 250 mg/ml stock solution of quinic acid (dissolved in sterile water) into melted food at 50˚C for a final concentration of 20 mg/ml of quinic acid.

For *D. melanogaster*, 25˚C was used as the normal developmental temperature. To keep the thermosensitive allele of GAL80 (GAL80$^{ts}$), a repressor of GAL4, active (so to switch off GAL4 expression), 18˚C was used, while 29˚C was used for its inactivation (so to switch on GAL4 expression). For all experiments using the GAL80$^{ts}$, embryogenesis happened at 18˚C, unless noted otherwise. This was to prevent a contribution of embryonic function to the larval phenotype.

Detailed genotypes, crosses, and culture regimens are listed in S1 Table.

### DNA cloning and *Drosophila* transgenics

A portion of the *cyp4g15* enhancer (GMR55B12, Flybase ID FBsf0000165617), which drives in the CG and (some) astrocyte-like glia, was amplified from genomic DNA extracted from *cyp4g15-GAL4* adult flies, with a minimal *Drosophila* synthetic core promoter [DSCP] [81] fused in C-terminal. For creating *cyp4g15-FRT-STOP-FRT-GAL4*, an FRT STOP cassette was amplified from a *UAS-FRT.STOP-Bxb1* plasmid (gift from MK. Mazouni), and the GAL4 sequence was amplified from the entry vector pENTR L2-GAL4::p65-L5 (gift from M. Landgraf). The 2 amplicons were joined together by overlapping PCRs. This *FRT-STOP-FRT-GAL4* amplicon together with the cyp4g15$^{DSCP}$ enhancer were inserted in the destination vector pDESThaw sv40 (gift from S. Stowers) using the Multisite gateway system [82] to generate a *cyp4g15$^{DSCP}$-FRT-STOP-FRT-GAL4* construct. The construct was integrated in the fly genome at an attP2 or attP40 docking sites through PhiC31 integrase-mediated transgenesis (Best-Gene). Several independent transgenic lines were generated and tested, and one was kept for each docking site.

**Table 2.** *Drosophila* transgenic lines used in this study.

| Strains | Source | Stock number/ Reference |
|---|---|---|
| $w^{1118}$ | BDSC | 5905 |
| *Nervana2::GFP (Nrv2::GFP)* | BDSC | 6828 |
| *shg::GFP* | Yohanns Bellaïche lab | [44] |
| *Nrx-IV::GFP* | Christian Klämbt lab | $Nrx^{454}$, [78] |
| *Nrg::GFP* | Kyoto | 110658 |
| $Nrg^{167}$*::GFP* | BDSC | 6844 |
| *dlg1::GFP* | BDSC | 50859 |
| *ATPalpha::GFP (CPTI)* | Kyoto DGGR | 115323 |
| *tubulin-GAL80$^{thermosensitive(ts)}$* | BDSC | 65406 |
| *yw, hs-FLP* | Andrea Brand lab | |
| *Cre recombinase* | BDSC | 851 |
| *FRT G13* | BDSC | 1956 |
| *CoinFLP* | BDSC | 58750 |
| *cyp4g15-GAL4* | BDSC | 39103 |
| *cyp4g15-FRT-STOP-FRT-GAL4* | This study | |
| *cyp4g15-QF2* | Spéder lab | [27] |
| *cyp4g15-FLP* | Spéder lab | [27] |
| *cyp4g15-mtd::Tomato* | Spéder lab | [27] |
| *elaV-GAL4$^{C155}$* | BDSC | 458 |
| *nSyb-GAL4* | Andrea Brand lab | |
| *worniu-Gal4 (chromosomes II and III)* | BDSC (Doe lab insertions) | 56553 & 56554 |
| *Dpn-FRT-STOP-FRT-LexA* | BDSC | 56162 |
| *grainyhead$^{D4}$-FLP (grh$^{D4}$-FLP)* | This study | |
| *tub-QS* | BDSC | 52112 |
| *TUG G13 MARCM line*<br>*y,w, hs-FLP; FRTG13, tubP-GAL80[LL2]/ (CyO, act-GFP[JMR1]); tubP-GAL4[LL7], UAS-mCD8-GFP[LL6]/ TM6B* | Bruno Bello | |
| *tubP-GAL80[LL2]* | BDSC | 5140 |
| *UAS-reaper* | Andrea Brand lab | |
| *UAS-mCD8::GFP* | BDSC | 5130 |
| *UAS-mCD8::RFP* | BDSC | 27399 |
| *UAS-myr::mCherry* | François Schweisguth lab | |
| *UAS-Histone2B::RFP* | Yohanns Bellaïche lab | [79] |
| *UAS-Raeppli CAAX 43E* | Generated from BDSC 55082 | [27] |
| *UAS-Raeppli NLS 53D* | Generated from BDSC 55087 | [27] |
| *UAS-Raeppli NLS 89A* | Generated from BDSC 55088 | [27] |
| *LexAOp-Raeppli-NLS 89A* | BDSC, This study | 55088 |
| *UAS-prospero RNAi* | Andrea Brand lab | |
| *UAS-brat RNAi* | Andrea Brand lab | |
| *UAS-Dpn* | Jurgen Knoblich lab | |
| *UAS-aPKC [CAAX]* | | [80] |
| *UAS-shg RNAi* | VDRC | 27082 |
| | VDRC | 103962 |
| *UAS-nrx IV RNAi (Chromosome III)* | BDSC | 32424 |
| *UAS-nrx IV RNAi (Chromosome II)* | VDRC | 9039 |

*(Continued)*

**Table 2.** (Continued)

| Strains | Source | Stock number/ Reference |
|---|---|---|
| *UAS-nrg RNAi* | BDSC | 37496 |
| *UAS-wrapper RNAi* (Chromosome III) | BDSC | 29561 |
| *UAS-wrapper RNAi* (Chromosome II) | VDRC | 105314 |
| *UAS-ATPα RNAI* | BDSC | 28073 |
| *UAS-RFP RNAi* | BDSC | 67852 |
| *UAS-mCherry RNAi* | BDSC | 35787 |
| *UAS-nrg$^{180}$* | BDSC | 24169 |
| *UAS-nrg$^{167}$* | BDSC | 24172 |
| *UAS-nrg$^{GPI}$* | BDSC | 24168 |
| *UAS-wrapper* | BDSC | 78535 |
| *QUAS-PTEN* | This study | |
| *shg$^{null}$* (*shg$^{R64a}$*) | Yohanns Bellaïche lab | |

The *grainyhead* (*grh*) D4 enhancer (4 kb from the second intron of the *grainyhead* gene), which has been reported to drive in postembryonic NSCs [8,42], was amplified from genomic DNA extracted from *grh (NB)-GAL4* adult flies, with an *hsp70* promoter fused in C-terminal. For creating *grh$^{D4}$-FLP*, the *FLP* DNA, which codes for the flippase enzyme, was amplified from the plasmid pMH596 (Addgene 52531) and was joined to the *grh$^{hsp70}$* enhancer using the Multisite gateway system in the destination vector pDESThaw sv40. The construct was integrated in the fly genome at an attP18 docking site through PhiC31 integrase-mediated transgenesis (BestGene). Several independent transgenic lines were generated and tested, and one was kept (*grh$^{D4}$-FLP*).

For creating *QUAS-PTEN*, the *PTEN* coding sequence was amplified from genomic DNA extracted from *UAS-PTEN* [83] adult flies, as described in [84]. This amplicon together with the QUAS sequence (pENTRY L1-QUAS-R5, gift from S.Stowers) were joined using the Multisite gateway system [82] in the destination vector pDESThaw sv40. The construct was integrated in the fly genome at an attP40 docking site through PhiC31 integrase-mediated transgenesis (BestGene). Several independent transgenic lines were generated and tested, and one was kept (*QUAS-PTEN*).

## Generation of UAS-Raeppli-NLS 89A and LexAOp-Raeppli-NLS 89A

The original construct (BDSC 55088), placing Raeppli NLS under the control of both UAS and LexAOp sequences, was crossed to a Cre recombinase line (BDSC 851) to randomly excise one of the 2 control sequences. The resulting lines were checked by PCR to determine whether they carried the UAS or LexAop version.

## Fixed tissue immunohistochemistry

For immunohistochemistry, CNS from staged larvae were dissected in PBS, fixed for 20 min in 4% formaldehyde diluted in PBS, washed 3 times in PBS-T (PBS+0.3% Triton X-100), and incubated 2 nights at 4°C with primary antibodies diluted in blocking solution (PBS-T, 5% bovine serum albumin, 2% normal goat serum). After washing 3 times in PBS-T, CNS were incubated overnight at 4°C or 3 to 4 h at room temperature with secondary antibodies (dilution 1:200) diluted in blocking solution. Brains were washed 3 times in PBS-T and mounted in Mowiol mounting medium on a borosilicate glass side (number 1.5; VWR International). For

the Nrx-IV antibody, CNS were fixed for 3 min in Bouin's fixative solution (Sigma Aldrich, HT10132), and the rest of the protocol was identical. Primary antibodies used were as follows: guinea pig anti-Dpn (1:5,000; [27]), chicken anti-GFP (1:2,000, Abcam ab13970), rat anti-ELAV (1:100, DSHB 7E8A10-c), mouse anti-ELAV (1:100, DSHB 9F8A9-c), rat anti-dE-cadherin (1:50, DSHB DCAD2), mouse anti-Armadillo (1:50, DSHB N2 7A1), rabbit anti-Repo (1:10,000, kind gift from B. Altenheim), mouse anti-Repo 1:100 (DSHB 8D12-c), mouse anti-Prospero (1:100, DSHB MR1A), rabbit anti-Asense (1:3,000, kind gift from the Yan lab), rabbit anti-Phospho-histone H3 (1:100, Millipore 06–570), rabbit anti-Nrx-IV (1:1,000; [51]), mouse anti-wrapper (1:20, DSHB 10D3, supernatant), mouse anti-Nrg$^{180}$ (1:50, DSHB BP104, supernatant), rabbit anti-Dcp-1 (1/100, Cell Signalling 9578S), rabbit anti-aPKc (1/500, Santa Cruz Biotechnology sc-17781), mouse anti-α-tubulin (1/200, clone DM1A, Cell Signalling Technology #3873), mouse anti-FasIII (1/50, DSHB 7G10, concentrate). Fluorescently conjugated secondary antibodies Alexa Fluor 405, Alexa Fluor 488, Alexa Fluor 546, and Alexa Fluor 633 (Thermo Fisher Scientific) were used at a 1:200 dilution. DAPI (4′,6-diamidino-2-phenylindole, Thermo Fisher Scientific 62247) was used to counterstain the nuclei.

## RNA FISH (HCR in situ hybridization)

We used the Multiplexed HCR RNA-FISH technique [85,86], using reagents and adapted protocols from Molecular Instruments. First, *wrapper* probe set was designed using software from Molecular Instruments (https://www.molecularinstruments.com/). RNA FISH was then performed as follows. First, CNS from staged larvae were dissected in PBS, fixed for 20 min in 4% formaldehyde diluted in PBS, washed 3 times 5 min in PBS and prehybridized in 200 µl of hybridization buffer for 30 min at 37˚C. Meanwhile, 0.8 pmol (0.8 µl of 1 µM stock) of the probe set was added to 200 µl of probe hybridisation buffer and prehybridised for 30 min at 37˚C. The prehybridization solution was removed from the fixed samples, which were then incubated with the probe solution at 37˚C overnight. Samples were washed 4 × 15 min with 500 µl warmed (37˚C) wash buffer, followed by 5 min with 500 µl of 50% wash buffer/50% 5XSSC-0.1% Tween and, finally, by 2 × 5 min with 500 µl of 5XSSC-0.1% Tween at room temperature. Samples were further preamplified with 100 µl of amplification buffer for 10 min at room temperature. In the meantime, 6 pmol of hairpin h1 and 6 pmol of hairpin h2 (2 µL of 3 µM stock) were individually prepared by heating at 95˚C for 90 s followed by cooling to room temperature for 30 min in the dark. Snap-cooled hairpins h1 and h2 are then added to 100 µL of amplification buffer at room temperature. The preamplification solution was removed from the samples, which were incubated in the hairpin solution overnight (>12 h) in the dark at room temperature. The samples were washed 2 × 5 min, 2 × 30 min, and 1 × 5 min with 5XSSC-0.1% Tween before mounting on slides.

For the ALH0 time point, 4 pmol (4 µl of 1 µM stock) of the probe set was used instead of 0.8 pmol.

## Image acquisition and processing

Confocal images were acquired using a laser scanning confocal microscope (Zeiss LSM 880, Zen software (2012 S4)) with a Plan-Apochromat 40×/1.3 oil objective. All brains were imaged as z-stacks with each section corresponding to 0.3 to 0.5 µm. The spectral mode was used for acquiring pictures of Raeppli clones. Images were subsequently analysed and processed using Fiji (Schindelin, J. 2012), Volocity (6.3 Quorum Technologies), and the Open-Source software Icy v2.1.4.0 (Institut Pasteur and France Bioimaging, licence GPLv3). Denoising was used for some images using the Remove noise function (Fine filter) in Volocity. Images were assembled using Adobe Illustrator 25.4.6.

### $shg^{null}$ MARCM clones

$shg^{64R}$, FRT42B; cyp4g15-mtd::Tomato flies were crossed to the TUG13 MARCM line (y,w, hs-FLP; FRTG13, tubP-GAL80[LL2]/ (CyO, act-GFP[JMR1]); tubP-GAL4[LL7], UAS-mCD8-GFP [LL6]/TM6B). The resulting progeny was let to develop at 25°C, then subjected to 37°C heat-shock either at 14 to 18 h after egg laying for 2 h or at ALH48 for 30 min, and dissected at ALH72 (S4C Fig).

### Quantification of Type II chambers and clones

Each CB in which either control or brat tumour Type II lineages were marked with Raeppli-NLS was manually scored throughout the entire dorso-ventral z-stack for the number of CG chambers containing marked lineages. A CG chamber was defined by the existence of a continuous Nrv2::GFP signal around a volume of adjacent cells. In the brat condition, some chambers were very big and convoluted and, thus, difficult to score. We decided to eliminate the corresponding CBs. The number of differently coloured clones and the total number of colours (for scattered cells) within one chamber (and within the entire CB) were also recorded manually.

### Quantification of NSC (Dpn staining) and NSC lineage encasing (Raeppli-NLS)

NSCs are identified by Dpn staining and NSC lineages by the use of the multicolour clonal tool Raeppli-NLS, whose early induction under the control of worniu-GAL4 leads to the labelling of clonally related cells with the same colour. CG are tracked by a membrane marker (either Nrv2::GFP or cyp4g15-myr::mtd-Tomato). A CG chamber is defined as a continuous layer of membranes around a group of cells, regardless of their identity. The continuity is manually assessed by visualisation through the entire z-stack, also making use of orthogonal views.

One NSC is scored as individually encased if no other NSC (Dpn⁺ cell) can be detected within the same CG chamber. Conversely, one NSC is scored as non-individually encased if at least one other NSC (Dpn⁺ cell) can be detected within the same CG chamber. The percentage of non-individually encased NSCs is calculated by dividing the count of non-individually encased NSCs by the total count of NSCs in each VNC.

One NSC lineage is scored as individually encased if no more than one colour of Raeppli-NLS can be detected within the same CG chamber. Conversely, one NSC lineage is scored as non-individually encased if at least one other colour for Raeppli-NLS can be detected within the same CG chamber (i.e., 2 different colurs are not fully separated by CG membrane). The percentage of non-individually encased NSC lineages is calculated by dividing the count of non-individually encased NSC lineages by the total count of NSC lineages in each VNC.

### Clonal analyses using CoinFLP in the CG

The Coin-FLP method [38] was used to induce rare clones of PTEN- or reaper-overexpressing CG cells, by crossing cyp4g15-FLP; CoinFLP GAL4::LexA; UAS-mCD8::RFP females to Nrv2:: GFP/CyO; UAS-PTEN or Nrv2::GFP, tub-GAL80ᵗˢ/CyO; UAS-reaper males. Crosses to Nrv2:: GFP/CyO and Nrv2::GFP, tub-GAL80ᵗˢ/CyO males were respectively used for control. For PTEN overexpression clones, larvae were maintained at 25°C and staged to ALH48-ALH72 for dissection. For reaper overexpression clones, larvae were maintained at 18°C for 96 h after larval hatching then shifted to 29°C for 48 h and dissected.

Clone volume (using the mCD8::RFP signal) was segmented and measured in Volocity 6.3 (Quorum Technologies) using adjusted protocols for intensity thresholding. Threshold were

corrected on each sample to exclude background fluorescence. NSC number per clone was scored manually by counting the number of NSCs encased in RFP$^+$ membrane. Number of non-encased NSCs around clones was scored manually by counting the NSCs, which were not individually encased by the CG (marked with Nrv2::GFP).

## Clonal analyses using CoinFLP in Type II NSC lineages

To generate *brat* RNAi clones in Type II NSC lineages, *grh-FLP$^{D4}$; CoinFLP GAL4::LexA* females were crossed to *Nrv2::GFP, UAS-mCD8::RFP/CyO; UAS-brat RNAi* males. A cross to *Nrv2::GFP, UAS-mCD8::RFP/CyO* males was used for control. Larvae were maintained at 25˚C and staged to ALH72 to ALH96 for dissection. RFP$^+$ clones were manually assessed for complete enclosure by CG membrane (marked with Nrv2::GFP).

## Cortex glial membrane volume measurements

Nrv2::GFP signal was used as a proxy for CG membrane signal. Each VNC was sampled with 2 rectangular prisms (x = 100 μm; y = 50 μm; z = 3 μm) on each part of the midline, centred on the neuron level of the chamber (where *nrx-IV* and *wrapper* phenotypes are the most visible and devoid of trachea or nerve signal). The total volume of CG membrane for these areas was segmented and measured in Volocity 6.3 (Quorum Technologies) using adjusted protocols for intensity thresholding. Threshold were corrected on each sample to exclude background fluorescence.

## Quantification of levels of RNAi knockdown

For each gene, one reporter of expression was chosen and the sum of its own volume in a defined volume within the VNC was measured under control and RNAi genotypes. Reporter volume was measured in Volocity 6.3 (Quorum Technologies) using adjusted protocols for intensity thresholding. Threshold were corrected on each sample to exclude background fluorescence. Each value was then normalised to the mean of the control. The conditions for each gene were as follows (Table 3).

**Table 3. Choices for the reporter and tissue volume used to quantify RNAi knockdowns corresponding to the different adhesion molecules.**

| Gene | Reporter | Volume of calculation | Reason for choice of volume | Figures |
|---|---|---|---|---|
| *shg* | Shg::GFP | - 120 × 120 × 10 μm<br>- From the NSC level up to before the neuropile | Centred on NSC lineages | S4D and S4E Fig |
| *nrx-IV* | Antibody | - 120 × 120 × 10 μm<br>- From the NSC level up to before the neuropile | - Avoiding SJ signal from the blood–brain barrier<br>- Centred on NSC lineages | S6B and S6C Fig |
| *wrapper* | mRNA | - 120 × 120 × 10 μm<br>- From the CG layer covering NSCs<br>- Normalised to CG volume | - Avoiding signal from the midline glia<br>- Accounting for decreased density of CG membrane in *wrapper* RNAi | S7A and S7B Fig |
| *nrg* | Nrg::GFP | - 2 times (40 × 40 × 2 μm)<br>- Centred on secondary lineages, each side of the midline | - Avoiding strong septate junction signal from the blood–brain barrier, including from the midline traversing channels<br>- Smaller CNS due to earlier timepoint | S8A and S8B Fig |
| | Nrg$^{167}$::GFP | | | S9E and S9F Fig |
| | BP104 (anti-Nrg$^{180}$) | - 120 × 120 × 5 μm<br>- From the NSC level up to before the neuropile | Smaller CNS due to earlier time point (5 μm depth instead of 10 μm) | S9C and S9D Fig |

## Quantification of NSC number and mitotic index

CNSs of the chosen genotypes were stained with Dpn and phospho-histone H3 antibodies to detect NSC identity and mitosis, respectively. Quantification was performed on NSCs from the thoracic part of the VNC. Mitotic phases (Prophase, Prometaphase/metaphase, Anaphase, and Telophase) were manually determined by the localization and pattern of PH3+ DNA and Dpn staining. Normalised mitotic index corresponds to the ratio between mitotic NSCs over all NSCs, then divided by the mean of this ratio for the control sample.

## Quantification of aPKC polarisation

Quantification was performed on NSCs from the thoracic part of the VNC. aPKC was detected using a specific antibody, mitotic spindles were labelled with anti-α-tubulin, and DNA was counterstained with DAPI. NSCs were recognised by their size. aPKc localization was assessed during metaphase, recognised through its stereotypic DNA staining (metaphasic plate), and scored as polarised when a clear crescent was seen restricted to part of the NSC. The mitotic spindle was considered aligned when perpendicular to the main axis of aPKC crescent and of the metaphasic plate and considered misaligned when titled compared to this axis. The mitotic spindle was finally scored as undetermined when we could not detect it on all its length, often when not parallel to the plan of confocal acquisition. Percentages per VNC were calculated by dividing the total count of each of the 3 categories by the sum of the counts.

## Quantification of apoptotic (Dcp-1+) cells

CNSs of the chosen genotypes were stained with anti-Dcp-1 antibody to detect apoptotic cells. Quantification was performed on the thoracic part of the VNC. Dcp-1+ cells were detected in Volocity 6.3 (Quorum Technologies) using adjusted protocols for intensity thresholding and object detection. Threshold was corrected on each sample to exclude background fluorescence.

## Quantification of axonal angle

Z stacks of Raeppli CAAX clones induced in NSC lineages were visualised in Volocity (6.3 Quorum Technologies), in a lateral view along the antero-posterior axis (xz axis). The brightest colour (mTFP1) was chosen and lines were drawn parallel to and following the main axonal projection for each clone. Pictures of the line were recorded through snapshots and imported into Icy v2.1.4.0 where the Angle Helper plugin was used to measure the angle formed with the intersection with the AP axis (Fig 10D).

## Quantification of FasIII signal

CNSs of the chosen genotypes were stained with a specific anti-FasIII antibody. Quantification was performed on a standardised section of the thoracic VNC spanning the 3 central FasIII+ NSC lineages (Fig 10G), cut left of the most anterior lineage and right of 2 commissural axons after the most posterior lineage. Total FasIII volume was detected in Volocity 6.3 (Quorum Technologies) using adjusted protocols for intensity thresholding. Threshold was corrected on each sample to exclude background fluorescence. Each value was normalised to the mean of the respective control sample.

## Behavioural analysis

The locomotor activity of individual flies was measured with the *Drosophila* ethoscope activity Open Source system [73] at 23°C. Young adult males (7 to 10 days) were individually placed in

6.5 cm transparent tubes containing 1.5 cm nutrient medium (agarose 2% (p/v), sucrose 5% (p/v)) closed with wax at one end and cotton at the other. A total of 20 tubes were positioned in each ethoscope, and flies were first entrained to 12 h:12 h LD cycles for 3 days and their activity was recorded for 3 more days. The activity data analysis was done with the R software, using Rethomics packages [87]. The percentage of time moving across LD cycles was measured as the fraction of time moving within 30-min intervals, whereas global time moving per fly corresponds to a total moving over total time ratio. Velocity was measured using a previously described tracking algorithm [87] and is expressed in relative units. Genotypes positions in 20 tubes arenas were changed from one experiment to another to avoid positional bias.

The genotypes and sample sizes were the following:

- Figs 11 and S11: *wor-GAL4, tub-GAL80$^{ts}$* x *w$^{1118}$*; *wor-GAL4, tub-GAL80$^{ts}$* x *shg RNAi$^{VDRC27082}$*; *wor-GAL4, tub-GAL80$^{ts}$* x *nrx-IV RNAi$^{BDSC32424}$*; *wor-GAL4, tub-GAL80$^{ts}$* x *nrg RNAi $^{BDSC37496}$* for knockdown in NSC lineages. A total of 35 non-induced flies and 55 induced flies, coming from 8 independent experiments, were analysed for each genotype.

- S12 Fig: *tub-GAL80$^{ts}$; cyp4g15-GAL4* x *w$^{1118}$*; *tub-GAL80$^{ts}$; cyp4g15-GAL4* x *wrapper RNAi $^{BDSC29561}$* for knockdown in CG. A total of 20 non-induced flies and 20 induced flies, coming from 3 independent experiments, were analysed for each genotype.

In non-induced flies, RNAi expression was always kept off by maintaining the progeny issued from the cross at 18˚C at all times (ALH0 to assay time). In induced flies, RNAi expression was only activated during larval stage: The progeny was kept at 18˚C from egg laying to ALH24, then shift to 29˚C from ALH24 to just after pupariation/early pupae (shifted back to 18˚C within 1 day at 29˚C from wandering L3 larvae), and finally maintained at 18˚C until the behavioural assay).

## Statistics and reproducibility

Statistical tests used for each experiment are stated in the figure legends. Statistical tests were performed using GraphPad Prism 7.0a or in R for the generalised linear model (Binomial regression with a Bernoulli distribution). For all box and whisker plots, whiskers mark the minimum and maximum, the box includes the 25th to 75th percentile, and the line in the box is the median. Individual values are superimposed.

For all knockdown (RNAi) and misexpression experiments, penetrance was always 100% (n numbers are indicated in the figure legends), and expressivity was represented through sample distribution for the parameter assessed.

## Supporting information

**S1 Fig. CG encase individual NSC lineages.** (**A**) Confocal picture of one NSC lineage labelled with a membrane marker (mTFP1-CAAX, grey). NSC lineages are marked with the multicolour lineage tracing Raeppli-CAAX (blue, white, orange, and red) under the control of the GAL/UAS system and induced at ALH0 using *hs-Flp* (*wor > raeppli-CAAX*). Larvae are dissected after 68 h at 29˚C from ALH0. One mTFP1$^+$ clone is shown (*n* > 45 clones). CG membrane was visualised with *Nrv2::GFP* (green). (**A'**) 3D reconstruction of the membrane signal of the NSC lineage shown in (**A**). (**B**) Representative confocal pictures of the expression pattern of *worniu-GAL4* (*wor >*) along the dorso-ventral axis of the VNC, at the NSC level, at the neuron level, and at the axon level. *UAS-myr-mCherry* was driven under the control of *wor-GAL4*. Larvae are dissected after 72 h at 25˚C from ALH0. *n* = 8 VNCs. *worniu-GAL4* expression is visualised by the membrane staining of mCherry (magenta), NSCs are labelled with anti-Dpn (grey), GMC and newborn neurons with anti-Prospero (green), and neurons with

anti-ElaV (blue). Of note, we found that *wor-GAL4* does not drive in a few Type I NSCs in the VNC (negative for the mCherry signal, yellow arrows). (**C**) Representative confocal z-projections (maximal intensity) of the expression pattern of *worniu-GAL4* (*wor >*) in control and under RNAi knockdown of *mCherry*. *wor-GAL4, tub-GAL80^{ts}* was used to drive from ALH0 *UAS-myr-mCherry* combined with either *UAS-GFP RNAi* (control, *n* = 7 VNCs) or *UAS-mCherry RNAi* (mCherry knockdown, *n* = 8 VNCs). Larvae are dissected after 68 h at 29˚C from ALH0. *worniu-GAL4* expression is visualised by the membrane staining of mCherry (magenta), NSCs are labelled with anti-Dpn (grey), GMC and newborn neurons with anti-Prospero (green), and neurons with anti-ElaV (blue). ALH, after larval hatching; CG, cortex glia; GMC, ganglion mother cell; NSC, neural stem cell; VNC, ventral nerve cord. (TIF)

**S2 Fig. CG encase individual tumour NSC lineages.** (**A**) Two different Type I tumours, *aPKC* (*Nrv2::GFP, wor-GAL4; tub-GAL80^{ts} > aPKC^{CAAX}*) and *dpn* (*Nrv2::GFP, wor-GAL4; tub-GAL80^{ts} > dpn*), were induced from ALH0. *aPKC* larvae were dissected after 48 h at 18˚C, followed by 48–52 h at 29˚C. *dpn* larvae were dissected after 72 h at 29˚C. *aPKC* tumours, *n* = 6 VNCs and *dpn* tumours, *n* = 6 VNCs. The organisation of CG membrane was monitored by *Nrv2::GFP* (green). NSCs are labelled with anti-Dpn (grey). (**B**) Representative confocal images of the progressive formation of individual encasing around Type II lineages, at ALH0, ALH16, ALH24, ALH30, ALH48, ALH72, and ALH96 at 25˚C. *n* ≥ 8 CBs for all time points. CG membrane is visualised with *Nrv2::GFP* (green), NSCs are labelled with anti-Dpn (grey), Type II NSCs are recognised by the absence of anti-Ase (magenta), and GMCs are labelled with anti-Pros (blue). (**C**) Quantification of the number of CG chambers containing Type II-derived lineages, in control (*wor > raeppli-NLS*) and *brat RNAi* (*wor > raeppli-NLS, brat RNAi*), from Fig 2D. Briefly, Type II (*brat RNAi*) tumours were induced together with the multicolour lineage tracing Raeppli-NLS. One out of 4 colours (blue, white, orange, and red) is stochastically expressed in the transformed NSCs upon induction. Larvae were dissected after 48 h at 18˚C, followed by 2 h heatshock at 37˚C and 48–52 h at 29˚C. See S1 Table for detailed genetics, timing, and conditions of larval rearing. Control (*n* = 8 CB) and *brat RNAi* (*n* = 7 CB). Data statistics: Mann–Whitney U test. Results are presented as box and whisker plots. (**D**) Quantification of the number of colours per chambers containing Type II-derived lineages in control and *brat RNAi*, from (**C**). Several colours can be found in one CG chamber for control Type II and especially Type II *brat* tumours, while average chamber number did not change (see (**C**)). An explanation is the non-controlled activation of Raeppli-NLS in tumour NSCs/INPs in which the original heatshock-induced recombination failed. This would lead to the generation of individual tumour NSC/INP-specific lineages, marked differently in the same chamber yet still coming from the same mother NSC. Results are represented as pie charts. (**E, F**) Uncontrolled Raeppli-NLS inductions happen in *brat* RNAi tumours. (**E**) Representative confocal picture (one slice) of a *brat* RNAi (*wor > raeppli-NLS, brat RNAi*) CB without (left panel, No heatshock) or with (right panel, Heatshock) induction of Raeppli-NLS by heatshock. The induction of Raeppli-NLS is under the control of a flippase (*hs-FLP*) required for the stochastic selection and expression of one of the 4 fluorophores (visualised in blue, white, orange, and red) through genetic recombination. *hs-FLP* is itself activated by heatshock. Without heatshock, *hs-FLP* is present but should not be activated and consequently Raeppli-NLS not expressed. The detection of multiple cells with different colours, mostly sparse (yellow arrows) or in small clones (yellow stars), in the No heatshock condition shows that *hs-FLP* has been activated despite the lack of heatshock in an uncontrolled fashion. The same mechanism can happen in the heatshock condition, in *brat* RNAi Type II NSC lineages in which the heatshock-controlled induction had failed, resulting in multiple colours within one CG chamber.

In this condition, CG chambers containing multiple colours also contain unlabelled cells. *brat* tumours were induced after 48 h at 18˚C from ALH0 and dissected 48–52 h at 29˚C later. In the heatshock condition, a 2-h heatshock at 37˚C was performed at the time of tumour induction. CG membrane is visualised with *Nrv2::GFP* (green). See S1 Table for detailed genetics, timing, and conditions of larval rearing. (**F**) Quantification of the sum of the number of colours in all Type II NSC chambers per CB from (**E**). *n* (No heatshock) = 8 CBs and *n* (Heatshock) = 9 CBs. Within one CB, the maximal number of colours was recorded in each Type II chamber and their sum per CB was determined. While each CB contains only 8 Type II mothers NSCs, the total number of Raeppli-NLS-marked clones is greater, fitting with the hypothesis that they come from individual Dpn$^+$ NSC-like cells issued from the dysregulated proliferation of the original NSC. Data statistics: Mann–Whitney U test. Results are presented as box and whisker plots. For all box and whisker plots: whiskers mark the minimum and maximum, the box includes the 25th–75th percentile, and the line in the box is the median. Individual values are superimposed. The data underlying this figure's quantifications can be found in S1 Data.
(TIF)

**S3 Fig. CG use intrinsic and generic NSC lineage cues to individually encase NSC lineages.** (**A**) Representative confocal pictures of the extent of individual encasing of NSC lineages by CG at T1 both for Control and Delayed chamber formation conditions, following the regimen described in Fig 3A. Top panel shows the whole thoracic VNC, and bottom panel a close-up of the yellow box. NSC lineages were marked with the multicolour lineage tracing Raeppli-NLS (blue, white, orange, and red), induced at ALH0 using *hs-Flp* (*wor > raeppli-nls*). Larvae are dissected after 72 h at 29˚C (T1). See S1 Table for detailed genetics, timing, and conditions of larval rearing. CG membrane was visualised with *Nrv2::GFP* (green). Dashed white lines highlight examples of NSC lineages already encased individually, and dashed yellow lines outline zones where NSC lineages are still not individually encased. (**B**) Quantification of the percentage of NSC lineages non-individually encased from (**A**). Control T1 (*n* = 9 VNCs) and Delayed chamber formation T1 (*n* = 10 VNCs). Data statistics: generalised linear model (Binomial regression with a Bernoulli distribution). Results are presented as box and whisker plots. (**C**) Schematic of the experiment designed to probe whether specific adhesions exist between individual CG cells and individual NSC lineages. Mosaic clonal analysis is used to induce apoptosis in a few CG in a random fashion (magenta), preventing the encasing of corresponding NSC lineages. If specific adhesions exist between individual CG cells and individual NSC lineages, neighbouring CG cells would not be able to bind to and encase these lineages. (**D**) Representative confocal picture of the CG network in control CG clone and in clone where apoptosis has been induced earlier (*reaper* OE). The CoinFLP system was used to generate wild-type and *reaper* OE clones in the CG. Clones were induced at late embryogenesis/early larval stage through the expression of *cyp4g15-FLP*. Larvae were then maintained at 18˚C for 96 h to prevent *reaper* expression and then shifted to 29˚C for 48 h and dissected. See S1 Table for genetics, timing, and conditions of larval rearing. The membrane of the clone is marked by mCD8::RFP (magenta). CG membrane was visualised with *Nrv2::GFP* (green), and NSCs are labelled with anti-Dpn. (**E**) Quantification of the clone volume (μm3), of the number of NSCs per clone and of the number of non-encased NSCs around the clone between wild-type and *reaper* OE clones from (**D**). Clone volume strongly decreases in *reaper* OE clones compared to control, corresponding to *reaper*-expressing CG cells left as membrane remnants. In accordance, the number of NSCs per clone dropped to zero. Yet, all NSCs neighbouring the clones were individually encased. Control (*n* = 14 clones) and *reaper* OE (*n* = 11 clones). Data statistics: Mann–Whitney U test for clone volumes. For the number of NSCs per clone and for the

number of non-encased NSCs around the clone, there is no variance, and, thus, statistics cannot be applied (NA, non-applicable). Results are presented as box and whisker plots. For all box and whisker plots: whiskers mark the minimum and maximum, the box includes the 25th–75th percentile, and the line in the box is the median. Individual values are superimposed. The data underlying this figure's quantifications can be found in S1 Data. ALH, after larval hatching; CG, cortex glia; NSC, neural stem cell; OE, overexpression; VNC, ventral nerve cord.
(TIF)

**S4 Fig. Intralineage adherens junctions are not absolutely required for individual encasing.** (**A**) Representative confocal pictures of the expression of Shg in Type I (*wor > pros RNAi*) and Type II (*wor > brat RNAi*) tumours from Fig 2A and 2B. *pros* tumour (*n* = 10 VNCs) and *brat* tumour (*n* = 10 CBs). The organisation of CG membrane was monitored by *Nrv2::GFP* (green). Tumour NSCs are labelled with anti-Dpn (grey), and Shg is detected with a specific antibody (magenta). (**B**) Representative confocal image of the expression of Shg in the OL (ALH72 at 25˚C). *n* = 6 OLs. Shg is detected with a specific antibody (magenta), CG membrane is visualised with *Nrv2::GFP* (green), NSCs are labelled with Dpn (grey), and neurons are labelled with ElaV (blue). (**C**) Confocal images of mutant clones of *shg* (null *shg*$^{R64}$ allele) in NSC lineages, generated during late embryogenesis (top panel) or at ALH48 at 25˚C (bottom panel) by heatshock induction through *hs-FLP*. Clones are analysed at ALH72 at 25˚C. *n* (embryogenesis) = 59 clones for 11 VNCs and *n*(ALH48) = 11 clones for 8 VNCs. See S1 Table for detailed genetics, timing, and conditions of larval rearing. The membrane of the clone is marked by mCD8::GFP (magenta). CG membrane is visualised with *cyp4g15-mtd::Tomato* (green), NSCs are labelled with Dpn (grey), Shg is detected with a specific antibody (blue, induction during embryogenesis), and neurons are labelled with ElaV (blue, induction at ALH48). (**D**) Representative confocal pictures of the loss of signal for Shg (monitored through Shg::GFP, grey) in different genetic conditions defined by the driver line (CG, *cyp4g15-GAL4*, *cyp >*; NSC lineages, *wor-GAL4*, *wor >*) and RNAi constructs. Larvae are dissected after 68 h at 29˚C from ALH0. (**E**) Quantification of the efficiency of *shg* knockdown in NSC lineages for different RNAi lines from (**D**). Shg levels are monitored by Shg::GFP. See Methods for details of the quantification. Control (*shg*::GFP/+) (*n* = 13 VNCs), *wor > shg RNAi*$^{VDRC27082}$ (*n* = 8 VNCs), and *wor > shg RNAi*$^{VDRC103962}$ (*n* = 8 VNCs). Data statistics: Kruskal–Wallis H test with Dunn's multiple comparisons test. *p* < 0.0001 for the Kruskal–Wallis H test on grouped dataset. *P* values from Dunn's multiple comparisons test are displayed on the graph. Results are presented as box and whisker plots, where whiskers mark the minimum and maximum, the box includes the 25th–75th percentile, and the line in the box is the median. Individual values are superimposed. (**F**) Representative confocal pictures of the thoracic VNC for *shg* knockdown by RNAi (*cyp > shg RNAi*, VDRC line 103962) in the CG (driver line *Nrv2::GFP, tub-GAL80*$^{ts}$; *cyp4g15-GAL4*), at the NSC level. *n* = 10 VNCs. Larvae are dissected after 68 h at 29˚C from ALH0. CG membrane is visualised by *Nrv2::GFP* (green), and NSCs are labelled with anti-Dpn (grey). The data underlying this figure's quantifications can be found in S1 Data. ALH, after larval hatching; CG, cortex glia; NSC, neural stem cell; OL, optic lobe; Shg, Shotgun; VNC, ventral nerve cord.
(TIF)

**S5 Fig. Intralineage adherens junctions are not absolutely required for individual encasing yet might provide robustness.** (**A**) Close-ups of confocal pictures assessing Shg levels in control, *shg* knockdown (VDRC line 27082) in NSC lineages (Fig 4E) and *shg* knockdown in NSC lineages plus delayed chamber formation (Fig 5B) at T2 (100 h at 29˚C). *n* ≥ 5 for all conditions. Shg levels are monitored with a specific antibody (magenta), NSC lineages are marked

with the multicolour lineage tracing Raeppli-NLS (*wor > raeppli-NLS*; blue, white, orange, and red), and CG membrane is visualised with *Nrv2::GFP* (green). See S1 Table for detailed genetics, timing, and conditions of larval rearing. (**B**) Representative confocal pictures of the extent of individual encasing of NSC lineages by the CG at T1 (72 h at 29˚C), following the regimen described in Fig 5A. Top panel shows the whole thoracic VNC, and bottom panel a close-up of the yellow box. NSC lineages were marked with the multicolour lineage tracing Raeppli-NLS (blue, white, orange, and red), induced at ALH0 using *hs-Flp* (*wor > raeppli-NLS*). See S1 Table for detailed genetics, timing, and conditions of larval rearing. CG membrane was visualised with *Nrv2::GFP* (green). Dashed white lines highlight examples of NSC lineages already encased individually, and dashed yellow lines outline zones where NSC lineages are still not individually encased. (**C**) Quantification of the percentage of NSC lineages non-individually encased at T1 in Control, in *shg* knockdown (VDRC line 27082) in NSC lineages, and in *shg* knockdown in NSC lineages plus delayed chamber formation. Control T1 (*n* = 9 VNCs), Delayed chamber formation T1 (*n* = 10 VNCs), and Delayed chamber formation + *shg* RNAi in NSC lineages T1 (*n* = 6 VNCs). Data statistics: generalised linear model (Binomial regression with a Bernoulli distribution). $p = 2.59 \times 10^{-18}$ for the grouped dataset. *P* values for individual comparisons test are displayed on the graph. Results are presented as box and whisker plots, where whiskers mark the minimum and maximum, the box includes the 25th–75th percentile, and the line in the box is the median. Individual values are superimposed. The data underlying this figure's quantifications can be found in S1 Data. ALH, after larval hatching; CG, cortex glia; NSC, neural stem cell; Shg, Shotgun; VNC, ventral nerve cord (TIF)

**S6 Fig. Nrx-IV is specifically required in NSC lineages for their individual encasing by the cortex glia.** (**A**) Representative confocal images of the expression of the septate junction components Dlg1, ATPα, Cora, Nrx-IV, and Nrg at ALH72 at 25˚C. $n \geq 6$ for all components. Dlg1 is monitored through a Dlg1::GFP fusion, ATPα through an ATPα::GFP fusion, Nrx-IV through a Nrx-IV::GFP fusion, Nrg through a Nrg::GFP fusion, and Cora is detected by a specific antibody (all magenta). CG membrane is visualised by *cyp4g15-mtd::Tomato* (green), NSCs are labelled with anti-Dpn (grey), and neurons are labelled with anti-ElaV (blue). (**B**) Representative confocal pictures of the thoracic VNC for control (*wor >—(x w^1118)*) and for *nrx-IV* knockdown by RNAi (*wor > nrx-IV RNAi*, BDSC line 324242) in NSC lineages (driver line *Nrv2::GFP, wor-GAL4; tub-GAL80^ts*). Larvae are dissected after 68 h at 29˚C from ALH0. Nrx-IV levels are monitored by a specific antibody (magenta), and CG membrane is visualised by *Nrv2::GFP* (green). (**C**) Quantification of the efficiency of *nrx-IV* knockdown in NSC lineages by RNAi from (**B**). Nrx-IV levels are monitored by a specific antibody. See Methods for details of the quantification. *wor >—(x w^1118)* (*n* = 6 VNCs) and *wor > nrx-IV RNAi^BDSC32424* (*n* = 7 VNCs). Data statistics: Mann–Whitney U test. Results are presented as box and whisker plots, where whiskers mark the minimum and maximum, the box includes the 25th–75th percentile, and the line in the box is the median. Individual values are superimposed. (**D**) Representative confocal pictures of the thoracic VNC for *nrx-IV* knockdown by RNAi (*wor > nrx-IV RNAi*, VDRC line GD9039) in NSC lineages (driver line *Nrv2::GFP, wor-GAL4; tub-GAL80^ts*). *n* = 9 VNCs. Larvae are dissected after 68 h at 29˚C from ALH0. CG membrane is visualised by *Nrv2::GFP* (green) and NSCs are labelled with anti-Dpn (grey). (**E**) Representative confocal pictures of thoracic VNCs for *nrx-IV* knockdown by RNAi (BDSC line 324242) in all neurons (*ElaV-GAL4* driver, *n* = 7 VNCs), in mature neurons (*nSyb-GAL4* driver, *n* = 7 VNCs), and in the CG (*cyp4g15-GAL4* driver, *n* = 10 VNCs). Larvae are dissected after 68 h at 29˚C from ALH0. CG membrane is visualised by *Nrv2::GFP* (green), and NSCs are labelled with anti-Dpn (grey). The data underlying this figure's quantifications can be found in

S1 Data. ALH, after larval hatching; ATPα, Na K-ATPase pump; CG, cortex glia; Cora, Cora-cle; Dlg1, Discs Large; Nrg, Neuroglian; Nrx-IV, Neurexin-IV; NSC, neural stem cell; Shg, Shotgun; VNC, ventral nerve cord.
(TIF)

**S7 Fig. A CG to NSC lineage interaction through Nrx-IV and Wrapper is required for individual encasing of NSC lineages by the CG.** (**A**) Representative confocal pictures at the NSC and neuropile levels of the extent of loss of signal for *wrapper* (monitored through the detection by RNA FISH of *wrapper* mRNA, magenta) under *wrapper* knockdown (*cyp > wrapper RNA$^{BDSC29561}$*) in CG (driver line *Nrv2::GFP, tub-GAL80$^{ts}$; cyp4g15-GAL4*) compared to control (*cyp >—(x w$^{1118}$)*). Larvae are dissected after 68 h at 29˚C from ALH0. White arrows indicate *wrapper* mRNA signal coming from the midline glia. (**B**) Quantification of the efficiency of *wrapper* knockdown in NSC lineages by RNAi from (**A**). *wrapper* levels are monitored through RNA FISH. See Methods for details of the quantification. *cyp >—(x w$^{1118}$)* (*n* = 8 VNCs) and *cyp > wrapper RNAi$^{BDSC29561}$* (*n* = 8 VNCs). Data statistics: unpaired Student *t* test. Results are presented as box and whisker plots. (**C**) Representative confocal pictures of the thoracic VNC for *wrapper* knockdown by RNAi (*cyp > wrapper RNA$^{iVDRC105314}$*) in the CG (driver line *Nrv2::GFP, tub-GAL80$^{ts}$; cyp4g15-GAL4*). Larvae are dissected after 68 h at 29˚C from ALH0. *n* = 8 VNCs. CG membrane is visualised by *Nrv2::GFP* (green), and NSCs are labelled with anti-Dpn (grey). (**D**) Representative confocal picture of the expression pattern of combined NSC lineages and CG drivers (*wor + cyp*) at the NSC level. *UAS-mCD8::GFP* and *UAS-Histone2B::RFP* were both driven under the control of *Nrv2::GFP, worniu-GAL4; cyp4g15-GAL4, tub-GAL80$^{ts}$*. Larvae are dissected after 68 h at 29˚C from ALH0. *n* = 6 VNCs. *GAL4* expression is visualised by the membrane staining of mCD8::GFP (magenta) and nuclear staining of Histone2B::RFP (magenta). NSCs are labelled with anti-Dpn (grey) and glia with anti-Repo (blue). Expression in the CG is detectable by the stereotypic membrane pattern, and co-localisation between Histone2B::RFP and Repo (white arrows). Expression in NSC lineages is detectable by the co-localisation between Histone2B::RFP and Dpn (yellow arrows) and the accumulation of RFP$^+$ progeny nuclei. (**E**) Quantification of the CG membrane volume per NSC for *shg* knockdown and *nrx-IV* knockdown compared to a control condition See Methods for details. *wor >—(x w$^{1118}$)* (*n* = 10 VNCs), *wor > shg$^{VDRCD27082}$* RNAi (*n* = 10 VNCs), *wor > nrx-IV$^{VDRC9039}$* RNAi (*n* = 10 VNCs). Data statistics: one-way ANOVA with Tukey's multiple comparisons test. *p* < 0.0001 for the one-way ANOVA test on grouped dataset. *P* values from Tukey's multiple comparisons test are displayed on the graph. Results are presented as box and whisker plots. (**F**) Representative confocal picture of the localisation of Wrapper in a thoracic VNC using a specific antibody (magenta). Larvae were dissected at ALH72 at 25˚C. *n* = 4 VNCs. CG membrane is visualised by *Nrv2::GFP* (green). For all box and whisker plots: whiskers mark the minimum and maximum, the box includes the 25th–75th percentile, and the line in the box is the median. Individual values are superimposed. The data underlying this figure's quantifications can be found in S1 Data. ALH, after larval hatching; CG, cortex glia; FISH, fluorescent in situ hybridization; Nrx-IV, Neurexin-IV; NSC, neural stem cell; VNC, ventral nerve cord.
(TIF)

**S8 Fig. Nrg is specifically required in NSC lineages for individual encasing and CG network integrity.** (**A**) Representative confocal pictures of the thoracic VNC for control (*wor >—(x w$^{1118}$)*) and for *nrg* knockdown by RNAi (*wor > nrg RNAi$^{BDSC37496}$*) in NSC lineages (driver line *Nrg::GFP; cyp4g15-mtd::Tomato, wor-GAL4; tub-GAL80$^{ts}$*). Larvae are kept 24 h at 18˚C, then dissected after 54 h at 29˚C. Nrg levels are monitored through *Nrg::GFP* (magenta) and NSCs are labelled with Dpn (grey). (**B**) Quantification of the efficiency of *nrg* knockdown

in NSC lineages by RNAi from (**A**). Nrg levels are monitored through *Nrg*::*GFP*. See Methods for details of the quantification. *wor* >—(x *w*$^{1118}$) (*n* = 9 VNCs) and *wor* > *nrg RNAi*$^{BDSC37496}$ (*n* = 9 VNCs). Data statistics: unpaired Student *t* test. Results are presented as box and whisker plots, where whiskers mark the minimum and maximum, the box includes the 25th–75th percentile, and the line in the box is the median. Individual values are superimposed. (**C**) Representative confocal pictures of thoracic VNCs for *nrg* knockdown (BDSC line 37496) by RNAi in all neurons (*ElaV-GAL4* driver, *n* = 8 VNCs), in mature neurons (*nSyb-GAL4* driver, *n* = 7 VNCs), and in the CG (*cyp4g15-GAL4* driver, *n* = 9 VNCs). For *ElaV* and *nSyb*, larvae are kept 24 h at 18˚C then dissected after 54 h at 29˚C. For *cyp*, larvae are dissected after 68 h at 29˚C from ALH0. CG membrane is visualised by *Nrv2*::*GFP* (green), and NSCs are labelled with anti-Dpn (grey). (**D**) Confocal close-up pictures of a condition in which *nrg* is knocked down by RNAi (BDSC line 37496) in NSC lineages marked with the multicolour lineage tracing Raeppli-NLS (blue, white, orange, and red; see Fig 8D and 8E). Raeppli-NLS is induced at ALH0 using *hs-Flp*, and RNAi after 24 h at 18˚C. Larvae are dissected 54 h after RNAi induction. CG membrane was visualised with *Nrv2*::*GFP* (green). See S1 Table for detailed genetics, timing, and conditions of larval rearing. The dashed yellow lines indicate a chamber in which only secondary neurons (blue) are found. The data underlying this figure's quantifications can be found in S1 Data. ALH, after larval hatching; CG, cortex glia; Nrg, Neuroglian; NSC, neural stem cell; VNC, ventral nerve cord.

(TIF)

**S9 Fig. Individual encasing of NSC lineages relies on strong intralineage adhesion through homophilic Nrg interaction.** (**A**) Representative confocal pictures of the expression of all Nrg isoforms (*Nrg*::*GFP*, magenta) compared to the Nrg$^{180}$ isoform (BP104, light blue) at ALH72 at 25˚C. *n* = 8 VNCs. Upper panel, top view. Lower panel, median cut though the NSC population. Note the absence of signal at the septate junctions (yellow arrowheads on the top view, magenta signal) for Nrg$^{180}$. (**B**) Representative confocal picture of the respective localisations of the Nrg$^{167}$ and Nrg$^{180}$ isoforms in a thoracic VNC, at ALH72 at 25˚C. *n* = 6 VNCs. The Nrg$^{167}$ isoform is visualised by a protein trap in the *nrg* gene leading to the preferential expression of this isoform (Nrg$^{167}$::GFP, yellow). The Nrg$^{180}$ isoform is detected with a specific antibody (BP104, light blue). (**C**) Representative confocal pictures and close-ups of thoracic VNCs for control (*wor* >—(x *w*$^{1118}$)) and for *nrg* knockdown by RNAi (*wor* > *nrg RNAi*$^{BDSC37496}$) in NSC lineages (driver line *Nrv2*::*GFP*, *wor-GAL4; tub-GAL80*$^{ts}$). Larvae are kept 24 h at 18˚C and then dissected after 54 h at 29˚C. Levels of the Nrg$^{180}$ isoform are monitored through staining with BP104 (light blue). CG membrane is visualised by *Nrv2*::*GFP* (green), and NSCs are labelled with anti-Dpn (grey). (**D**) Quantification of the efficiency of the knockdown of the Nrg$^{180}$ isoform (BP104 antibody) in NSC lineages by RNAi (BDSC line 37496) from (**C**). See Methods for details of the quantification. *wor* >—(x *w*$^{1118}$) (*n* = 8 VNCs) and *wor* > *nrg RNAi*$^{BDSC37496}$ (*n* = 10 VNCs). Data statistics: unpaired Student *t* test. Results are presented as box and whisker plots. (**E**) Representative confocal pictures and close-ups of thoracic VNCs for control (*wor* >—(x *w*$^{1118}$)) and for *nrg* knockdown by RNAi (*wor* > *nrg RNAi*$^{BDSC37496}$) in NSC lineages (driver line *Nrv2*::*GFP*, *wor-GAL4; tub-GAL80*$^{ts}$). Larvae are kept 24 h at 18˚C and then dissected after 54 h at 29˚C. Levels of the Nrg$^{167}$ isoform are monitored through *Nrg*$^{167}$::*GFP* (yellow). NSCs are labelled with anti-Dpn (grey) and neurons with anti-ElaV (blue). (**F**) Quantification of the efficiency of the knockdown of the Nrg$^{167}$ isoform (*Nrg*$^{167}$::*GFP*) in NSC lineages by RNAi from (**E**). See Methods for details of the quantification. *wor* >—(x *w*$^{1118}$) (*n* = 10 VNCs) and *wor* > *nrg RNAi*$^{BDSC37496}$ (*n* = 9 VNCs). Data statistics: unpaired Student *t* test. Results are presented as box and whisker plots. For all box and whisker plots: whiskers mark the minimum and maximum, the box includes the 25th–75th percentile,

and the line in the box is the median. Individual values are superimposed. The data underlying this figure's quantifications can be found in S1 Data. ALH, after larval hatching; CG, cortex glia; Nrg, Neuroglian; NSC, neural stem cell; VNC, ventral nerve cord.
(TIF)

**S10 Fig. Loss of Nrx-IV and Nrg adhesions in NSC lineages during development does not alter NSC survival and proliferation.** (**A**) Quantification of NSC number in the VNC for *shg* (line VDRC27082) knockdown and *nrx-IV* (line BDSC32424) knockdown in NSC lineages compared to control (driver line *Nrv2::GFP, wor-GAL4; tub-GAL80$^{ts}$*). Larvae are dissected after 68 h at 29˚C from ALH0. *wor >—(x w$^{1118}$)* (*n* = 10 VNCs), *wor > shg* RNAI$^{VDRC27082}$ (*n* = 10 VNCs), and *wor > nrx-IV* RNAI$^{BDSC32424}$ (*n* = 10 VNCs). Data statistics: one-way ANOVA with Tukey's multiple comparisons test. *p* = 0.8257 for the one-way ANOVA test on grouped dataset. *P* values from Tukey's multiple comparisons test are displayed on the graph. Results are presented as box and whisker plots. (**B**) Quantification of NSC number in the VNC for *nrg* (line BDSC37496) knockdown in NSC lineages compared to control (driver line *Nrv2:: GFP, wor-GAL4; tub-GAL80$^{ts}$*). Larvae are dissected after 68 h at 29˚C from ALH0. *wor >—(x w$^{1118}$)* (*n* = 10 VNCs) and *wor > nrg* RNAI$^{BDSC37496}$ (*n* = 10 VNCs). Data statistics: unpaired Student *t* test. Results are presented as box and whisker plots. (**C**) Mitotic index (left panel) and distribution of the mitotic phases (right panel) for control (*wor >—(x w$^{1118}$)*) and *nrx-IV* knockdown (*wor > nrx-IV RNAi*, line BDSC32424) in NSC lineages (driver line *Nrv2::GFP, wor-GAL4; tub-GAL80$^{ts}$*). Larvae are dissected after 68 h at 29˚C from ALH0. *wor >—(x w$^{1118}$)* (*n* = 10 VNCs) and *wor > nrx-IV* RNAI$^{BDSC32424}$ (*n* = 9 VNCs). Data statistics: unpaired Student *t* test for mitotic index and two-way ANOVA with a Šidak's multiple comparison test for mitotic phases. For the mitotic index, results are presented as box and whisker plots. For mitotic phases, stacked bars represent the respective percentage between aligned, misaligned, and undetermined spindles. Bars represent the SEM. There is no significant difference for any of the 4 phases. (**D**) Mitotic index (left panel) and distribution of the mitotic phases (right panel) for control (*wor >—(x w$^{1118}$)*) and *nrg* knockdown (*wor > nrg RNAi*, line BDSC37496) in NSC lineages (driver line *Nrv2::GFP, wor-GAL4; tub-GAL80$^{ts}$*). Larvae are kept 24 h at 18˚C and then dissected after 54 h at 29˚C. *wor >—(x w$^{1118}$)* (*n* = 8 VNCs) and *wor > nrg* RNAI$^{BDSC37496}$ (*n* = 10 VNCs). Data statistics: unpaired Student *t* test for mitotic index and two-way ANOVA with a Šidak's multiple comparisons test for mitotic phases. For the mitotic index, results are presented as box and whisker plots. For mitotic phases, stacked bars represent the respective percentage between aligned, misaligned, and undetermined spindles. Bars represent the SEM. There is no significant difference for any of the 4 phases. (**E**) Mitotic index (left panel) and distribution of the mitotic phases (right panel) for control (*wor >—(x w$^{1118}$)*) and *shg* knockdown (*wor > shg RNAi*, line VDRC27082) in NSC lineages (driver line *Nrv2:: GFP, wor-GAL4; tub-GAL80$^{ts}$*). Larvae are dissected after 68 h at 29˚C from ALH0. *wor >—(x w$^{1118}$)* (*n* = 10 VNCs) and *wor > shg* RNAI$^{VDRC27082}$ (*n* = 7 VNCs). Data statistics: unpaired Student *t* test for mitotic index and two-way ANOVA with a Šidak's multiple comparison test for mitotic phases. For the mitotic index, results are presented as box and whisker plots. For mitotic phases, stacked bars represent the respective percentage between aligned, misaligned, and undetermined spindles. Bars represent the SEM. There is no significant difference for any of the 4 phases. (**F**) Representative confocal pictures of the localisation of the asymmetric factor aPKC during metaphase in NSCs for *shg* (line VDRC27082), *nrx-IV* (line VDRC9039), and *nrg* (line BDSC37496) knockdown in NSC lineages (driver line *Nrv2::GFP, wor-GAL4; tub-GAL80$^{ts}$*) compared to control. Larvae are dissected after 68 h at 29˚C from ALH0 for *shg* knockdown, *nrx-IV* knockdown, and corresponding control. For *nrg* knockdown and corresponding control, larvae are kept 24 h at 18˚C and then dissected after 54 h at 29˚C. Only the

control for one of the 2 induction regimens is represented. CG membrane is visualised by *Nrv2::GFP* (green), and mitotic spindles by an anti-α-tubulin staining (yellow). aPKC is detected with a specific antibody (magenta), and neurons are labelled with anti-ElaV (light blue). (**G**) Quantification of the distribution between polarised and non-polarised aPKC staining from (**F**) for *shg*, *nrx-IV*, and *nrg* knockdown in NSC lineages compared to control. *wor* >—(*x w*[1118]) (*n* = 196 metaphasic NSCs), *wor* > *shg* RNAI[VDRC27082] (*n* = 144 metaphasic NSCs), *wor* > *nrx-IV* RNAI[VDRC9039] (*n* = 119 metaphasic NSCs), and *wor* > *nrg* RNAI[BDSC37496] (*n* = 136 metaphasic NSCs). Results are presented as pie charts. (**H**) Quantification of the percentage between aligned, misaligned, and undetermined mitotic spindles (in respect to aPKC polarisation) per VNC for *shg*, *nrx-IV*, and *nrg* knockdown in NSC lineages compared to control. *wor* >—(*x w*[1118]) (*n* = 10 VNCs), *wor* > *shg* RNAI[VDRC27082] (*n* = 10 VNCs), *wor* > *nrx-IV* RNAI[VDRC9039] (*n* = 10 VNCs), and *wor* > *nrg* RNAI[BDSC37496] (*n* = 9 VNCs). Data statistics: Two-way ANOVA with a Dunnett's multiple comparison test. Stacked bars represent the respective percentage between aligned, misaligned, and undetermined spindles. Bars represent the SEM. There is no significant difference for any of the 3 classes. (**I**) Representative confocal maximal z-projections of thoracic VNCs stained with Dcp-1 (white) to visualise apoptotic cells for *shg* (line VDRC27082) and *nrx-IV* (line VDRC9039) knockdown in NSC lineages (driver line *Nrv2::GFP, wor-GAL4; tub-GAL80*[ts]) compared to control. Larvae are dissected after 68 h at 29˚C from ALH0. (**J**) Quantification of the total number of Dcp-1[+] cells per VNC from (**I**). *wor* >—(*x w*[1118]) (*n* = 11 VNCs), *wor* > *shg* RNAI[VDRC27082] (*n* = 11 VNCs), and *wor* > *nrx-IV* RNAI[VDRC9039] (*n* = 11 VNCs). Data statistics: one-way ANOVA with Tukey's multiple comparison test. *p* = 0.9501 for the one-way ANOVA test on grouped dataset. *P* values from Tukey's multiple comparisons test are displayed on the graph. Results are presented as box and whisker plots. (**K**) Representative confocal maximal z-projections of thoracic VNCs stained with Dcp-1 (white) to visualise apoptotic cells, for *nrg* knockdown *shg* (line BDSC37496) in NSC lineages (driver line *Nrv2::GFP, wor-GAL4; tub-GAL80*[ts]) compared to control. Larvae are kept 24 h at 18˚C and then dissected after 54 h at 29˚C. (**L**) Quantification of the total number of Dcp-1[+] cells per VNC from (**K**). *wor* >—(*x w*[1118]) (*n* = 11 VNCs) and *wor* > *nrg* RNAI[BDSC37496] (*n* = 11 VNCs). Data statistics: Mann–Whitney U test. Results are presented as box and whisker plots. For all box and whisker plots: whiskers mark the minimum and maximum, the box includes the 25th–75th percentile, and the line in the box is the median. Individual values are superimposed. The data underlying this figure's quantifications can be found in S1 Data. ALH, after larval hatching; Dcp-1, *Drosophila* cleaved caspase 1; Nrg, Neuroglian; Nrx-IV, Neurexin-IV; NSC, neural stem cell; SEM, Standard error of the mean; VNC, ventral nerve cord.
(TIF)

**S11 Fig. Loss of Nrx-IV and Nrg adhesions in NSC lineages during development results in locomotor hyperactivity in the adults.** (**A**, **B**) For all metrics. Control (*wor-GAL4, tub-Gal80*[ts] *x w*[1118]), *n* = 35 non-induced adult males and *n* = 55 induced adult males. *shg* RNAi (*wor-GAL4, tub-Gal80*[ts] *x shg RNAi*[VDRC27082]), *n* = 35 non-induced adult males and *n* = 55 induced adult males. *nrx-IV* RNAi (*wor-GAL4, tub-Gal80*[ts] *x nrx-IV RNAi*[BDSC32424]), *n* = 35 non-induced adult males and *n* = 55 induced adult males. *nrg* RNAi (*wor-GAL4, tub-Gal80*[ts] *x nrg RNAI*[BDSC37496]), *n* = 35 non-induced adult males and *n* = 55 induced adult males. (**A**) Percentage of global time moving during the Dark period (% moving, ratio between total moving time and total time) in non-induced (flies always kept at 18˚C before the recordings) and induced (flies shifted to 29˚C from early larval stage to early pupal stage) conditions. Data statistics: Kruskal–Wallis H test with Dunn's multiple comparisons test for both non-induced and induced conditions. *p* (non-induced) < 0.0001 and *p* (induced) < 0.0001 for the Kruskal–

Wallis H test on grouped dataset. *P* values from Dunn's multiple comparisons test are displayed on the graph. Results are presented as box and whisker plots. (**B**) Percentage of global time moving during the Light period (% moving, ratio between total moving time and total time) in non-induced (flies always kept at 18˚C before the recordings) and induced (flies shifted to 29˚C from early larval stage to early pupal stage) conditions. Data statistics: Kruskal–Wallis H test with Dunn's multiple comparisons test for both non-induced and induced conditions. *p* (non-induced) = 0.0588 and *p* (induced) < 0.0001 for the Kruskal–Wallis H test on grouped dataset. *P* values from Dunn's multiple comparisons test are displayed on the graph. Results are presented as box and whisker plots. For all box and whisker plots: whiskers mark the minimum and maximum, the box includes the 25th–75th percentile, and the line in the box is the median. Individual values are superimposed. The data underlying this figure's quantifications can be found in S1 Data. Nrg, Neuroglian; Nrx-IV, Neurexin-IV; NSC, neural stem cell.
(TIF)

**S12 Fig. Loss of Wrapper adhesions in the CG during development results in locomotor hyperactivity in the adults.** (**A**) Schematics of the rearing regimen for non-induced and induced conditions. In non-induced conditions, animals are constantly kept at 18˚C before the recordings, a temperature allowing the repression of the GAL4/UAS system by the thermosensitive GAL80$^{ts}$ and thus blocking expression of the RNAi under the control of the CG driver (*tub-Gal80$^{ts}$; cyp4g15-GAL4*). In induced conditions, the animals are transiently shifted to 29˚C from the early larval stage to the early pupal stage, a temperature allowing the expression of GAL4/UAS system and thus of the RNAi. Animals were kept at 18˚C both during the embryonic development and from the early pupal stage. Adult flies were assessed 7–10 days after eclosion. See S1 Table for detailed genetics, timing, and conditions of larval rearing. (**B-D**) Control, (*tub-Gal80$^{ts}$; cyp4g15-GAL4 x w$^{1118}$*), *n* = 20 non-induced adult males and *n* = 20 induced adult males. *wrapper* RNAi (*tub-Gal80$^{ts}$; cyp4g15-GAL4 x wrapper* RNAi$^{BDSC29561}$), *n* = 20 non-induced adult males and *n* = 20 induced adult males. (**B**) Plot representing the percentage of global time moving (measured as the fraction of time moving within 30-min intervals), in non-induced and induced conditions. (**C**) Fraction (%) of the time moving across LD cycles (% moving, ratio between total moving time and total time) in non-induced and induced conditions. Data statistics: unpaired Student *t* test for non-induced condition and Mann–Whitney U test for induced condition. Results are presented as box and whisker plots. (**D**) Mean velocity (in relative units) across LD cycles in non-induced and induced conditions. Data statistics: unpaired Student *t* test for both non-induced and induced conditions. Results are presented as box and whisker plots. For all box and whisker plots: whiskers mark the minimum and maximum, the box includes the 25th–75th percentile, and the line in the box is the median. Individual values are superimposed. The data underlying this figure's quantifications can be found in S1 Data.
(TIF)

**S1 Table. Genotypes, crosses, and regimens of fly culture.**
(PDF)

**S2 Table. Phenotypic screen of knocking down components of occluding junction on the cortex glia architecture.** RNAis against selected components of *Drosophila* occluding junctions were individually driven in the CG (driver line *Nrv2::GFP, tub-GAL80$^{ts}$; cyp4g15-GAL4*), and larvae were dissected after 68 h at 29˚C from ALH0. The CG membrane, labelled by Nrv2::GFP, was qualitatively assessed for alteration of normal overall morphology and

individual encasing ($n \geq 7$ VNCs). Change in survival was also recorded.
(TIF)

**S1 Data. Source data for all quantifications.**
(XLSX)

## Acknowledgments

We thank Y. Bellaïche, A. Brand, Y-N Jan, M. Landgraf, R. Le Borgne, C. Klämbt, and C. Maurange for reagents and strains. L. Arbogast generated the *cyp4g15-FRT-STOP-FRT-GAL4*, *grh^{D4}-FLP* and QUAS-PTEN constructs. We are grateful to F. Rouyer for helpful discussion about behavioural experiments. We thank R. Levayer, F. Schweisguth, and F. Rouyer for constructive comments on the paper. Stocks obtained from the Bloomington Drosophila Stock Center (NIH P40OD018537) and from the Vienna Drosophila Resource Center were used in this study. Antibodies were obtained from the Developmental Studies Hybridoma Bank, created by the NICHD of the NIH, and maintained at The University of Iowa.

## Author Contributions

**Conceptualization:** Agata Banach-Latapy, Vincent Rincheval, Pauline Spéder.

**Data curation:** Agata Banach-Latapy, Vincent Rincheval, Pauline Spéder.

**Formal analysis:** Agata Banach-Latapy, Vincent Rincheval, Pauline Spéder.

**Funding acquisition:** Isabelle Guénal, Pauline Spéder.

**Investigation:** Agata Banach-Latapy, Vincent Rincheval, David Briand, Pauline Spéder.

**Methodology:** Agata Banach-Latapy, Vincent Rincheval, David Briand, Pauline Spéder.

**Project administration:** Pauline Spéder.

**Resources:** Isabelle Guénal, Pauline Spéder.

**Supervision:** Isabelle Guénal, Pauline Spéder.

**Validation:** Agata Banach-Latapy, Vincent Rincheval, David Briand, Pauline Spéder.

**Visualization:** Agata Banach-Latapy, Vincent Rincheval, David Briand, Pauline Spéder.

**Writing – original draft:** Agata Banach-Latapy, Vincent Rincheval, Pauline Spéder.

**Writing – review & editing:** Agata Banach-Latapy, Vincent Rincheval, Pauline Spéder.

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
