## [Editor Report · Decision Letter 0]

26 Jul 2023

Dear Dr Spéder, 

Thank you for submitting your revised manuscript entitled "Differential adhesion during development establishes individual neural stem cell niches and shapes adult behaviour" for consideration as a Research Article by PLOS Biology.

Your revision has now been evaluated by the PLOS Biology editorial staff as well as by the academic editor, and I am writing to let you know that we would like to send your manuscript back to the original reviewers.

Once your full submission is complete, your paper will undergo a series of checks in preparation for peer review. After your manuscript has passed the checks it will be sent out for review. To provide the metadata for your submission, please Login to Editorial Manager (https://www.editorialmanager.com/pbiology) within two working days, i.e. by Jul 28 2023 11:59PM.

Kind regards,

Luke

Lucas Smith, Ph.D.

Senior Editor

PLOS Biology

lsmith@plos.org

---

## [Decision Letter · Decision Letter 1]

25 Aug 2023

Dear Dr Spéder,

Thank you for your patience while we considered your revised manuscript "Differential adhesion during development establishes individual neural stem cell niches and shapes adult behaviour" for consideration as a Research Article at PLOS Biology. Your revised study has now been evaluated by the PLOS Biology editors, the Academic Editor and by the original Reviewers 1 and 3. 

As you will see in their comments, appended below, Reviewer 1 is completely satisfied, and Reviewer 3 agrees the manuscript has addressed many of the previous issues raised in the last round of review. However, Reviewer 3 has highlighted several lingering concerns and provided a number of specific suggestions that we think should be adopted, to further improve the paper. As a note, in his/her point 3, Reviewer 3 continues to have concerns with Supp Figure 7D - and after discussing this with the Academic Editor, we agree that without providing additional data, this figure panel and associated claims should be removed. 

In light of the reviews, we are pleased to offer you the opportunity to address the remaining points from the reviewer 3 in a revision that we anticipate should not take you very long. We will then assess your revised manuscript and your response to the reviewers' comments with our Academic Editor aiming to avoid further rounds of peer-review, although might need to consult with the reviewers, depending on the nature of the revisions.

**IMPORTANT: As you address these last reviewer requests, we also ask that you attend to the following editorial requests: 

1) BLURB: In the relevant section of our online system, please provide a blurb which (if accepted) will be included in our weekly and monthly Electronic Table of Contents, sent out to readers of PLOS Biology, and may be used to promote your article in social media. The blurb should be about 30-40 words long and is subject to editorial changes. It should, without exaggeration, entice people to read your manuscript. It should not be redundant with the title and should not contain acronyms or abbreviations.

2) DATA: Thank you providing the underlying data used to generate the graphs in your figures. Can you please add a note to each relevant figure legend (including supplemental) that references this data? For example, you can add the sentence "the data underlying this figure can be found in ___" and reference the relevant files. 

3) Per journal policy, any code that you have generated to support the conclusions of your manuscript should be made available without restrictions upon publication. If you used or generated code for this study, please ensure that the code is sufficiently well documented and reusable, and that your Data Statement in the Editorial Manager submission system accurately describes where your code can be found.

**IMPORTANT - SUBMITTING YOUR REVISION**

*Resubmission Checklist*

*Published Peer Review*

*PLOS Data Policy*

*Blot and Gel Data Policy*

Sincerely,

Luke

Lucas Smith, Ph.D.

Senior Editor

PLOS Biology

lsmith@plos.org

REVIEWS:

Reviewer #1: The authors addressed all our concerns and suggestions in the revised manuscript. We support publication of this study.

Reviewer #3: The authors have made a considerable effort to improve the paper and have addressed several of my criticisms. There are many more quantifications, and the FISH and RNAi evidence that wrapper is expressed in CG are now compelling. However, other aspects have not been dealt with and will require improvement. The work is interesting and would have broad appeal for the PLoS Biology readership, the methodology appears to be rigorous and the microscopy is beautiful, but the interpretation of findings, the subjective narrative and conclusions lack rigour in places, and presentation of the evidence in support of conclusions needs to be improved. The following revisions are essential to support the current conclusions.

1. Text, genotypes, poor labelling, etc: some limited improvements have been made, but text and many figures still lack essential information to enable scientists reading the paper to verify the authors' claims against the data. Precise information should be provided in figures independently of the manuscript body text. Some labelling is good, some is not. For example, Figure 3 and Supp Fig 3: "Delayed chamber PTEN", "PTEN clones", "PTEN ON", lack information on whether these are for PTEN visualisation, PTEN over-expression, PTEN mutant or PTEN knock-down, in which cell type, or what type of clone. Figure legend has not been improved either, as for B there is no explanation, C says "PTEN conditional block", E "CG growth was blocked (PTEN)". "Blocking" is not a functional genetics term. In Figure 2: "pros tumours" is unclear whether this corresponds to pros GOF, LOF, mutant or RNAi, nor with which driver. This should be specified as it is for other figures (eg SuppFig2D). In Figure 3, what exact timings are T1 and T2? It should be possible to understand the data by reading the figures and figure legends, without reference to the manuscript text and the narrative. Currently it is not, and the text is often not explicit either. The authors have provided a list of full genotypes - or crosses - which is helpful and necessary, but it is slow and very hard work to check it and an abbreviated form must also be provided in all figures for all data sets. 

 Terms must be defined before use: cell fate, cell identity and cell lineage are used interchangeably and in places it is ambiguous whether the authors refer to eg NSC/GMC/neuron cell fate or NSC1 vs NSC 2, neuron A vs neuron B, lineage/cell identity. 

To conclude, in the revised version, information on the experimental source of the evidence on which claims are grounded must be explicitly provided in the figures and figure legends for all data sets and improved within text. 

2. Quantifications have improved greatly, which is excellent, but some are still missing (eg Supp Fig 3) - information must include how many samples they analysed and of these, how often they observed the phenotype (ie the penetrance), for all data sets. In some points, "penetrance 100%" is provided without sample size, which is meaningless. In some places, interpretations lack rigour as they do not always accurately reflect the data. For example, authors state that "We then found that clonal tumour-like growth coming from single dysregulated NSCs (marked by the same colour) were contained within one CG chamber, both for Type I pros tumours and Type II brat tumours (Fig. 2C-D)." However, Supp Figure 2D wor>brat-RNAi shows that only 50% of clones contain one single colour. The authors interpret that Raepli did not work properly in the other 50% of cases. If so, Raepli data should not be included in the paper. Alternatively, if there are more Dpn+ NSC within a wor>brat-RNAi clone, Raepli should mark their lineages in different colours, as it did. This would mean that CG no longer distinguish lineages within a tumorous clone - the opposite interpretation to the authors'. Authors must clarify this. They must also modify their Interpretations consistently with the quantifications, regardless of their original narrative.

3. On the recommendation to authors to either provide evidence that 3xP3-GFP has been removed from the CRIMIC lines or remove the data and the claims, the authors did not act on these. Instead, the authors argued: "…We agree that the 3xP3-GFP of the CRIMIC lines is expressed in glial cells, at least partly in the CG. However, this cannot explain the nuclear, neat His::mRFP signal, which co-localises with Repo, a marker for glial nuclei (previous Supp. Fig. 5A) and thus which can only come from the CRIMIC enhancer…". I'm afraid this is invalid, these are not clean data as it is not possible to distinguish the origin of the glial signal. Stainings using CRIMIC lines that still carry 3xP3-GFP cannot be used in experiments on glial cells, because 3xP3GFP is very strong and visible throughout channels in glia, including in nuclei. The data provided in Supplementary Figure 7D contain artefacts and must be removed, as well as the associated claim. 

1. Statistics: On the request for p values for the data sets, authors argued: "All p-values were, and still are, displayed on the graphs themselves". The authors did provide p values for the post-hoc multiple comparisons tests, but they did not and still have provided the p values for the grouped data set tests in Figure 5C, Figure 7F, Figure 9F,I, Figure 10E, J, Figure 11D,E. For example, in Figure 11D, the figure has the p values of the post-hoc Dunn tests, but neither the figure nor the figure legend provide the p value of the Kruskal Wallis H test. For all data sets with more than 3 different sample types, the authors must provide the p values of the total data set (eg Kruskal Wallis Anova p<0.001 given either over figure or in figure legend next to the name of the test applied, and post-hoc multiple comparisons Dunn test *p<0.05 provided in figure graph). Also, standard test names should be provided (eg Unpaired Student t test; Mann Whitney U test). 

Other:

Figure 1 should be discussed within Results, not Introduction.

Shg LOF: authors argued: "… we backed up these data using null shg mutant clones (in which the loss of Shg signal was also checked for extra-safety, previous Supp. Fig. 3F, now Supp. Fig. 55A), and this genetic approach cannot fit the issue with the RNAi not working." However, Supp Fig 5A shows shg knock-down, not mutant clones.

Abstract: "corset"? is not a helpful metaphor

"We expressed…PTEN" should be over-expressed.

Figure 7F: "We then used the same approach to assess NSC encasing upon double knockdown of nrx-IV and wrapper ….. These data pinpoint an additive effect of nrx-IV and wrapper knockdowns …". This is incorrect. If the phenotype of the double knock-down is stronger than those of the single knock-downs, then there is a synergistic phenotype - not additive. Synergistic (not additive) phenotypes reveal a genetic interaction; additive phenotypes mean the genes do not interact.

The manuscript and its future impact would greatly benefit from being much more concise.

---

## [Editor Report · Decision Letter 2]

28 Sep 2023

Dear Dr Spéder,

Thank you for the submission of your revised Research Article "Differential adhesion during development establishes individual neural stem cell niches and shapes adult behaviour" for publication in PLOS Biology, and for addressing the last reviewer and editorial requests in this revision. On behalf of my colleagues and the Academic Editor, Bing Ye, I am pleased to say that we can in principle accept your manuscript for publication, provided you address any remaining formatting and reporting issues. These will be detailed in an email you should receive within 2-3 business days from our colleagues in the journal operations team; no action is required from you until then. Please note that we will not be able to formally accept your manuscript and schedule it for publication until you have completed any requested changes.

PRESS

Sincerely, 

Lucas Smith, Ph.D.

Senior Editor

PLOS Biology

lsmith@plos.org